# SALKG: Learning From Knowledge Graph Explanations for Commonsense Reasoning

**Aaron Chan**♣, **Jiashu Xu**♣, **Boyuan Long**♣,
**Soumya Sanyal**♣, **Tanishq Gupta**◇*, **Xiang Ren**♣
♣University of Southern California, ◇IIT Delhi
{chanaaro, boyuanlo, jiashuxu, soumyasa, xiangren}@usc.edu,
Tanishq.Gupta.mt617@maths.iitd.ac.in

## Abstract

Augmenting pre-trained language models with knowledge graphs (KGs) has achieved success on various commonsense reasoning tasks. However, for a given task instance, the KG, or certain parts of the KG, may not be useful. Although KG-augmented models often use attention to focus on specific KG components, the KG is still always used, and the attention mechanism is never explicitly taught which KG components should be used. Meanwhile, saliency methods can measure how much a KG feature (*e.g.*, graph, node, path) influences the model to make the *correct* prediction, thus explaining which KG features are useful. This paper explores how saliency explanations can be used to improve KG-augmented models' performance. First, we propose to create coarse (*Is the KG useful?*) and fine (*Which nodes/paths in the KG are useful?*) saliency explanations. Second, to motivate saliency-based supervision, we analyze oracle KG-augmented models which directly use saliency explanations as extra inputs for guiding their attention. Third, we propose SALKG, a framework for KG-augmented models to learn from coarse and/or fine saliency explanations. Given saliency explanations created from a task's training set, SALKG jointly trains the model to predict the explanations, then solve the task by attending to KG features highlighted by the predicted explanations. On three commonsense QA benchmarks (CSQA, OBQA, CODAH) and a range of KG-augmented models, we show that SALKG can yield considerable performance gains — up to 2.76% absolute improvement on CSQA. [2]

## 1 Introduction

Natural language processing (NLP) systems generally need common sense to function well in the real world [15]. However, NLP tasks do not always provide the requisite commonsense knowledge as input. Moreover, commonsense knowledge is seldom stated in natural language, making it hard for pre-trained language models (PLMs) [11, 35] — *i.e.*, text encoders — to learn common sense from corpora alone [9, 38]. In contrast to corpora, a knowledge graph (KG) is a rich, structured source of commonsense knowledge, containing numerous facts of the form (`concept1`, `relation`, `concept2`). As a result, many methods follow the ***KG-augmented model*** paradigm, which augments a text encoder with a graph encoder that reasons over the KG (Fig. 2). KG-augmented models have outperformed text encoders on various commonsense reasoning (CSR) tasks, like question answering (QA) (Fig. 1) [31, 5, 36, 61], natural language inference (NLI) [7, 57], and text generation [33, 65].

Since KGs do not have perfect knowledge coverage, they may not contain useful knowledge for all task instances (*e.g.*, if the KG in Fig. 1 only consisted of the gray nodes). Also, even if the KG is useful overall for a given task instance, only some parts of the KG may be useful (*e.g.*, the

---

*Work done while TG interned remotely at USC.

[2]Code and data are available at: `https://github.com/INK-USC/SalKG`.

green nodes in Fig. 1). Ideally, a KG-augmented model would know both if the KG is useful and which parts of the KG are useful. Existing KG-augmented models always assume the KG should be used, but do often use attention [54] to focus on specific KG components (*e.g.*, nodes [13, 47, 60], paths [56, 46, 5]) when predicting. Still, the attention mechanism is supervised (end-to-end) only by the task loss, so the model is never *explicitly* taught which KG components should be used. Without component-level supervision, the attention mechanism is more likely to overfit to spurious patterns.

How can we better teach the model whether each KG feature (*e.g.*, graph, node, path) is useful for solving the given task instance? Using the task's ground truth labels, *saliency methods* [2] can score each KG feature's influence on the model making the correct prediction. Whereas attention weights show which KG features the model already used, saliency scores indicate which KG features the model should use. By binarizing these scores, we are able to produce saliency explanations, which can serve as simple targets for training the model's attention mechanism. For example, Fig. 1 shows saliency explanations [`market=1`, `produce=1`, `trading=0`, `merchant=1`, `store=0`, `shop=0`], stating that `market`, `produce`, and `merchant` are useful nodes for answering the question.

In this paper, we investigate how saliency explanations can be used to improve KG-augmented models' performance. First, we propose to create *coarse* (graph-level) and *fine* (node-/path-level) saliency explanations. Since KGs have features at different granularities, saliency explanations can supply a rich array of signals for learning to focus on useful KG features. To create coarse explanations, we introduce an ensemble-based saliency method which measures the performance difference between a KG-augmented model and its corresponding non-KG-augmented model. To create fine explanations, we can adapt any off-the-shelf saliency method, *e.g.*, gradient-based [10] or occlusion-based [30]. Second, to demonstrate the potential of saliency-based supervision, we analyze the performance of *oracle* KG-augmented models, whose attention weights are directly masked with coarse and/or fine saliency explanations.

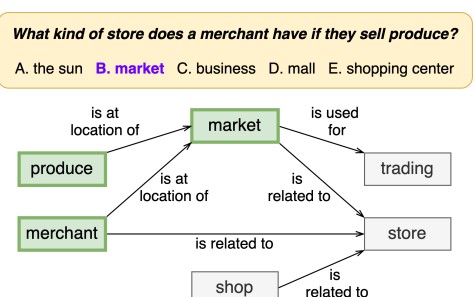

Figure 1: **KG Saliency Explanations for Commonsense QA.** Across different questions, the KG's usefulness can vary considerably. *Coarse* explanations indicate if the KG is useful overall, while *fine* explanations highlight useful nodes or paths. Here, the fine explanations state that the `market`, `produce`, and `merchant` nodes are useful, while the other nodes are not.

Third, as motivated by our oracle model analysis, we propose the *Learning from **Sal**iency Explanations of **KG**-Augmented Models* (**SALKG**) framework. Given coarse and/or fine explanations created from thse task's training set, SALKG jointly trains the model to predict the explanations, then solve the task by attending to KG features highlighted in the predicted explanations. Using saliency explanations to regularize the attention mechanism can help the model generalize better to unseen instances, especially when coarse and fine explanations are used together as complementary learning signals. Indeed, on three standard commonsense QA benchmarks (CSQA, OBQA, CODAH) and a range of KG-augmented models, we show that SALKG can achieve considerable performance gains.

## 2 Preliminaries

Since KGs abundantly provide structured commonsense knowledge, KG-augmented models are often helpful for solving CSR tasks. CSR tasks are generally formulated as multi-choice QA (discriminative) tasks [52, 39, 23], but sometimes framed as open-ended response (generative) [33, 32] tasks. Given that multi-choice QA has been more extensively studied, we consider CSR in terms of multi-choice QA. Here, we present the multi-choice QA problem setting (Fig. 1) and the structure of KG-augmented models (Fig. 2).

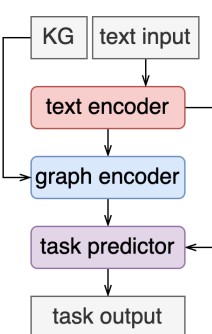

Figure 2: **KG-Augmented Models** fuse knowledge from text and KG inputs to solve CSR tasks.

**Problem Definition** Given a question $q$ and set of answer choices $A = \{a_i\}$, a multi-choice QA model aims to predict a plausibility score $\rho(q, a_i)$ for each $(q, a_i)$ pair, so that the predicted answer $\hat{a} = \arg\max_{a_i \in A} \rho(q, a_i)$ matches the target answer $a^*$. Let $q \oplus a_i$ be the text statement formed from $(q, a_i)$, where $\oplus$ denotes concatenation. For example, in Fig. 1, the text statement

for $q \oplus a^*$ would be: *What kind of store does a merchant have if they sell produce? market.* We abbreviate $q \oplus a_i$ as $x_i$ and its plausibility score as $\rho(x_i)$.

**KG-Augmented Models** KG-augmented models use additional supervision from knowledge graphs to solve the multi-choice QA task. They encode the text and KG inputs individually as embeddings, then fuse the two embeddings together to use for prediction. A KG is denoted as $\tilde{\mathcal{G}} = (\tilde{\mathcal{V}}, \tilde{\mathcal{R}}, \tilde{\mathcal{E}})$, where $\tilde{\mathcal{V}}, \tilde{\mathcal{R}}$, and $\tilde{\mathcal{E}}$ are the KG's nodes (concepts), relations, and edges (facts), respectively. An ***edge*** is a directed triple of the form $e = (c_1, r, c_2) \in \tilde{\mathcal{E}}$, in which $c_1, c_2 \in \tilde{\mathcal{V}}$ are ***nodes***, and $r \in \tilde{\mathcal{R}}$ is the ***relation*** between $c_1$ and $c_2$. A ***path*** is a connected sequence of edges in the KG. When answering a question, the model does not use the entire KG, since most information in $\tilde{\mathcal{G}}$ is irrelevant to $x_i$. Instead, the model uses a smaller, ***contextualized KG*** $\mathcal{G}_i = (\mathcal{V}_i, \mathcal{R}_i, \mathcal{E}_i)$, which is built from $\tilde{\mathcal{G}}$ using $x_i$. $\mathcal{G}_i$ can be constructed heuristically by extracting edges from $\tilde{\mathcal{G}}$ [31, 37], generating edges with a PLM [5], or both [56, 60]. In this paper, we consider KG-augmented models where $\mathcal{G}_i$ is built by heuristically by extracting edges from $\tilde{\mathcal{G}}$ (see Sec. A.1 for more details), since most KG-augmented models follow this paradigm. If $x_i$ and $\mathcal{G}_i$ are not discussed in the context of other answer choices, then we further simplify $x_i$'s and $\mathcal{G}_i$'s notation as $x$ and $\mathcal{G}$, respectively. Since the model never uses the *full* KG at once, we use "KG" to refer to $\mathcal{G}$ in the rest of the paper.

As in prior works [31, 5], a KG-augmented model $\mathcal{F}_{\text{KG}}$ has three main components: ***text encoder*** $f_{\text{text}}$, ***graph encoder*** $f_{\text{graph}}$, and ***task predictor*** $f_{\text{task}}$ (Fig. 2). Meanwhile, its corresponding non-KG-augmented model $\mathcal{F}_{\text{No-KG}}$ has no graph encoder but has a slightly different task predictor $\bar{f}_{\text{task}}$ which only takes **x** as input. In both $\mathcal{F}_{\text{KG}}$ and $\mathcal{F}_{\text{No-KG}}$, the task predictor outputs $\rho(x)$. Let **x** and **g** be the embeddings of $x$ and $\mathcal{G}$, respectively. Then, the workflows of $\mathcal{F}_{\text{KG}}$ and $\mathcal{F}_{\text{No-KG}}$ are defined below:

$$\mathbf{x} = f_{\text{text}}(x); \quad \mathbf{g} = f_{\text{graph}}(\mathcal{G}, \mathbf{x}); \quad \mathcal{F}_{\text{KG}}(x, \mathcal{G}) = f_{\text{task}}(\mathbf{x} \oplus \mathbf{g}); \quad \mathcal{F}_{\text{No-KG}}(x) = \bar{f}_{\text{task}}(\mathbf{x}).$$

Typically, $f_{\text{text}}$ is a PLM [11, 35], $f_{\text{graph}}$ is a graph neural network (GNN) [13, 47] or edge/path aggregation model [31, 5, 46], and $f_{\text{task}}$ and $\bar{f}_{\text{task}}$ are multilayer perceptrons (MLPs). In general, $f_{\text{graph}}$ reasons over $\mathcal{G}$ by encoding either nodes or paths, then using soft attention to pool the encoded nodes/paths into **g**. Let $\mathcal{L}_{\text{task}}$ be the task loss for training $\mathcal{F}_{\text{KG}}$ and $\mathcal{F}_{\text{No-KG}}$. For multi-choice QA, $\mathcal{L}_{\text{task}}$ is cross-entropy loss, with respect to the distribution over $A$. For brevity, when comparing different models, we may also refer to $\mathcal{F}_{\text{KG}}$ and $\mathcal{F}_{\text{No-KG}}$ as KG and No-KG, respectively.

## 3 Creating KG Saliency Explanations

Now, we show how to create coarse and fine saliency explanations, which tell us if the KG or certain parts of the KG are useful. These explanations can be used as extra inputs to mask oracle models' attention (Sec. 4) or as extra supervision to regularize SALKG models' attention (Sec. 5). We first abstractly define a ***unit*** as either $\mathcal{G}$ itself or a component of $\mathcal{G}$. A unit can be a graph, node, path, *etc.*, and we categorize units as ***coarse*** (the entire graph $\mathcal{G}$) or ***fine*** (a node or path within $\mathcal{G}$) (Table 1). Given a model and task instance $(x, \mathcal{G})$, we define an ***explanation*** as a *binary* indicator of whether a unit $u$ of $\mathcal{G}$ is useful for the model's prediction on $(x, \mathcal{G})$. If $u$ is useful, then $u$ should strongly influence the model to solve the instance correctly. By making explanations binary, we can easily use explanations as masks or learning targets (since binary labels are easier to predict than real-valued scores) for attention weights.

### 3.1 Coarse Saliency Explanations

Since $\mathcal{G}$ may not always be useful, a KG-augmented model should ideally know when to use $\mathcal{G}$. Here, the unit $u$ is the graph $\mathcal{G}$. Given instance $(x, \mathcal{G})$, a coarse saliency explanation $y_{\text{c}}(x, \mathcal{G}) \in \{0, 1\}$ indicates if $\mathcal{G}$ helps the model solve the instance. By default, $\mathcal{F}_{\text{KG}}$ assumes $\mathcal{G}$ is used, so we propose an ensemble-based saliency formulation for $y_{\text{c}}(x, \mathcal{G})$. That is, we define $y_{\text{c}}(x, \mathcal{G})$ as stating if $\mathcal{F}_{\text{KG}}$ (*i.e.*, uses $\mathcal{G}$) or $\mathcal{F}_{\text{No-KG}}$ (*i.e.*, does not use $\mathcal{G}$) should be used to solve $(x, \mathcal{G})$. Under this formulation, each $(x, \mathcal{G})$ has coarse units $\mathcal{G}$ and `None`, where `None` means "$\mathcal{G}$ is not used".

| Explanation Setting | Unit |
|---|---|
| Coarse | KG |
| Fine (MHGRN) | Node |
| Fine (PathGen) | Path |
| Fine (RN) | Path |

Table 1: **KG unit types** used for different explanation modes (Sec. 3) and graph encoders (Sec. 4.2).

To get $y_{\text{c}}(x, \mathcal{G})$, we begin by computing coarse saliency score $s_{\text{c}}(x, \mathcal{G}) \in \mathbb{R}$, which we define as the performance difference between $\mathcal{F}_{\text{KG}}$ and $\mathcal{F}_{\text{No-KG}}$. For QA input $x_i = q \oplus a_i$ and its KG $\mathcal{G}_i$, let $p_{\text{KG}}(x_i, \mathcal{G}_i)$ and $p_{\text{No-KG}}(x_i)$ be the confidence probabilities for $x_i$ predicted by $\mathcal{F}_{\text{KG}}$ and $\mathcal{F}_{\text{No-KG}}$, respectively.

Ideally, a QA model should predict higher probabilities for answer choices $a_i$ that are correct, and vice versa. To capture this notion, we define $s_\mathrm{c}(x_i, \mathcal{G}_i)$ in Eq. 1, where $a^*$ denotes the correct answer. Note that $s_\mathrm{c}(x_i, \mathcal{G}_i)$ is positive if $p_\mathrm{KG}(x_i, \mathcal{G}_i)$ is higher than $p_\mathrm{No\text{-}KG}(x_i)$ for correct

$$
\begin{aligned}
&s_\mathrm{c}(x_i, \mathcal{G}_i) \\
&= \begin{cases} p_\mathrm{KG}(x_i, \mathcal{G}_i) - p_\mathrm{No\text{-}KG}(x_i), & a_i = a^*, \\ p_\mathrm{No\text{-}KG}(x_i) - p_\mathrm{KG}(x_i, \mathcal{G}_i), & a_i \neq a^*. \end{cases} \quad (1)
\end{aligned}
$$

choices and lower for incorrect choices. We obtain $y_\mathrm{c}(x_i, \mathcal{G}_i)$ by binarizing $s_\mathrm{c}(x_i, \mathcal{G}_i)$ to 0 or 1 based on whether it is greater than or less than a threshold $T$, respectively. If $y_\mathrm{c}(x_i, \mathcal{G}_i) = 1$, then the KG is useful, and vice versa. See the appendix for more details about why we use ensemble-based saliency for coarse explanations (Sec. A.2) and how we tune $T$ (Sec. A.6).

### 3.2 Fine Saliency Explanations

Even if $\mathcal{G}$ is useful, not every part of $\mathcal{G}$ may be useful. Hence, fine saliency explanations can identify which parts of a KG are actually useful. For a given instance $(x, \mathcal{G})$, we denote the fine saliency explanation for a fine unit $u$ in $\mathcal{G}$ as $y_\mathrm{f}(u; x, \mathcal{G}) \in \{0, 1\}$. Fine units can be nodes, paths, *etc.* in the KG. If a graph encoder $f_\mathrm{graph}$ encodes a certain type of unit, it is natural to define $y_\mathrm{f}(u; x, \mathcal{G})$ with respect to such units. For example, MHGRN [13] encodes $\mathcal{G}$'s nodes, so we define MHGRN's fine saliency explanations with respect to nodes. Similar to coarse saliency explanations, to obtain $y_\mathrm{f}(u; x, \mathcal{G})$, we first compute fine saliency score $s_\mathrm{f}(u; x, \mathcal{G}) \in \mathbb{R}$, and then binarize it. For a QA input $x_i = q \oplus a_i$ and its KG $\mathcal{G}_i$, let $u_{ij}$ be the $j^{th}$ fine unit in $\mathcal{G}_i$ and $p_\mathrm{KG}(x_i, \mathcal{G}_i)$ denote $\mathcal{F}_\mathrm{KG}$'s predicted probability for $x_i$. There are many existing saliency methods (*a.k.a.* attribution methods) [10, 51, 30] for calculating the importance score of an input, with respect to a model and a given label. While $s_\mathrm{f}(u_{ij}; x_i, \mathcal{G}_i)$ can be computed via any saliency method, we use gradient-based and occlusion-based methods, since they are the most common types of saliency methods [2].

Let $\phi(u_{ij}; x_i, \mathcal{G}_i)$ denote the raw saliency score given by some saliency method. Gradient-based methods measure an input's saliency via the gradient of the model's output with respect to the input. We use the *gradient×input* (Grad) method [10], where $\phi(u_{ij}; x_i, \mathcal{G}_i)$ is the dot product of $u_{ij}$'s embedding and the gradients of $p_\mathrm{KG}(x_i, \mathcal{G}_i)$ with respect to $u_{ij}$. Occlusion-based methods measure an input's saliency as how the model's output is affected by erasing that input. We use the *leave-one-out* (Occl) method [30], where $\phi(u_{ij}; x_i, \mathcal{G}_i)$ is the decrease in $p_\mathrm{KG}(x_i, \mathcal{G}_i)$ if $u_{ij}$ is removed from $\mathcal{G}_i$, *i.e.*, $\phi(u_{ij}; x_i, \mathcal{G}_i) = p_\mathrm{KG}(x_i, \mathcal{G}_i)$ - $p_\mathrm{KG}(x_i, \mathcal{G}_i \setminus u_{ij})$.

Intuitively, a unit is more useful if it increases the probability of correct answer choice $a^*$, and vice versa. Thus, we define the saliency score $s_\mathrm{f}(u_{ij}; x_i, \mathcal{G}_i)$ for unit $u_{ij}$ as Eq. 2. Next, we binarize the saliency scores to get $y_\mathrm{f}(u_{ij}; x_i, \mathcal{G}_i)$, by selecting the top-$k$%-scoring units in $\mathcal{G}_i$ and setting $y_\mathrm{f}(u_{ij}; x_i, \mathcal{G}_i) = 1$ (*i.e.*, $u_{ij}$ is useful) for these units. For

$$
\begin{aligned}
&s_\mathrm{f}(u_{ij}; x_i, \mathcal{G}_i) \\
&= \begin{cases} \phi(u_{ij}; x_i, \mathcal{G}_i), & a_i = a^* \\ -\phi(u_{ij}; x_i, \mathcal{G}_i), & a_i \neq a^* \end{cases} \quad (2)
\end{aligned}
$$

all other units in $\mathcal{G}$, we set $y_\mathrm{f}(u_{ij}; x_i, \mathcal{G}_i) = 0$ (*i.e.*, $u_{ij}$ is not useful). See the appendix for more details about the fine saliency methods (Sec. A.3) and tuning threshold $k$ (Sec. A.6).

## 4 ORACLE: Using KG Saliency Explanations as Inputs

In this section, we analyze KG saliency explanations' potential to improve KG-augmented models' performance. Recall that creating saliency explanations requires the task's ground truth labels (Sec. 3), so directly using test set explanations is infeasible. Still, before exploring ways to leverage training set explanations (Sec. 5), we first establish upper bounds on how much models can benefit from saliency explanations. Here, we study three key questions: **(1)** *Does the model improve when provided oracle access to coarse/fine explanations?* **(2)** *Are coarse and fine explanations complementary?* **(3)** *How do gradient-based explanations compare to occlusion-based explanations?*

### 4.1 ORACLE Models

ORACLE models are KG-augmented models with oracle access to saliency explanations. An ORACLE model uses ground truth labels to create explanations (even at inference time), and then uses the explanations as extra inputs to perform hard attention over the units. We define the model attention weights that are modified based on saliency explanations as *saliency weights*. Below, we introduce the ORACLE-Coarse, ORACLE-Fine, and ORACLE-Hybrid models, shown in Fig. 3a-c.

| Model | Output | Saliency Weights |
|---|---|---|
| ORACLE-Coarse | $\mathcal{F}_c^*(x,\mathcal{G}) = y_c(x,\mathcal{G})\mathcal{F}_{KG}(x,\mathcal{G}) + (1-y_c(x,\mathcal{G}))\mathcal{F}_{No\text{-}KG}(x)$ | $[y_c(x,\mathcal{G}), 1-y_c(x,\mathcal{G})]$ |
| ORACLE-Fine | $\mathcal{F}_f^*(x,\mathcal{G}) \sim \mathcal{F}_{KG}(x,\mathcal{G})$ | $\hat{y}_f(x,\mathcal{G}) \odot y_f(x,\mathcal{G})$ |
| ORACLE-Hybrid | $\mathcal{F}_h^*(x,\mathcal{G}) = y_h(x,\mathcal{G})\mathcal{F}_f^*(x,\mathcal{G}) + (1-y_h(x,\mathcal{G}))\mathcal{F}_{No\text{-}KG}(x)$ | $[y_h(x,\mathcal{G}), 1-y_h(x,\mathcal{G})]$ |

Table 2: **Comparison of ORACLE Models.** For each ORACLE Model, we show its output and saliency weights. Note that the explanations are given (not predicted), so there is no $\mathcal{L}_{sal}$. While $\mathcal{F}_c^*$ and $\mathcal{F}_h^*$ are both ensembles of $\mathcal{F}_{KG}$ and $\mathcal{F}_{No\text{-}KG}$, $\mathcal{F}_f^*$ has the same architecture as $\mathcal{F}_{KG}$ (denoted by $\sim$) besides the attention masking.

| | CSQA Test Accuracy (%) | | | | | | OBQA Test Accuracy (%) | | | | | |
|---|---|---|---|---|---|---|---|---|---|---|---|---|
| | MHGRN | | PathGen | | RN | | MHGRN | | PathGen | | RN | |
| Model | BERT | RoBERTa | BERT | RoBERTa | BERT | RoBERTa | BERT | RoBERTa | BERT | RoBERTa | BERT | RoBERTa |
| No-KG | 55.44 | 70.59 | 55.44 | 70.59 | 55.44 | 70.59 | 53.60 | 68.40 | 53.60 | 68.40 | 53.60 | 68.40 |
| KG | 56.57 | 73.33 | 56.65 | 72.04 | 55.60 | 71.07 | 53.20 | 69.80 | 55.00 | 67.80 | 58.60 | 70.20 |
| No-KG + KG | 56.57 | 71.39 | 57.45 | 73.00 | 56.73 | 68.49 | 55.60 | 70.60 | 54.40 | 70.6 | 53.40 | 69.60 |
| ORACLE-Coarse | 66.16 | 81.39 | 68.57 | 80.10 | 67.28 | 79.69 | 70.60 | 79.40 | 65.00 | 76.60 | 69.00 | 79.00 |
| ORACLE-Fine (Grad) | 74.86 | 76.15 | 79.61 | 87.35 | 81.39 | 83.24 | 67.60 | 72.60 | 73.80 | 73.40 | 68.00 | 62.80 |
| ORACLE-Fine (Occl) | 91.06 | 87.99 | 79.61 | 75.34 | 73.73 | 68.41 | 77.00 | 71.20 | 83.60 | 62.60 | 55.60 | 61.40 |
| ORACLE-Hybrid (Grad) | 85.50 | 84.21 | **90.49** | 92.83 | **92.26** | 93.56 | 80.80 | 84.80 | 85.60 | **92.80** | **85.40** | **86.80** |
| ORACLE-Hybrid (Occl) | 95.89 | 98.63 | 88.96 | **96.78** | 85.25 | **95.25** | 87.00 | 89.60 | 92.80 | 90.60 | 67.40 | 80.60 |

Table 3: **ORACLE Performance on CSQA and OBQA**

**ORACLE-Coarse** ORACLE-Coarse ($\mathcal{F}_c^*$) uses coarse explanations to do hard attention over $\mathcal{F}_{KG}$'s and $\mathcal{F}_{No\text{-}KG}$'s predictions. First, $\mathcal{F}_{KG}$ and $\mathcal{F}_{No\text{-}KG}$ are trained separately, then frozen. Next, for each instance $(x,\mathcal{G})$, they are used to create a coarse explanation $y_c(x,\mathcal{G}) \in \{0,1\}$. Then, $\mathcal{F}_c^*$ is defined as an ensemble model that performs hard attention over coarse units ($\mathcal{G}$ and `None`) by weighting $\mathcal{F}_{KG}$'s prediction with $y_c(x,\mathcal{G})$ and $\mathcal{F}_{No\text{-}KG}$'s prediction with $1 - y_c(x,\mathcal{G})$ (Table 2; Fig. 3a). In other words, $y_c(x,\mathcal{G})$ and $1 - y_c(x,\mathcal{G})$ are the saliency weights for $\mathcal{F}_c^*$.

**ORACLE-Fine** ORACLE-Fine ($\mathcal{F}_f^*$) has the same architecture as $\mathcal{F}_{KG}$ and uses fine explanations to do hard attention over fine units (*i.e.*, nodes or paths in $\mathcal{G}$). First, $\mathcal{F}_{KG}$ is trained, then frozen. As usual, $\mathcal{F}_{KG}$ uses soft attention over fine units in $\mathcal{G}$ to compute graph embedding **g** (Sec. 2). Then, for each fine unit $u$ in $\mathcal{G}$, $\mathcal{F}_{KG}$ is used to create fine explanation $y_f(u;x,\mathcal{G}) \in \{0,1\}$. Let $\hat{y}_f(u;x,\mathcal{G}) \in [0,1]$ denote $\mathcal{F}_f^*$'s soft attention weight for $u$. We train $\mathcal{F}_f^*$ the same way as $\mathcal{F}_{KG}$, except each $\hat{y}_f(u;x,\mathcal{G})$ is (hard attention) masked with $y_f(u;x,\mathcal{G})$, *i.e.*, $\hat{y}_f(u;x,\mathcal{G}) \leftarrow \hat{y}_f(u;x,\mathcal{G}) \odot y_f(u;x,\mathcal{G})$, where $\odot$ denotes element-wise multiplication (Table 2; Fig. 3b). This means only units with $y_f(u;x,\mathcal{G}) = 1$ will have $\hat{y}_f(u;x,\mathcal{G}) > 0$ and thus be able to influence $\mathcal{F}_f^*$'s prediction. Let $y_f(x,\mathcal{G})$ and $\hat{y}_f(x,\mathcal{G})$ denote the explanations and soft attention weights, respectively, for all units in the graph. Then, $\hat{y}_f(x,\mathcal{G}) \odot y_f(x,\mathcal{G})$ are the saliency weights for $\mathcal{F}_f^*$.

**ORACLE-Hybrid** ORACLE-Hybrid ($\mathcal{F}_h^*$) unifies ORACLE-Coarse and ORACLE-Fine as a single model, thus leveraging the coarse-fine hierarchy inherent in KG saliency explanations. First, $\mathcal{F}_f^*$ (which uses fine explanations) and $\mathcal{F}_{No\text{-}KG}$ are separately trained, then frozen. Then, for each $(x,\mathcal{G})$, $\mathcal{F}_f^*$ and $\mathcal{F}_{No\text{-}KG}$ are used to create $y_h(x,\mathcal{G}) \in \{0,1\}$, which we define as the coarse explanation for $\mathcal{F}_f^*$ and $\mathcal{F}_{No\text{-}KG}$. $y_h(x,\mathcal{G})$ is computed the same way as $y_c(x,\mathcal{G})$, besides replacing $\mathcal{F}_{KG}$ with $\mathcal{F}_f^*$. Finally, similar to $\mathcal{F}_c^*$, $\mathcal{F}_h^*$ is an ensemble that performs hard attention over coarse units by weighting $\mathcal{F}_f^*$'s prediction with $y_h(x,\mathcal{G})$ and $\mathcal{F}_{No\text{-}KG}$'s prediction with $1 - y_h(x,\mathcal{G})$ (Table 2; Fig. 3c). That is, $y_h(x,\mathcal{G})$ and $1 - y_h(x,\mathcal{G})$ are the saliency weights for $\mathcal{F}_h^*$.

## 4.2 Evaluation Protocol

We use the CSQA [52] and OBQA [39] multi-choice QA datasets. For CSQA, we use the accepted in-house data split from [31], as the official test labels are not public. As in prior works, we use the ConceptNet [49] KG for both datasets. We report accuracy, the standard metric for multi-choice QA. For $\mathcal{F}_{No\text{-}KG}$ and $\mathcal{F}_{KG}$, we pick the best model over three seeds, then use them to create explanations for ORACLE models. We use thresholds $T = 0.01$ and $k = 10$ for coarse and fine explanations, respectively. For text encoders, we use BERT(-Base) [11] and RoBERTa(-Large) [35]. For graph encoders, we use MHGRN [13], PathGen [56], and Relation Network (RN) [46, 31]. MHGRN has node units, while PathGen and RN have path units. As ***baseline models***, we use $\mathcal{F}_{No\text{-}KG}$, $\mathcal{F}_{KG}$, and $\mathcal{F}_{No\text{-}KG} + \mathcal{F}_{KG}$, where $\mathcal{F}_{No\text{-}KG} + \mathcal{F}_{KG}$ is an ensemble whose prediction is the mean of $\mathcal{F}_{No\text{-}KG}$'s and $\mathcal{F}_{KG}$'s predictions. ORACLE and baseline models are trained only with task loss $\mathcal{L}_{task}$.

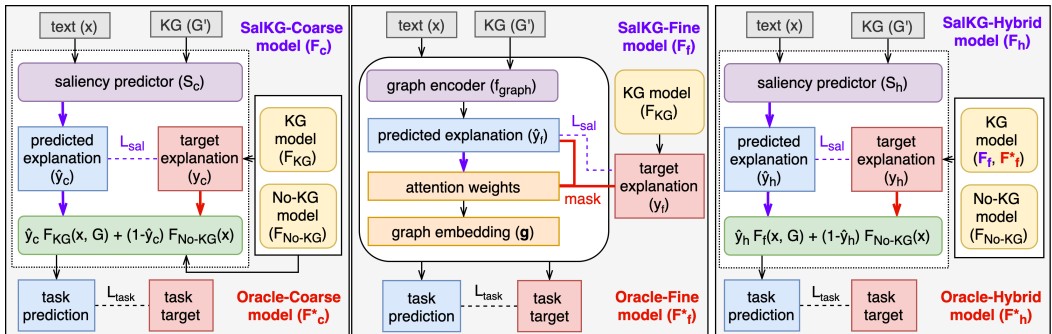

Figure 3: **Schematics for ORACLE and SALKG Models.** Red arrows indicate the ORACLE pipeline, where the target explanation is provided as input. Purple arrows indicate the SALKG pipeline, where the target explanation is used as supervision for the predicted explanation. In SALKG-Coarse and SALKG-Hybrid, the saliency predictor has the same architecture as $\mathcal{F}_{\text{KG}}$. Meanwhile, ORACLE-Fine and SALKG-Fine (shown as white module, with text encoder and task predictor omitted) both have the same architecture as $\mathcal{F}_{\text{KG}}$.

## 4.3   Analysis

In Table 3, we show CSQA and OBQA performance for the baseline and ORACLE models. We analyze these results via the three questions below.

***Does the model improve when provided oracle access to coarse/fine explanations?*** Yes. ORACLE-Coarse beats all baselines, while ORACLE-Fine beats all baselines except on OBQA RN+RoBERTa. These results motivate us to develop a framework for models to improve performance by learning from coarse/fine explanations. Also, on average, ORACLE-Fine outperforms ORACLE-Coarse, which suggests that fine explanations may often provide richer signal than their coarse counterparts. Indeed, fine explanations indicate the saliency of every unit in the KG, while coarse explanations only indicate the saliency of the KG as a whole.

***Are coarse and fine explanations complementary?*** Yes. Across all settings, ORACLE-Hybrid performs significantly better than ORACLE-Coarse and ORACLE-Fine. This suggests that coarse and fine explanations are complementary and that it is effective to leverage both hierarchically.

***How do gradient-based explanations compare to occlusion-based explanations?*** Overall, Grad and Occl perform similarly. Grad performs better on some settings (*e.g.*, MHGRN), while Occl performs better on others (*e.g.*, RN). See Table 8 and Sec. A.9 for more Grad *vs.* Occl experiments.

In our ORACLE pilot study, KG-augmented models achieve large performance gains when given explanations as input. This suggests that, if oracle explanations can somehow be *predicted* accurately during inference without using ground truth labels, then KG-augmented models can still achieve improvements without directly using explanations as input. This motivates us to train KG-augmented models with explanation-based supervision via SALKG, which we describe in Sec. 5.

## 5   SALKG: Using KG Saliency Explanations as Supervision

Based on the analysis from Sec. 4.3, we propose the SALKG framework for KG-augmented models to learn from coarse/fine saliency explanations. Whereas ORACLE models (Sec. 4.1) use explanations directly as extra inputs, SALKG models only use them as extra supervision during the training phase. With explanations created from the training set via $\mathcal{F}_{\text{KG}}$ and $\mathcal{F}_{\text{No-KG}}$, SALKG models are jointly trained to predict the explanations (via saliency loss $\mathcal{L}_{\text{sal}}$) and use the predicted explanations to solve the task (via task loss $\mathcal{L}_{\text{task}}$). Thus, SALKG models have the following objective: $\mathcal{L}_{\text{S}} = \mathcal{L}_{\text{task}} + \lambda \mathcal{L}_{\text{sal}}$, where $\lambda \geq 0$ is a loss weighting parameter. This multitask objective not only encourages SALKG models to focus on useful KG units for solving the task, but also to learn more general graph/node/path representations. Below, we present SALKG-Coarse, SALKG-Fine, and SALKG-Hybrid models.

**SALKG-Coarse** Unlike ORACLE-Coarse, SALKG-Coarse ($\mathcal{F}_{\text{c}}$) is not given oracle coarse explanation $y_{\text{c}}(x, \mathcal{G})$ as input. Instead, a saliency predictor $\mathcal{S}_{\text{c}}$ (with the same architecture as $\mathcal{F}_{\text{KG}}$) is trained to predict the oracle coarse explanation. $\mathcal{S}_{\text{c}}$ predicts coarse explanation as probability $\hat{y}_{\text{c}}(x, \mathcal{G}) \in [0, 1]$. $\mathcal{F}_{\text{c}}$'s output is an ensemble that does soft attention over coarse units by weighting $\mathcal{F}_{\text{KG}}$'s and $\mathcal{F}_{\text{No-KG}}$'s predictions with saliency weights $\hat{y}_{\text{c}}(x, \mathcal{G})$ and $1 - \hat{y}_{\text{c}}(x, \mathcal{G})$, respectively (Table 4; Fig. 3a). Here, $\mathcal{L}_{\text{sal}}(\hat{y}_{\text{c}}(x, \mathcal{G}), y_{\text{c}}(x, \mathcal{G}))$ is the cross-entropy loss.

| Model | Output | Saliency Weights | Saliency Loss ($\mathcal{L}_{\text{sal}}$) |
|---|---|---|---|
| SALKG-Coarse | $\mathcal{F}_{\text{c}}(x,\mathcal{G}) = \hat{y}_{\text{c}}(x,\mathcal{G})\,\mathcal{F}_{\text{KG}}(x,\mathcal{G}) + (1 - \hat{y}_{\text{c}}(x,\mathcal{G}))\,\mathcal{F}_{\text{No-KG}}(x)$ | $[\hat{y}_{\text{c}}(x,\mathcal{G}), 1 - \hat{y}_{\text{c}}(x,\mathcal{G})]$ | CE$(\hat{y}_{\text{c}}(x,\mathcal{G}), y_{\text{c}}(x,\mathcal{G}))$ |
| SALKG-Fine | $\mathcal{F}_{\text{f}}(x,\mathcal{G}) \sim \mathcal{F}_{\text{KG}}(x,\mathcal{G})$ | $\hat{y}_{\text{f}}(x,\mathcal{G})$ | KL$(\hat{y}_{\text{f}}(x,\mathcal{G}), y_{\text{f}}(x,\mathcal{G}))$ |
| SALKG-Hybrid | $\mathcal{F}_{\text{h}}(x,\mathcal{G}) = \hat{y}_{\text{h}}(x,\mathcal{G})\mathcal{F}_{\text{f}}(x,\mathcal{G}) + (1 - \hat{y}_{\text{h}}(x,\mathcal{G}))\mathcal{F}_{\text{No-KG}}(x)$ | $[\hat{y}_{\text{h}}(x,\mathcal{G}), 1 - \hat{y}_{\text{h}}(x,\mathcal{G})]$ | CE$(\hat{y}_{\text{h}}(x,\mathcal{G}), y_{\text{h}}(x,\mathcal{G}))$ |

Table 4: **Comparison of SALKG Models.** For each SALKG Model, we show its output, saliency weights, and $\mathcal{L}_{\text{sal}}$. While $\mathcal{F}_{\text{c}}$ and $\mathcal{F}_{\text{h}}$ are both ensembles, $\mathcal{F}_{\text{f}}$ has the same architecture as $\mathcal{F}_{\text{KG}}$ (denoted by $\sim$). "CE" denotes cross-entropy loss, while "KL" denotes KL divergence loss.

**SALKG-Fine** Similarly, SALKG-Fine ($\mathcal{F}_{\text{f}}$) is not given oracle fine explanation $y_{\text{f}}(u; x, \mathcal{G})$ as input, although both have the same architecture as $\mathcal{F}_{\text{KG}}$. Instead, for each fine unit $u$, $\mathcal{F}_{\text{f}}$'s attention mechanism is trained to predict $y_{\text{f}}(u; x, \mathcal{G})$ as soft attention weight $\hat{y}_{\text{f}}(u; x, \mathcal{G}) \in [0, 1]$ (Table 4; Fig. 3b). As before, $\hat{y}_{\text{f}}(x, \mathcal{G}) = [\hat{y}_{\text{f}}(u; x, \mathcal{G})]_{u \in \mathcal{G}}$ are the soft attention weights for $(x, \mathcal{G})$, while $y_{\text{f}}(x, \mathcal{G}) = [y_{\text{f}}(u; x, \mathcal{G})]_{u \in \mathcal{G}}$ are the fine explanations for $(x, \mathcal{G})$. Then, $\hat{y}_{\text{f}}(x, \mathcal{G})$ are the saliency weights for $\mathcal{F}_{\text{f}}$, trained with KL divergence loss $\mathcal{L}_{\text{sal}}(\hat{y}_{\text{f}}(x, \mathcal{G}), y_{\text{f}}(x, \mathcal{G}))$.

**SALKG-Hybrid** Similar to the other SALKG variants, SALKG-Hybrid ($\mathcal{F}_{\text{h}}$) does not use any oracle explanations. Like in SALKG-Coarse, a saliency predictor $\mathcal{S}_{\text{h}}$ is trained to predict oracle coarse explanation $y_h(x, \mathcal{G})$ (Sec. 4.1). Predicted coarse explanation probabilities $\hat{y}_{\text{h}}(x, \mathcal{G}) \in [0, 1]$ are then used as soft attention over coarse units by weighting $\mathcal{F}_{\text{f}}$'s and $\mathcal{F}_{\text{No-KG}}$'s predictions with weights $\hat{y}_{\text{h}}(x, \mathcal{G})$ and $1 - \hat{y}_{\text{h}}(x, \mathcal{G})$, respectively (Table 4; Fig. 3c). Here, $\mathcal{L}_{\text{sal}}(\hat{y}_{\text{h}}(x, \mathcal{G}), y_{\text{h}}(x, \mathcal{G}))$ is cross-entropy loss.

# 6 Experiments

## 6.1 Evaluation Protocol

We evaluate SALKG models on the CSQA [52], OBQA [39], and CODAH [6] multi-choice QA datasets (Sec. A.5). In addition to the baselines in Sec. 4.2, we consider two new baselines, RANDOM and HEURISTIC, which help show that coarse/fine saliency explanations provide strong learning signal for KG-augmented models to focus on useful KG features. We follow the same evaluation protocol in Sec. 4.2, except we now also report mean and standard deviation performance over multiple seeds. See Sec. A.4 for a more detailed description of the evaluation protocol.

**RANDOM** RANDOM is a variant of SALKG where each unit's explanation is random. RANDOM-Coarse is like SALKG-Coarse, but with each $y_{\text{c}}(x, \mathcal{G})$ uniformly sampled from $\{0, 1\}$. RANDOM-Fine is like SALKG-Fine, but randomly picking $k\%$ of units in $\mathcal{G}$ to set $y_{\text{f}}(u; x, \mathcal{G}) = 1$. RANDOM-Hybrid is like SALKG-Hybrid, but with each $y_{\text{h}}(x, \mathcal{G})$ uniformly sampled from $\{0, 1\}$ as well as using RANDOM-Fine instead of SALKG-Fine.

**HEURISTIC** Each $\mathcal{G}$ has three node types: question nodes (*i.e.*, nodes in $q$), answer nodes (*i.e.*, nodes in $a_i$), and intermediate nodes (*i.e.*, other nodes) [31]. Let QA nodes be nodes in $q$ or $a_i$. HEURISTIC is a variant of SALKG where each unit's explanation is based on the presence of QA nodes in $\mathcal{G}$. Let $\bar{N}$ be the mean number of QA nodes per KG (in train set), and let $N(\mathcal{G})$ be the number of QA nodes in $\mathcal{G}$. HEURISTIC-Coarse is like SALKG-Coarse, except $y_{\text{c}}(x, \mathcal{G}) = 1$ if and only if $N(\mathcal{G}) > \bar{N}$. HEURISTIC-Fine is like SALKG-Fine, but how $y_{\text{f}}(u; x, \mathcal{G})$ is set depends on whether the fine units are nodes or paths. For node units, $y_{\text{f}}(u; x, \mathcal{G}) = 1$ if and only if $u$ is a QA node. For path units, $y_{\text{f}}(u; x, \mathcal{G}) = 1$ if and only if $u$ consists only of QA nodes. HEURISTIC-Hybrid is like SALKG-Hybrid, but with $y_{\text{h}}(x, \mathcal{G}) = 1$ if and only if $N(\mathcal{G}) > \bar{N}$, while HEURISTIC-Fine is used instead of SALKG-Fine.

## 6.2 Main Results

Table 5 shows performance on CSQA, while Table 6 shows performance on OBQA and CODAH. Best performance is highlighted in green , second-best performance is highlighted in blue , and

best non-SALKG performance is highlighted in red (if it is not already green or blue). For SALKG (unlike ORACLE), we find that Occl usually outperforms Grad, so we only report Occl performance in Tables 5-6. For a comparison of Grad and Occl on SALKG, see Table 8 and Sec. A.9. Being an ensemble, No-KG + KG tends to beat both No-KG and KG if both have similar performance. Otherwise, No-KG + KG's performance is in between No-KG's and KG's.

Across all datasets, we find that SALKG-Hybrid and SALKG-Coarse are consistently the two best models. On CSQA, SALKG-Hybrid has the highest performance on BERT+MHGRN,

| | CSQA Test Accuracy (%) | | | | | |
|---|---|---|---|---|---|---|
| | MHGRN | | PathGen | | RN | |
| Model | BERT | RoBERTa | BERT | RoBERTa | BERT | RoBERTa |
| No-KG | 53.13 (±2.34) | 69.65 (±1.06) | 53.13 (±2.34) | 69.65 (±1.06) | 53.13 (±2.34) | 69.65 (±1.06) |
| KG | 57.48 (±0.89) | 73.14 (±0.78) | 56.54 (±0.73) | 72.58 (±0.57) | 56.46 (±1.22) | 71.37 (±1.20) |
| No-KG + KG | 56.14 (±2.28) | 72.15 (±0.67) | 57.29 (±1.30) | 72.44 (±0.72) | 55.98 (±1.98) | 71.15 (±0.81) |
| Random-Coarse | 55.04 (±1.44) | 71.06 (±1.09) | 55.09 (±1.08) | 71.15 (±1.06) | 55.15 (±1.23) | 69.06 (±2.96) |
| Random-Fine | 54.69 (±2.54) | 73.09 (±1.06) | 54.66 (±0.97) | 71.26 (±3.19) | 49.88 (±1.75) | 69.08 (±1.95) |
| Random-Hybrid | 52.43 (±2.60) | 71.93 (±0.77) | 55.24 (±0.58) | 71.35 (±0.34) | 54.36 (±0.35) | 70.12 (±0.35) |
| Heuristic-Coarse | 55.55 (±2.29) | 72.15 (±0.84) | 56.92 (±0.18) | 72.57 (±0.49) | 56.42 (±1.11) | 71.18 (±0.77) |
| Heuristic-Fine | 52.54 (±1.67) | 71.50 (±1.01) | 54.00 (±1.89) | 71.11 (±0.93) | 52.04 (±2.13) | 65.08 (±3.67) |
| Heuristic-Hybrid | 56.35 (±0.81) | 72.58 (±0.32) | 56.83 (±0.48) | 71.33 (±0.87) | 54.38 (±3.30) | 65.07 (±2.02) |
| SALKG-Coarse | 57.98 (±0.90) | 73.64 (±1.05) | 57.75 (±0.77) | 73.07 (±0.25) | 57.50 (±1.25) | 73.11 (±1.13) |
| SALKG-Fine | 54.36 (±2.34) | 70.00 (±0.81) | 54.39 (±2.03) | 72.12 (±0.91) | 54.30 (±1.41) | 71.64 (±1.51) |
| SALKG-Hybrid | 58.70 (±0.65) | 73.37 (±0.12) | 59.87 (±0.42) | 72.67 (±0.65) | 58.78 (±0.14) | 74.13 (±0.71) |

Table 5: SALKG Performance on CSQA

| | OBQA Test Accuracy (%) | | | CODAH Test Accuracy (%) | |
|---|---|---|---|---|---|
| Model (RoBERTa) | MHGRN | PathGen | RN | MHGRN | PathGen |
| No-KG | 68.73 (±0.31) | 68.73 (±0.31) | 68.73 (±0.31) | 83.96 (±0.79) | 83.96 (±0.79) |
| KG | 68.87 (±2.16) | 68.40 (±1.59) | 66.80 (±4.73) | 84.02 (±1.27) | 84.02 (±1.62) |
| No-KG + KG | 68.53 (±0.95) | 69.67 (±1.45) | 69.40 (±0.35) | 84.08 (±1.46) | 84.69 (±1.48) |
| Random-Coarse | 68.11 (±1.12) | 67.18 (±4.13) | 65.02 (±2.57) | 83.48 (±0.91) | 84.68 (±1.65) |
| Random-Fine | 57.60 (±5.33) | 55.13 (±7.00) | 48.53 (±4.82) | 74.77 (±6.90) | 80.48 (±1.23) |
| Random-Hybrid | 68.33 (±0.40) | 69.53 (±0.31) | 69.27 (±0.12) | 83.86 (±0.69) | 83.75 (±0.60) |
| Heuristic-Coarse | 69.24 (±2.47) | 65.58 (±6.08) | 64.29 (±3.06) | 82.64 (±0.10) | 82.52 (±0.18) |
| Heuristic-Fine | 57.27 (±3.76) | 51.80 (±2.95) | 50.53 (±3.51) | 82.25 (±1.43) | 82.55 (±2.03) |
| Heuristic-Hybrid | 68.47 (±0.23) | 68.40 (±0.00) | 68.60 (±0.20) | 82.16 (±2.11) | 82.73 (±1.51) |
| SALKG-Coarse | 69.93 (±0.56) | 70.02 (±0.55) | 71.29 (±0.57) | 85.79 (±1.83) | 85.43 (±1.88) |
| SALKG-Fine | 64.82 (±0.97) | 51.51 (±0.87) | 62.29 (±0.85) | 84.08 (±1.14) | 83.36 (±0.81) |
| SALKG-Hybrid | 70.20 (±0.69) | 69.80 (±0.49) | 70.47 (±0.91) | 85.17 (±0.54) | 84.42 (±0.64) |

Table 6: SALKG Performance on OBQA and CODAH

BERT+PathGen, BERT+RN, and RoBERTa+RN, while SALKG-Coarse is the best on RoBERTa+MHGRN and RoBERTa+PathGen. In particular, on RoBERTa+RN, BERT+RN, and BERT+PathGen, SALKG-Hybrid beats $\max$(No-KG, KG, No-KG + KG) by large margins of 2.76%, 2.58%, and 2.32%, respectively. Meanwhile, OBQA and CODAH, SALKG is not as dominant but still yields improvements overall. On OBQA, SALKG-Coarse is the best on RoBERTa+RN (beating $\max$(No-KG, KG, No-KG + KG) by 1.89%) and RoBERTa+PathGen, while SALKG-Hybrid performs best on RoBERTa+MHGRN. On CODAH, SALKG-Coarse gets the best performance on both RoBERTa+MHGRN (beating $\max$(No-KG, KG, No-KG + KG) by 1.71%) and RoBERTa+PathGen. SALKG-Coarse outperforming SALKG-Hybrid on OBQA and CODAH indicates that local KG supervision from fine explanations may not be as useful for these two datasets. On the other hand, SALKG-Fine is consistently weaker than SALKG-Hybrid and SALKG-Coarse, but still shows slight improvement for RoBERTa+RN on CSQA. These results show that learning from KG saliency explanations is generally effective for improving KG-augmented models' performance, especially in CSQA when both coarse and fine explanations are used to provide complementary learning signals for SALKG-Hybrid. Furthermore, across all datasets, we find that SALKG outperforms RANDOM and HEURISTIC on every setting. This is evidence that explanations created from saliency methods can provide better learning signal than those created randomly or from simple heuristics.

**Comparison to Published CSQA Baselines**   To further demonstrate that SALKG models perform competitively, we also compare SALKG (using MHGRN and PathGen) to the many KG-augmented model baseline results published in [13, 56, 60], for the CSQA in-house split. The baselines we consider are RN [46], RN + Link Prediction [13], RGCN [47], GAT [55], GN [4], GconAttn [57], MHGRN [13], and PathGen [56]. For the non-SALKG versions of MHGRN, PathGen, and RN, we quote the published results. Since these published results average over four seeds (instead of three), we report SALKG results over four seeds in Table 7. We find that most of the listed SALKG variants can outperform all of the baselines. For MHGRN, SALKG-Coarse (MHGRN) performs the best overall, SALKG-Hybrid (MHGRN) beats vanilla MHGRN, and SALKG-Fine (MHGRN) is on par with vanilla MHGRN. For PathGen, SALKG-Hybrid (PathGen) and SALKG-Coarse (PathGen) both slightly outperform vanilla PathGen, while SALKG-Fine (PathGen) performs worse.

**CSQA Leaderboard Submission** In addition to our experiments on the CSQA in-house split, we evaluated SALKG on the CSQA official split by submitting SALKG to the CSQA leaderboard. Since the best models on the CSQA leaderboard use the ALBERT [24] text encoder, and PathGen was the highest graph encoder on the leaderboard out of the three we experimented with, we trained SALKG-Hybrid (ALBERT+PathGen), which achieved a test accuracy of 75.9%. For reference, a previously submitted ALBERT+PathGen achieved a test accuracy of 75.6% on the CSQA leaderboard. This result suggests that the proposed SALKG training procedure can yield some improvements over baselines that do not use explanation-based regularization.

| Model (RoBERTa) | CSQA Test Accuracy (%) |
|---|---|
| RN [46] | 70.08 ($\pm$0.21) |
| RN + Link Prediction [56] | 69.33 ($\pm$0.98) |
| RGCN [47] | 68.41 ($\pm$0.66) |
| GAT [55] | 71.20 ($\pm$0.72) |
| GN [4] | 71.12 ($\pm$0.45) |
| GconAttn [57] | 69.88 ($\pm$0.47) |
| MHGRN [13] | 71.11 ($\pm$0.81) |
| PathGen [56] | 72.68 ($\pm$0.42) |
| SALKG-Coarse (MHGRN) | **74.01** ($\pm$0.14) |
| SALKG-Fine (MHGRN) | 72.68 ($\pm$1.46) |
| SALKG-Hybrid (MHGRN) | **73.87** ($\pm$0.48) |
| SALKG-Coarse (PathGen) | **72.76** ($\pm$0.12) |
| SALKG-Fine (PathGen) | 71.21 ($\pm$1.31) |
| SALKG-Hybrid (PathGen) | **73.03** ($\pm$0.84) |

Table 7: **Comparison of SALKG to Published CSQA Baselines.** SALKG models that outperform all baselines are shown in **bold**.

***Why does SALKG-Fine perform poorly?*** In general, SALKG-Fine does not perform as well as SALKG-Coarse and SALKG-Hybrid. Often, SALKG-Fine is noticeably worse than KG and No-KG. Recall that the KG model and SALKG-Fine model both assume that the KG should always be used to solve the given instance. Still, the success of SALKG-Coarse shows that the KG sometimes may not be useful. But why does SALKG-Fine almost always perform worse than the KG model?

We believe it is because SALKG-Fine is more committed to the flawed assumption of universal KG usefulness. Whereas the KG model is trained to solve the task always using the KG as context, SalKG-Fine is trained to both solve the task always using the KG as context (*i.e.*, global KG supervision) and attend to specific parts of the KG (*i.e.*, local KG supervision). Since SALKG-Fine is trained with both global and local KG supervision, it is much more likely to overfit, as the KG is not actually useful for all instances. That is, for training instances where the KG should not be used, SALKG-Fine is pushed to not only use the KG, but also to attend to specific parts of the KG. This leads to a SalKG-Fine model that does not generalize well to test instances where the KG is not useful.

To address this issue, we proposed the SALKG-Hybrid model, which is designed to take the best of both SALKG-Coarse and SALKG-Fine. For a given instance, SALKG-Hybrid uses its SALKG-Coarse component to predict whether the KG is useful, then uses its SALKG-Fine component to attend to the useful parts of the KG only if the KG is predicted to be useful. Indeed, we find that SALKG-Hybrid performs much better than SALKG-Fine and is the best model overall on CSQA. These results support our hypothesis about why SALKG-Fine performs relatively poorly.

## 6.3 Ablation Studies

In Table 8, we validate our SALKG design choices with ablation studies. We report dev accuracy for BERT+MHGRN and BERT+PathGen on CSQA.

***Are ensemble-based coarse explanations effective?*** By default, SALKG-Coarse uses our proposed ensemble-based coarse explanations (Sec. 3.1). Alternatively, we consider using Grad and Occl to create coarse explanations. For Grad, we compute $\phi$ the same way as in Sec. 3.2, except using graph embedding $\mathbf{g}$ instead of node/path embeddings. Since a zero vector would have zero gradient, this is equivalent to comparing $\mathbf{g}$ to a zero vector baseline. For Occl, we compute $\phi$ as the decrease in $p_{KG}$ if $\mathbf{g}$ is replaced

| | CSQA Dev Accuracy (%) | |
|---|---|---|
| Model (BERT) | MHGRN | PathGen |
| SALKG-Coarse | **59.49** ($\pm$0.05) | **60.72** ($\pm$0.58) |
| - w/ Grad | 56.84 ($\pm$2.27) | 56.18 ($\pm$2.31) |
| - w/ Occl | 57.60 ($\pm$0.74) | 56.32 ($\pm$1.66) |
| SALKG-Fine (Occl) | **57.28** ($\pm$0.95) | **59.13** ($\pm$2.35) |
| - w/ Grad | 56.05 ($\pm$1.03) | 58.80 ($\pm$1.08) |
| SALKG-Hybrid (Occl) | 59.92 ($\pm$0.31) | **60.88** ($\pm$0.05) |
| - w/ Grad | **60.17** ($\pm$0.21) | 59.71 ($\pm$0.08) |
| SALKG-Fine (Occl) | **57.28** ($\pm$0.95) | **59.13** ($\pm$2.35) |
| - w/ Random Prune | 50.61 ($\pm$0.68) | 54.10 ($\pm$2.13) |
| - w/ Heuristic Prune | 50.72 ($\pm$0.46) | 50.53 ($\pm$0.74) |
| SALKG-Fine (Occl) | **57.28** ($\pm$0.95) | **59.13** ($\pm$2.35) |
| - w/ BCE Sal. Loss | 50.83 ($\pm$1.75) | 55.15 ($\pm$2.58) |

Table 8: **Ablation Studies.** Best model in **bold**.

with a zero vector. For both Grad and Occl, we set $s_{\mathrm{c}} = \phi$. In Table 8, we see that our default SALKG-Coarse significantly outperforms SALKG-Coarse with both Grad and Occl. In Sec. A.2, we further discuss why Grad and Occl are ill-suited for creating coarse explanations.

***For SALKG, is Occl better than Grad?*** In Tables 5-6, we report SALKG-Fine and SALKG-Hybrid performance with Occl fine explanations. In Table 8, we compare Occl and Grad on SALKG-Fine and SALKG-Hybrid. Overall, Occl slightly outperforms Grad, although Grad beats Occl on MHGRN

for SALKG-Hybrid. Their relative performance could also depend on the choice of top-$k$%, which we plan to explore later. In Sec. A.9, we further compare Occl and Grad on other settings.

***How does* SALKG-Fine*'s soft KG pruning compare to hard KG pruning?*** SALKG-Fine does soft pruning of unhelpful fine units via soft attention. We compare SALKG-Fine to two baselines where the KG is filtered via hard pruning, which cannot be easily incorporated into end-to-end training. For RANDOM Prune and HEURISTIC Prune, we respectively create RANDOM and HEURISTIC explanations, then hard prune all negative units from the KG. The KG-augmented model then uses the pruned KG as its KG input. In Table 8, we see that SALKG-Fine significantly outperforms the two baselines, showing the benefits of jointly training the model on saliency and QA prediction.

***Is it effective to train* SALKG-Fine *with KL divergence?*** We train SALKG-Fine's explanation predictor (*i.e.*, attention mechanism) using KL divergence as the saliency loss. Thus, within a KG, the distribution over attention weights constitutes a single prediction. Alternatively, we could treat each attention weight as a separate prediction and train the attention mechanism using binary cross entropy (BCE) loss. In Table 8, we find that using KL divergence yields much higher performance than using BCE loss. This suggests that the attention weights should not be trained separately, as each attention weight is highly dependent on other attention weights in the same KG.

### 6.4 Case Studies

We visualize coarse/fine explanations created from BERT+PathGen on CSQA, with 1-hop or 2-hop paths as fine units. For coarse explanations, we show examples of positive (*i.e.*, useful) and negative KGs. Since KGs are too large to show here, we uniformly sample three paths per KG. For the positive KG example, the question is *James loved to play violin. He did it in his spare time because he found it what?*, the answer choice is *relaxing*, and the target answer is *relaxing*. Its paths are: **(1)** `play` –[is related to]–> `x` <–[is used for]– `relaxing`, **(2)** `violin` –[is used for]–> `x` –[is used for]–> `relaxing`, and **(3)** `time` <–[has subevent]– `x` –[has subevent]–> `relax`. For the negative KG example, the question is *Where do soldiers not deployed eat their food?*, the answer choice is *neighbor's house*, and the target answer is *military base*. Its paths are: **(1)** `soldier` <–[is related to]– `x` <–[is related to]– `house`, **(2)** `eat` –[is related to]–> `x` –[is at location of]–> `house`, and **(3)** `food` <–[is related to]– `x` –[is at location of]–> `house`. For fine explanations, we show examples of positive and negative paths from the same KG. Here, the question is *Where can you find a bar before traveling a long distance?*, the answer choice is *airport*, and the target answer is *airport*. The positive path is: `bar` –[is at location]–> `airport`. The negative path is: `travel` <–[is used for]– `x` –[is at location]– `airport`. We can roughly see that the positive KGs/paths are useful for predicting the correct answer, and vice versa. However, as shown in [45], the model's judgment of KG/path usefulness may not always align with human judgment. See Sec. A.16 for more illustrative examples of coarse/fine explanations.

## 7 Related Work

**Creating Model Explanations** Many methods aim to explain PLMs' predictions by highlighting important tokens in the model's text input. Such methods are usually gradient-based [51, 29, 10], attention-based [40, 53, 14, 25], or occlusion-based [12, 42, 22, 30]. Similarly, for graph encoders, a number of works use post-hoc optimization to identify important nodes [19, 62] or subgraphs [62] in the graph input. Meanwhile, KG-augmented models' attention weights can be used to explain which parts of the KG are important [31, 13, 34, 56, 60]. These KG explanations can be interpreted as identifying knowledge in the KG that is complementary to the knowledge encoded in the PLM.

**Learning From Model Explanations** Besides manual inspection, explanations can be used in various ways, like extra supervision or regularization [43, 17, 41, 1], pruned inputs [21, 3, 28], additional inputs [16, 8], and intermediate variables [58, 66, 44]. The most similar work to ours is [43], which proposed training a student model to mimic a teacher model's predictions by regularizing the student model's attention via text explanations created from the teacher model. However, [43] aims to evaluate explanations, while our goal is to improve performance via explanations. To the best of our knowledge, SALKG is the first to supervise KG-augmented models with KG explanations.

See Sec. A.20 for a more comprehensive overview of the related literature.

## 8 Conclusion

In this paper, we proposed creating coarse and fine explanations for KG-augmented models, then using these explanations as extra inputs (ORACLE) or supervision (SALKG). Across three commonsense QA

benchmarks, SALKG achieves strong performance, especially when both coarse and fine explanations are used. In future work, we plan to explore incorporating active learning into SALKG, so that models can also leverage explanation-based feedback from humans about KG saliency.

# 9 Acknowledgments

This research is supported in part by the Office of the Director of National Intelligence (ODNI), Intelligence Advanced Research Projects Activity (IARPA), via Contract No. 2019-19051600007, the DARPA MCS program under Contract No. N660011924033, the Defense Advanced Research Projects Agency with award W911NF-19-20271, NSF IIS 2048211, NSF SMA 1829268, and gift awards from Google, Amazon, JP Morgan, and Sony. We would like to thank all of our collaborators at the USC INK Research Lab for their constructive feedback on this work.

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
