# A Appendix

## A.1 Construction of the Contextualized KG

In Sec. 2, we defined the full KG as $\tilde{\mathcal{G}} = (\tilde{\mathcal{V}}, \tilde{\mathcal{R}}, \tilde{\mathcal{E}})$, where $\tilde{\mathcal{V}}$, $\tilde{\mathcal{R}}$, and $\tilde{\mathcal{E}}$ are all of the KG's nodes (concepts), relations, and edges (facts), respectively. For each instance, we assume access to $\tilde{\mathcal{G}}$ but do not use the entire KG in practice. Given a question $q$ and an answer choice $a_i$ for some instance, we construct the contextualized KG, $\tilde{\mathcal{G}}_i = (\mathcal{V}_i, \mathcal{R}_i, \mathcal{E}_i)$ by heuristically extracting edges from $\tilde{\mathcal{G}}$, following the approach taken by most prior KG-augmented model works [13, 56, 31].

$\tilde{\mathcal{G}}_i = (\mathcal{V}_i, \mathcal{R}_i, \mathcal{E}_i)$ is built differently for node-based models and path-based models, and we describe both types of contextualized KG construction procedures below. Note that these procedures are not designed by us, but simply follow what was proposed and shown to work well in the KG-augmented models' original papers [13, 56]. Thus, we do not experiment with different contextualized KG construction procedures, since it is out of the scope of our work.

Let us define the KG nodes mentioned in $q$ and $a_i$ as QA nodes. For example, for the question *What would you put in a teakettle?* and answer choice *water*, the QA nodes would be `put`, `teakettle`, and `water`. We ground raw mentions of QA nodes to the KG via spaCy-based lemmatization and stop-word filtering [18].

For node-based models (MHGRN [13]), we select $\mathcal{V}_i \subseteq \tilde{\mathcal{V}}$ as the QA nodes and all nodes in the QA nodes' 1-hop KG neighborhood. Next, we choose $\mathcal{R}_i \subseteq \tilde{\mathcal{R}}$ as all of the relations between concepts in $\mathcal{V}_i$. Finally, we take $\mathcal{E}_i \subseteq \tilde{\mathcal{E}}$ as all of the edges involving $\mathcal{V}_i$ and $\mathcal{R}_i$.

For path-based models (PathGen [56], RN [13, 4]), we select $\tilde{\mathcal{G}}_i$ as all 2-hop paths between all question-answer node pairs. Thus, $\mathcal{V}_i \subseteq \tilde{\mathcal{V}}$ consists of the QA nodes as well as all intermediate nodes in the 2-hop paths. Meanwhile, $\mathcal{R}_i \subseteq \tilde{\mathcal{R}}$ and $\mathcal{E}_i \subseteq \tilde{\mathcal{E}}$ consist of all relations and edges within the 2-hop paths. When reasoning over the 2-hop paths, the model does not actually use the intermediate nodes, perhaps in order to keep the path more general [13, 56].

## A.2 Alternative Formulation of Coarse Saliency Explanations

SALKG-Coarse uses coarse explanations, which state whether $\mathcal{G}$ or `None` (*i.e.*, no $\mathcal{G}$) should be used for the given task instance. By default, SALKG-Coarse uses our proposed ensemble-based coarse explanations (Sec. 3.1). In this case, the coarse explanations decide between $\mathcal{G}$ and `None` at the *prediction* level. That is, the coarse explanations correspond to saliency weights which perform attention over $\mathcal{F}_{\text{KG}}$'s and $\mathcal{F}_{\text{No-KG}}$'s predictions.

**Graph Embedding Based Explanations** In Sec. 6.3, we also considered applying coarse explanations at the graph embedding level. In this case, using $\mathcal{G}$ corresponds to using graph embedding **g**, while using `None` corresponds to using some baseline embedding **b** that does not contain any information from $\mathcal{G}$. **b** could be a zero vector, random vector, *etc.* Our experiments in Sec. 6.3 — with **b** as a zero vector and Grad/Occl as saliency methods — show that this approach does not yield good empirical results. We believe the issue is that **b** does not contain any `None`-specific information. Recall that the ensemble-based SALKG's prediction is a weighted sum of $\mathcal{F}_{\text{KG}}$'s and $\mathcal{F}_{\text{No-KG}}$'s predictions, which means we interpolate between $\mathcal{F}_{\text{KG}}$'s and $\mathcal{F}_{\text{No-KG}}$'s predictions. Here, $\mathcal{F}_{\text{No-KG}}$'s prediction actually contains meaningful information about $\mathcal{F}_{\text{No-KG}}$. On the other hand, it does not make sense to interpolate between **g** and **b**, since **b** does not have any meaningful information. We also considered learning **b** when training the KG model, but this would require a complicated multitask learning setup where the KG and No-KG models are jointly trained using **g** and **b**, respectively.

## A.3 Implementation Details for Grad-Based Fine Saliency Explanations

In Sec. 3.2, we discussed the *gradient×input* (Grad) [10] method for computing raw fine saliency scores $\phi$. For multi-choice QA, assume we are given text statement $x_i = q \oplus a_i$ (formed from question $q$ and answer choice $a_i$), KG $\mathcal{G}_i$, unit $u_{ij}$, and $u_{ij}$'s embedding $\mathbf{u}_{ij} \in \mathbb{R}^d$ in $\mathcal{G}_i$. Also, let

$u_{ij}^{(\ell)}$ be the $\ell$-th element of $u_{ij}$. Then, $\phi$ is computed as follows:

$$\phi(u_{ij}; x_i, \mathcal{G}_i) = \begin{cases} \sum_{\ell=1}^{d} \mathbf{u}_{ij}^{(\ell)} \frac{\partial p_{\text{KG}}(x_i, \mathcal{G}_i)}{\partial \mathbf{u}_{ij}^{(\ell)}}, & a_i = a^* \\ -\sum_{\ell=1}^{d} \mathbf{u}_{ij}^{(\ell)} \frac{\partial p_{\text{KG}}(x_i, \mathcal{G}_i)}{\partial \mathbf{u}_{ij}^{(\ell)}}, & a_i \neq a^* \end{cases} \tag{3}$$

Depending on the type of graph encoder used, a unit may or may not be given to the model as a single embedding. While node-based graph encoders take node embeddings as input, path-based graph encoders do not take path embeddings as input. Instead path-based graph encoders take node and relation embeddings as input, then form path embeddings from these node and relation embeddings.

As a result, for Grad, the computation of $\phi$ is slightly different between node-based and path-based graph encoders. For node-based encoders, unit embedding $\mathbf{u}_{ij}$ is just a node embedding. Thus, a node's $\phi$ score is computed directly using Eq. 3. For path-based encoders, given a path, we first use Eq. 3 to compute a separate $\phi$ score for each node embedding and relation embedding in the path. Then, we compute the path's $\phi$ score as the sum of the $\phi$ scores of its constituent nodes and relations.

### A.4 Evaluation Protocol

We present a more detailed description of the evaluation protocol used to obtain the results in Sec. 6. First, define non-explanation models (No-KG, KG, and No-KG + KG) as models that are not regularized with any kind of explanation, and define explanation models (RANDOM, HEURISTIC, SALKG) as models that are regularized with some kind of explanation. Second, each non-explanation model's performance is reported as the average over three seeds, which we denote as the non-explanation seeds. Also, recall that each explanation model is built from No-KG and/or KG models. Third, for each of the three non-explanation seeds, we train the explanation model on three more seeds, which we call the explanation seeds. After that, we compute the explanation model performance by averaging over [three non-explanation seeds] × [three explanation seeds] = [nine total seeds].

We summarize the evaluation protocol below:

- Non-explanation seeds: 1, 2, 3
- Explanation seeds: A, B, C
- Non-explanation performance: *average*(1, 2, 3)
- Explanation performance: *average*(1A, 1B, 1C, 2A, 2B, 2C, 3A, 3B, 3C)

### A.5 Dataset Details

Below are more detailed descriptions of the three datasets used for the experiments in Sec. 6. All datasets and resources used in this paper are publicly available and free for any researcher to use.

**CommonsenseQA (CSQA)** [52] is a multi-choice QA dataset whose questions require commonsense reasoning to solve. Questions and answer choices in CSQA are derived from ConceptNet [49]. The official (OF) data split has 9741/1221/1140 questions for OFtrain/OFdev/OFtest. Since the labels for OFtest are not publicly available, we use the in-house (IH) data split introduced in [31] and used in many subsequent works [13, 56, 60]. The in-house data split has 8500/1221/1241 questions for IHtrain/IHdev/IHtest, where the IHtrain and IHtest are obtained by partitioning OFtrain.

**OpenbookQA (OBQA)** [39] is a multi-choice QA dataset which aims to simulate open-book science exams. OBQA has 4957/500/500 elementary-school-level science questions for train/dev/test, but also provides a supplementary "open book" resource containing 1326 core science facts. To solve questions from OBQA, the model needs to reason over both information from the open book and commonsense knowledge from the KG (*i.e.*, ConceptNet).

**CODAH** [6] is a multi-choice QA dataset which augments the SWAG [63] sentence completion dataset with more difficult, adversarially-created questions. Similar to SWAG, CODAH's questions are designed to require commonsense reasoning to solve. CODAH contains 2801 questions, and its official split specifies five folds, which balance the distribution of question categories per fold. Thus, by default, performance is evaluated by averaging over the five folds. However, due to computational constraints, we only evaluate on the first fold and compare to the baselines presented in Sec. 4.2 and Sec. 6, rather than to previously published methods.

| Top-k% | CSQA Test Accuracy (%) | | OBQA Test Accuracy (%) | |
|---|---|---|---|---|
| | MHGRN | PathGen | MHGRN | PathGen |
| 2 | 72.66 (±1.52) | 69.86 (±1.11) | 66.47 (±1.27) | 61.33 (±2.69) |
| 5 | 72.58 (±0.74) | **71.64** (±3.17) | **69.13** (±0.81) | **64.80** (±1.40) |
| 10 | **73.65** (±0.21) | 71.39 (±1.54) | 65.07 (±1.70) | 51.60 (±1.13) |
| 30 | 71.98 (±0.47) | 69.76 (±0.44) | 63.47 (±1.14) | 61.87 (±4.61) |
| 50 | 72.93 (±0.84) | 71.04 (±0.05) | 63.27 (±3.00) | 63.60 (±1.71) |
| 70 | 72.04 (±1.05) | 70.13 (±0.66) | 65.80 (±1.91) | 64.40 (±0.40) |

Table 9: **SALKG-Fine Performance for Different top-k% Thresholds.** We report performance for RoBERTa+MHGRN and RoBERTa+PathGen on CSQA and OBQA. Best model is shown in **bold**.

## A.6   Threshold Tuning for Creating Explanations

**Tuning $T$ Threshold for Coarse Explanations**   Recall that coarse explanations are binarized via threshold $T$ (Sec. 3.1). To set $T$, we manually tune $T$ to maximize ORACLE-Coarse's dev accuracy. This can be done efficiently, since ORACLE-Coarse does not require any training. We use a sweep of $T = [0.01, 0.02, 0.03, 0.04, 0.05]$ and find that $T = 0.01$ yields best performance overall.

**Tuning top-$k$% Threshold for Fine Explanations**   Recall that fine explanations are binarized via threshold $k$, used to set the top-$k$% of units as positive (Sec. 3.2). To set $k$, we manually tune $k$ to maximize SALKG-Coarse's dev accuracy. Table 9 shows the performance of RoBERTa+MHGRN and RoBERTa+PathGen on CSQA and OBQA, across different values of $k$. Due to computational constraints, we report the average performance across [best non-explanation seed] × [three explanation seeds] = [three total seeds], as opposed to the default [three non-explanation seed] × [three explanation seeds] = [nine total seeds] (Sec. A.4). We use a sweep of $k = [5, 10, 30, 50]$ and find that $k = 5$ yields best performance overall, although there is not a clear trend that smaller $k$ is better. In this paper, we used $k = 10$ for all experiments, so it may be promising to further explore tuning $k$ in the future.

## A.7   Additional Details about ORACLE Models

We provide more details about ORACLE-Coarse and ORACLE-Fine. Given the coarse saliency explanations, ORACLE-Coarse simply involves choosing the "correct" prediction — between $\mathcal{F}_{\text{KG}}$'s and $\mathcal{F}_{\text{No-KG}}$'s predictions — for each answer choice. Given that $\mathcal{F}_{\text{KG}}$'s and $\mathcal{F}_{\text{No-KG}}$'s predictions are simply loaded from disk, this process runs very quickly, since it does not require additional training. On the other hand, ORACLE-Fine involves training the KG-augmented model while applying the fine saliency explanations as a binary mask to the graph encoder's attention weights.

## A.8   Additional SALKG Results on CODAH

In this section, we present additional SALKG results on CODAH. These additional results consist of RoBERTa+RN, BERT+MHGRN, BERT+PathGen, and BERT+RN, all using threshold top-10%. Also, across all settings, we report both Grad and Occl results for SALKG-Fine and SALKG-Hybrid. Due to computational constraints, we report the average performance across [best non-explanation seed] × [three explanation seeds] = [three total seeds], as opposed to the default [three non-explanation seed] × [three explanation seeds] = [nine total seeds] (Sec. A.4). These results are shown in Table 10, along with the RoBERTa+MHGRN and RoBERTa+PathGen results from Table 6.

First, we see that SALKG-Hybrid (either Grad or Occl) performs the best on all settings except RoBERTa+PathGen. For RoBERTa+PathGen, RANDOM-Coarse and RANDOM-Hybrid perform the best, although some SALKG models perform almost as well. RANDOM's strong performance is likely due to us reporting performance for the best non-explanation seed, rather than averaging over three non-explanation seeds. Second, for SALKG-Fine, Occl beats Grad on all settings except RoBERTa+PathGen. Third, for SALKG-Hybrid, Occl beats Grad on BERT+MHGRN, BERT+PathGen, and BERT+RN, while Grad beats Occl on RoBERTa+MHGRN and RoBERTa+PathGen.

| | CODAH Test Accuracy (%) | | | | | |
|---|---|---|---|---|---|---|
| | **MHGRN** | | **PathGen** | | **RN** | |
| **Model** | BERT | RoBERTa | BERT | RoBERTa | BERT | RoBERTa |
| No-KG | 60.96 (±1.27) | 83.96 (±0.79) | 60.96 (±1.27) | 83.96 (±0.79) | 60.96 (±1.27) | 83.96 (±0.79) |
| KG | 58.68 (±1.63) | 84.02 (±1.27) | 58.80 (±2.01) | 84.02 (±1.62) | 55.92 (±1.04) | 82.64 (±0.85) |
| No-KG + KG | 60.60 (±1.30) | 84.08 (±1.46) | 60.42 (±1.14) | 84.69 (±1.48) | 58.62 (±1.53) | 84.08 (±0.55) |
| Random-Coarse | 60.78 (±0.38) | 84.62 (±0.55) | 61.74 (±0.28) | **86.07** (±0.89) | 57.84 (±0.83) | 84.14 (±0.65) |
| Random-Fine | 58.50 (±0.91) | 84.02 (±0.89) | 54.47 (±1.55) | 75.74 (±4.71) | 54.53 (±1.40) | 76.10 (±4.65) |
| Random-Hybrid | 62.16 (±0.00) | 84.80 (±0.10) | 61.74 (±0.55) | 84.68 (±0.18) | 62.40 (±0.10) | 84.14 (±0.65) |
| Heuristic-Coarse | 58.38 (±0.00) | 85.11 (±0.10) | 61.08 (±0.00) | 85.59 (±0.00) | 59.70 (±0.10) | 83.60 (±0.00) |
| Heuristic-Fine | 60.18 (±1.36) | 83.72 (±0.92) | 55.98 (±0.28) | 82.64 (±2.61) | 54.71 (±3.07) | 81.80 (±2.77) |
| Heuristic-Hybrid | 62.16 (±0.00) | 84.80 (±0.10) | 61.98 (±0.31) | 85.23 (±0.00) | 62.28 (±0.10) | 85.35 (±0.10) |
| Salkg-Coarse | 61.02 (±0.10) | 85.41 (±0.18) | 61.20 (±0.28) | 85.95 (±0.18) | 61.74 (±0.21) | 84.98 (±0.42) |
| Salkg-Fine (Occl Top-10%) | 60.00 (±1.26) | 84.08 (±1.14) | 57.72 (±1.09) | 83.36 (±0.81) | 59.16 (±2.15) | 83.78 (±1.41) |
| Salkg-Fine (Grad Top-10%) | 59.16 (±0.38) | 84.20 (±1.17) | 57.36 (±0.75) | 83.00 (±1.51) | 55.86 (±0.79) | 83.66 (±0.89) |
| Salkg-Hybrid (Occl Top-10%) | **62.28** (±0.10) | 85.71 (±0.10) | **62.04** (±0.45) | 84.44 (±0.63) | **62.58** (±0.10) | **85.11** (±0.28) |
| Salkg-Hybrid (Grad Top-10%) | 60.48 (±0.21) | **88.17** (±0.10) | 61.02 (±0.10) | 85.17 (±0.28) | 61.38 (±0.68) | **85.11** (±0.55) |

Table 10: **Salkg Performance on CODAH for Additional Settings.** Building upon the CODAH results in Table 6 (RoBERTa+MHGRN and RoBERTa+PathGen), we additionally report results for RoBERTa+RN, BERT+MHGRN, BERT+PathGen, and BERT+RN, all using threshold top-10%. We also report both Grad and Occl results for Salkg-Fine and Salkg-Hybrid. Best model is shown in **bold**.

## A.9 Additional Salkg Results for Grad *vs.* Occl

In Tables 11-12, we compare Grad *vs.* Occl on CSQA and OBQA, respectively. Due to computational constraints, we report the average test accuracy across [best non-explanation seed] × [three explanation seeds] = [three total seeds], as opposed to the default [three non-explanation seed] × [three explanation seeds] = [nine total seeds] (Sec. A.4). For Salkg-Fine and Salkg-Hybrid on CSQA, we find that Occl beats Grad on all settings, except Salkg-Fine on RoBERTa+RN. However, for Salkg-Fine on OBQA, Grad beats Occl on RoBERTa+PathGen, BERT+RN, and RoBERTa+RN, while Occl beats Grad on BERT+MHGRN, RoBERTa+MHGRN, and BERT+PathGen. Meanwhile, for Salkg-Hybrid on OBQA, Occl beats Grad on all settings except BERT+PathGen. Thus, we see that Occl generally outperforms Grad, although Grad can beat Occl on certain settings.

## A.10 Comparison to Published OBQA Baselines

To further demonstrate that Salkg models perform competitively, we also compare Salkg to the many KG-augmented model baseline results published in [13, 56, 60], for OBQA. The baselines we consider are RN, RN + Link Prediction, RGCN, GconAttn, MHGRN, and PathGen. For the non-Salkg versions of MHGRN, PathGen, and RN, we quote the published results. Since these published results average over four seeds (instead of three), we report Salkg results over four seeds in Table 13. For OBQA, we find that vanilla PathGen (quoted from published results) performs the best, while Salkg-Hybrid (MHGRN) and Salkg-Hybrid (PathGen) are almost as good. These OBQA results indicate that our reproduction of vanilla PathGen may not have been optimally tuned, thus limiting the performance of the Salkg models built upon PathGen. We plan to investigate this issue in future work.

| | CSQA Test Accuracy (%) | | | | | |
|---|---|---|---|---|---|---|
| | **MHGRN** | | **PathGen** | | **RN** | |
| **Model** | BERT | RoBERTa | BERT | RoBERTa | BERT | RoBERTa |
| Salkg-Fine (Grad) | 55.44 (±1.22) | 72.95 (±1.44) | 57.10 (±0.81) | 70.10 (±0.28) | 56.14 (±1.97) | **72.12** (±0.14) |
| Salkg-Fine (Occl) | **56.78** (±2.14) | **73.65** (±0.21) | **57.64** (±2.12) | **71.39** (±1.54) | **56.86** (±0.41) | 71.58 (±1.10) |
| Salkg-Hybrid (Grad) | 59.07 (±0.56) | 72.79 (±0.20) | 57.53 (±0.43) | 71.39 (±0.14) | 57.29 (±0.29) | 71.98 (± 0.28) |
| Salkg-Hybrid (Occl) | **59.12** (±0.28) | **73.41** (±0.16) | **60.35** (±0.32) | **73.11** (±1.00) | **58.80** (±0.19) | **74.64** (±0.09) |

Table 11: **CSQA Performance Comparison for Salkg Grad *vs.* Occl Models.** Best model between Grad and Occl is shown in **bold**.

| | OBQA Test Accuracy (%) | | | | | |
| | MHGRN | | PathGen | | RN | |
| Model | BERT | RoBERTa | BERT | RoBERTa | BERT | RoBERTa |
|---|---|---|---|---|---|---|
| SALKG-Fine (Grad) | 53.40 (±0.69) | 58.80 (±8.66) | 55.33 (±0.31) | **67.87** (±1.81) | **56.53** (±0.31) | **68.87** (±1.67) |
| SALKG-Fine (Occl) | **53.93** (±1.01) | **65.07** (±1.70) | **55.40** (±0.53) | 51.60 (±1.13) | 55.67 (±0.90) | 62.33 (±0.90) |
| SALKG-Hybrid (Grad) | 53.80 (±0.20) | 69.47 (±0.31) | **55.67** (±0.64) | 69.93 (±0.61) | 53.20 (±0.72) | 69.40 (±0.20) |
| SALKG-Hybrid (Occl) | **56.20** (±0.20) | **70.73** (±0.12) | 55.33 (±0.23) | **70.07** (±0.12) | **53.93** (±0.42) | **70.80** (±0.00) |

Table 12: **OBQA Performance Comparison for SALKG Grad *vs.* Occl Models.** Best model between Grad and Occl is shown in **bold**.

| Model (RoBERTa) | OBQA Test Accuracy (%) |
|---|---|
| RN [46] | 65.20 (±1.18) |
| RN + Link Prediction [56] | 66.30 (±0.48) |
| RGCN [47] | 62.45 (±1.57) |
| GconAttn [57] | 64.75 (±1.48) |
| MHGRN [13] | 66.85 (±1.19) |
| PathGen [56] | **71.20** (±0.96) |
| SALKG-Coarse (MHGRN) | 69.85 (±0.30) |
| SALKG-Fine (MHGRN) | 64.65 (±1.62) |
| SALKG-Hybrid (MHGRN) | 70.75 (±0.10) |
| SALKG-Coarse (PathGen) | 69.70 (±0.93) |
| SALKG-Fine (PathGen) | 54.30 (±5.84) |
| SALKG-Hybrid (PathGen) | 70.00 (±0.16) |

Table 13: **Comparison of SALKG to Published OBQA Results.** Best model is shown in **bold**.

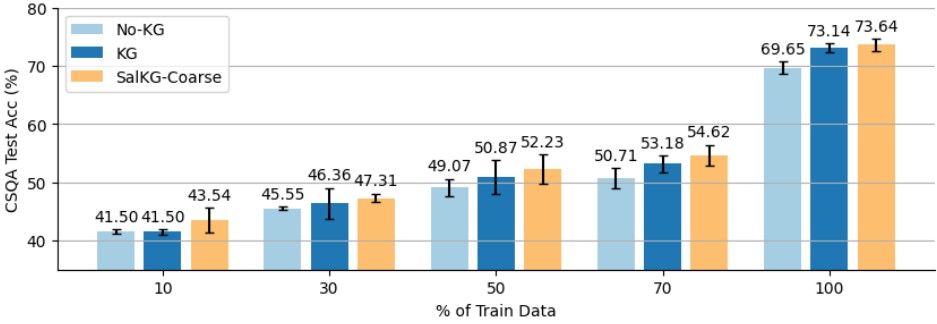

Figure 4: **Low-Resource Learning.** CSQA test accuracy for No-KG, KG, and SALKG-Coarse, when using varying amounts of training data.

## A.11  Low-Resource Learning

In Fig. 4, we show CSQA performance for different models in low-resource settings. Specifically, we experiment with low-resource learning by training the model on 10%, 30%, 50%, or 70% of the training data. For reference, we also include CSQA performance when using 100% of the training data. Here, we consider No-KG (RoBERTa), KG (MHGRN), and SALKG-Coarse (RoBERTa+MHGRN). Across all settings, we find that SALKG-Coarse outperforms both No-KG and KG, suggesting that regularizing the model with coarse explanations can provide a helpful inductive bias for generalizing from limited training data.

## A.12  Analyzing the Impact of Coarse Explanations

SALKG-Coarse is based on the insight that KG information may help the model on some instances but hurt on others. Thus, even if KG outperforms No-KG on average, No-KG may still correctly predict some instances that KG got wrong. SALKG-Coarse takes advantage of such complementary predictions between No-KG and KG, in order to achieve performance higher than $\max(\text{No-KG}, \text{KG})$. As shown by RoBERTa+PathGen and RoBERTa+RN on OBQA (Table 6), SALKG-Coarse can still beat $\max(\text{No-KG}, \text{KG}, \text{No-KG} + \text{KG})$ even when No-KG outperforms KG.

| Question Set | Question Percentage (%) |
|---|---|
| No-KG Correct | 55.44 |
| KG Correct | 56.65 |
| Only No-KG Correct | 9.43 |
| Only KG Correct | 10.64 |
| Both Correct | 46.01 |
| Both Incorrect | 33.92 |
| At Least One Incorrect | 66.08 |
| SALKG-Coarse Correct | 56.65 |
| ORACLE-Coarse Correct | 68.57 |

Table 14: **Impact of Coarse Explanations.** Using BERT+PathGen on CSQA, we present a performance breakdown for various question sets, in order to analyze why SALKG-Coarse is able to beat No-KG and KG.

In Table 14, we analyze the performance of BERT (*i.e.*, No-KG), PathGen (*i.e.*, KG), SALKG-Coarse (BERT+PathGen), and ORACLE-Coarse (BERT+PathGen) on various sets of questions in CSQA. Due to computational constraints, each model's performance here is reported for one seed (instead of using the protocol described in Sec. A.4), so these results are not directly comparable to those in Table 5. Through this performance breakdown, we can isolate the potential improvement contributed by each base model to SALKG-Coarse. We begin by looking at the questions for which SALKG-Coarse has no influence. These are the 46.01% of questions correctly answered by both models and the 33.92% of questions incorrectly answered by both models. Since SALKG-Coarse is trained to choose between the two models' predictions, SALKG-Coarse's output is fixed if both models make the same prediction. This leaves 20.07% of questions that were correctly answered by exactly one of the two models: 9.43% were from No-KG, while the other 10.64% were from KG. This 20.07% of constitutes the complementary predictions leveraged by SALKG-Coarse.

Based on this question-level analysis, we would estimate the ORACLE-Coarse accuracy to be 66.08%, the percentage of questions that at least one model answered correctly. However, as stated in Sec. 3.1, coarse saliency targets are created at the answer choice level (not question level), which offers us more flexibility to choose between No-KG and KG. As a result, ORACLE-Coarse's accuracy is actually 68.57%. This leaves SALKG-Coarse (56.65%) significant room for improvement, perhaps through better model architecture and training.

### A.13 Comparing Salient and Non-Salient KG Units

This paper explores learning from explanations of KG units' saliency (*i.e.*, usefulness). Overall, our focus is on how using salient KG units can yield improve model performance. In this subsection, we also analyze whether salient and non-salient KG units, as determined by our coarse/fine explanation methods, can differ in other ways that are not directly related to performance (Table 15). For both coarse and fine explanations, we use the BERT+MHGRN model on CSQA, where MHGRN is a node-based graph encoder (Sec. 4.2). Recall that Q nodes and A nodes are nodes (*i.e.*, concepts) mentioned in the given question and answer choice, respectively (Sec. 6.1).

For coarse explanations, we use the ensemble-based explanations introduced in Sec. 3.1. We compare salient and non-salient KGs with respect to the number of nodes in the KG (# nodes), percentage of Q nodes in the KG (% Q nodes), percentage of A nodes in the KG (% A nodes), clustering coefficient (cluster coeff.), and average node degree (degree). These results are shown in Table 15a. We see that these metrics are not very discriminative, as salient and non-salient KGs perform similarly on all of these metrics.

For fine explanations, we use the Grad-based explanations described in Sec. 3.2 and Sec. A.3. We compare salient and non-salient nodes with respect to the percentage of Q nodes among salient/non-salient nodes in the KG (% Q nodes), percentage of A nodes among salient/non-salient nodes in the KG (% A nodes), and node degree (degree). These results are shown in Table 15b. Here, we see that %Q nodes and %A nodes are actually quite discriminative metrics between salient and non-salient nodes. On average, the percentage of Q nodes among salient nodes (16.84%) is 56.07% greater than the percentage of Q nodes among non-salient nodes (10.79%). Similarly, on average, the percentage of A nodes among salient nodes (10.00%) is 65.02% greater than the percentage of Q nodes among non-salient nodes (6.06%). However, compared to %Q nodes and %A nodes, degree is not as discriminative. This indicates that the difference between salient and non-salient nodes may be more semantic than structural.

| Metric | Salient | Non-Salient |
|---|---|---|
| # nodes | 125.88 | 120.57 |
| % Q nodes | 9.09 | 9.17 |
| % A nodes | 2.94 | 3.12 |
| cluster coeff. | 4.26E-1 | 4.25E-1 |
| degree | 9.89 | 9.78 |

(a) Salient *vs.* Non-Salient KGs.

| Metric | Salient | Non-Salient |
|---|---|---|
| % Q nodes | 16.84 | 10.79 |
| % A nodes | 10.00 | 6.06 |
| degree | 15.41 | 13.11 |

(b) Salient *vs.* Non-Salient Nodes.

Table 15: **Salient *vs.* Non-Salient KG Units.** Using BERT+MHGRN on CSQA, we compare salient and non-salient KG units. In (a), we compare salient and non-salient KGs, as determined by coarse explanations. In (b), we compare salient and non-salient nodes, as determined by fine explanations.

| | CSQA Test Accuracy (%) | | | | | |
|---|---|---|---|---|---|---|
| | **MHGRN** | | **PathGen** | | **RN** | |
| Model | BERT | RoBERTa | BERT | RoBERTa | BERT | RoBERTa |
| KG (Relation) | 52.89 ($\pm$0.73) | 67.41 ($\pm$0.84) | 52.35 ($\pm$0.60) | 70.08 ($\pm$0.38) | 54.15 ($\pm$0.40) | 68.95 ($\pm$1.58) |
| SALKG-Coarse (Relation) | **55.86** ($\pm$0.48) | **72.53** ($\pm$0.50) | **56.07** ($\pm$0.44) | **71.55** ($\pm$0.85) | **56.93** ($\pm$0.51) | **72.43** ($\pm$0.96) |
| SALKG-Fine (Relation) | 52.58 ($\pm$0.70) | 68.84 ($\pm$0.67) | 53.32 ($\pm$0.61) | 71.23 ($\pm$1.21) | 53.94 ($\pm$0.63) | 69.80 ($\pm$0.64) |
| SALKG-Hybrid (Relation) | 51.28 ($\pm$0.70) | 69.84 ($\pm$0.57) | 53.33 ($\pm$0.55) | 70.34 ($\pm$1.03) | 52.41 ($\pm$1.11) | 68.77 ($\pm$0.80) |
| KG (Node) | 53.63 ($\pm$0.70) | 67.35 ($\pm$0.41) | **55.60** ($\pm$0.16) | 70.51 ($\pm$1.69) | 54.15 ($\pm$2.27) | 70.48 ($\pm$1.71) |
| SALKG-Coarse (Node) | **55.75** ($\pm$0.60) | **71.83** ($\pm$0.60) | 55.43 ($\pm$0.55) | **71.36** ($\pm$0.81) | **56.14** ($\pm$0.73) | **71.20** ($\pm$0.72) |
| SALKG-Fine (Node) | 53.60 ($\pm$0.83) | 66.81 ($\pm$1.09) | 53.13 ($\pm$0.99) | 70.80 ($\pm$1.55) | 54.02 ($\pm$0.84) | 71.08 ($\pm$1.02) |
| SALKG-Hybrid (Node) | 51.14 ($\pm$1.03) | 69.58 ($\pm$0.77) | 50.80 ($\pm$0.83) | 69.85 ($\pm$0.72) | 53.24 ($\pm$0.72) | 69.57 ($\pm$1.14) |
| KG | 57.48 ($\pm$0.89) | 73.14 ($\pm$0.78) | 56.54 ($\pm$0.73) | 72.58 ($\pm$0.57) | 56.46 ($\pm$1.22) | 71.37 ($\pm$1.20) |
| SALKG-Coarse | 57.98 ($\pm$0.90) | **73.64** ($\pm$1.05) | 57.75 ($\pm$0.77) | **73.07** ($\pm$0.25) | 57.50 ($\pm$1.25) | 73.11 ($\pm$1.13) |
| SALKG-Fine | 54.36 ($\pm$2.34) | 70.00 ($\pm$0.81) | 54.39 ($\pm$2.03) | 72.12 ($\pm$0.91) | 54.30 ($\pm$1.41) | 71.64 ($\pm$1.51) |
| SALKG-Hybrid | **58.70** ($\pm$0.65) | 73.37 ($\pm$0.12) | **59.87** ($\pm$0.42) | 72.67 ($\pm$0.65) | **58.78** ($\pm$0.14) | **74.13** ($\pm$0.71) |

Table 16: **SALKG Performance Comparison on CSQA with Perturbed KGs.** Best performance in **bold**.

## A.14 Robustness to KG Perturbation

Table 16 shows the CSQA performance of KG and SALKG models subjected to different forms of KG perturbation. Relation perturbation (Relation) permutes the relation labels of all edges in the KG, while node perturbation (Node) permutes the node labels of all nodes in the KG. These perturbation methods are designed to alter the semantics of the KG. For relation perturbation and node perturbation, SALKG-Coarse (Node) performs best on almost all settings, with KG (Node) barely beating SALKG-Coarse for node perturbation on BERT+PathGen. However, with KG perturbation, SALKG-Hybrid does not perform as well, sometimes even worse than KG and SALKG-Fine. This may be because SALKG-Hybrid relies most heavily on fine explanations, making it especially sensitive to KG perturbation.

We also compare these KG-perturbed models to models without any KG perturbation. As expected, across all settings, the KG-perturbed models outperform the non-KG-perturbed models. Interestingly, we find that SALKG-Coarse is most robust to KG perturbation. For BERT+RN and RoBERTa+RN, SALKG-Coarse (Relation) is less than 1% worse than SALKG-Coarse. This makes sense, since SALKG-Coarse relies least on the KG. For a given instance, SALKG-Coarse has the option to completely ignore KG information when making its prediction. When the KG is perturbed, it would be advantageous for SALKG-Coarse to focus only on the text input.

## A.15 Statistical Significance of Main Results

In this section, we verify the statistical significance of our results in Sec. 6.2. For each setting in Tables 5-6 (except RoBERTa+PathGen on CODAH), we perform the two-sided unpaired T-test with

| | CSQA p-values | | | | | |
|---|---|---|---|---|---|---|
| | **MHGRN** | | **PathGen** | | **RN** | |
| Model | BERT | RoBERTa | BERT | RoBERTa | BERT | RoBERTa |
| Best SALKG Model *vs.* Best Non-SALKG Model | 0.1235 | 0.4238 | 0.0701 | 0.2690 | 0.1336 | 0.0441 |

Table 17: **SALKG T-Test Results on CSQA.** For each setting in Table 5, we perform the T-test between the best SALKG model and the best non-SALKG model.

| Model (RoBERTa) | OBQA p-values | | | CODAH p-values | |
| --- | --- | --- | --- | --- | --- |
| | MHGRN | PathGen | RN | MHGRN | PathGen |
| Best SALKG Model *vs.* Best Non-SALKG Model | 0.2909 | 0.8890 | 0.0005 | 0.1223 | 0.2823 |

Table 18: **SALKG T-Test Results on OBQA and CODAH.** For each setting in Table 6, we perform the T-test between the best SALKG model and the best non-SALKG model.

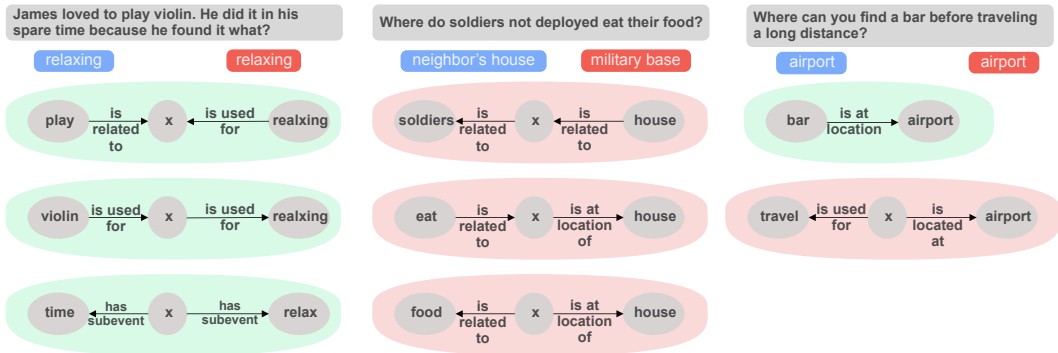

Figure 5: **Examples of coarse/fine saliency explanations.** Illustration of examples presented in Sec. 6.4. Blue denotes given answer choice, while red denotes target answer.

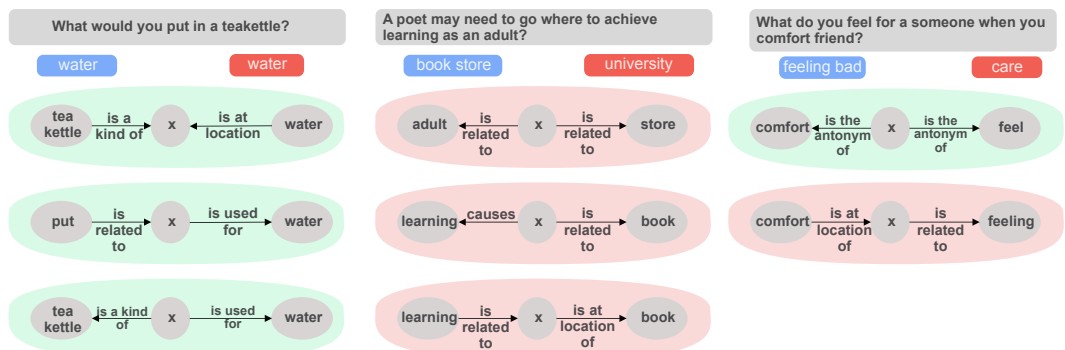

Figure 6: **More examples of coarse/fine saliency explanations.** Illustration of examples presented in Sec. A.16. Blue denotes given answer choice, while red denotes target answer.

unequal variance between the best SALKG model and the best non-SALKG model. The $p$-values are shown in Tables 17-18.

If we use threshold $\alpha = 0.1$ (*i.e.*, $p < 0.1$), then we find that SalKG yields statistically significant improvements on CSQA BERT+PathGen, CSQA RN+RoBERTa, and OBQA RN+RoBERTa. If we use threshold $\alpha = 0.05$ (*i.e.*, $p < 0.05$), then we find that SalKG yields statistically significant improvements on CSQA RN+RoBERTa and OBQA RN+RoBERTa. In particular, the improvement on OBQA RN+RoBERTa is very statistically significant, with $p = 0.0005$. Our T-test results show that SalKG can produce significant performance gains on a number of model-dataset settings, while yielding competitive performance in other settings.

## A.16 Case Studies: Qualitative Analysis of KG Saliency Explanations

In this section, we build upon Sec. 6.4 and illustrate more examples of coarse/fine explanations created from BERT+PathGen on CSQA, with 1-hop or 2-hop paths as fine units. Notice that 2-hop paths consist of two nodes and two relations, with the intermediate node replaced with a placeholder node x, following [13]. By constructing 2-hop paths this way, the model is able to learn from more general 2-hop paths.

First, for coarse explanations, we provide more examples of positive (*i.e.*, useful) and negative KGs.

- For the positive KG example, the question is *What would you put in a teakettle?*, the answer choice is *water*, and the target answer is *water*. Its paths are: **(1)** `teakettle` –[is a kind of]–> x <–[is at location]– `water` , **(2)** `put` –[is related to]–> x –[is used for]–> `water` , and **(3)** `teakettle` –[is a kind of]–> x –[is used for]–> `water` .

- For the negative KG example, the question is *A poet may need to go where to achieve learning as an adult?*, the answer choice is *book store*, and the target answer is *university*. Its paths are: **(1)** `adult` <–[is related to]– x –[is related to]–> `store` , **(2)** `learning` <–[causes]– x <–[is related to]– `book` , and **(3)** `learning` –[is related to]–> x –[is at location of]–> `book` .

Second, we provide more examples of fine explanations. Here, the question is *What do you feel for a someone when you comfort friend?*, the answer choice is *feeling bad*, and target answer is *care*. The positive path is: `comfort` <–[is the antonym of]– x –[is the antonym of]–> `feel` . The negative path is: `comfort` –[is at location of]–> x –[is related to]–> `feeling` .

The examples from Sec. 6.4 are shown in Fig. 5. The examples introduced in this subsection (Sec. A.16) are shown in Fig. 6. Again, in the coarse/fine explanations, we can roughly see that the positive KGs/paths tend to be useful for predicting the correct answer, and vice versa. However, note that the model's judgment of KG/path usefulness may not necessarily align with human judgment [45].

## A.17    User Studies: Quantitative Analysis of KG Saliency Explanations

To better understand the role and limitations of KG saliency explanations, we quantitatively analyze KG saliency explanations in the context of two user studies. In both user studies, the goal is to measure KG saliency explanations' plausibility, *i.e.*, how closely the explanations align with human judgment.

Note that explanation plausibility is orthogonal to our paper's main claims, since we argue that KG saliency explanations can be used as additional supervision for improving performance, not that the explanations are plausible. Nonetheless, these user studies may still provide some useful insights about KG saliency explanations.

### A.17.1    User Study 1: Coarse Saliency Explanations

The first user study measures how well the coarse (graph-level) explanations align with human judgment of usefulness. Given a RoBERTa+PathGen model, we begin by uniformly sampling 25 high-saliency (positive) KGs and 25 low-saliency (negative) KGs from the CSQA training set. Recall that whether a KG is high-saliency or low-saliency was determined by coarse explanations (Sec. 3.1) generated with respect to the given model.

Note that each KG corresponds to one answer choice of a question, so each question in CSQA has up to five corresponding KGs. To ensure that none of the KGs in our sample come from the same question, we ended up pruning two high-saliency and two low-saliency KGs, yielding a final sample of 23 high-saliency and 23 low-saliency KGs.

| Graph Type | Usefulness Score |
|---|---|
| High-Saliency Graph | 0.929 ± 0.734 |
| Low-Saliency Graph | 0.935 ± 0.764 |

Table 19: **Human Evaluation of Coarse Saliency Explanations.** Human-annotated usefulness scores for high- (positive) and low- (negative) saliency graphs.

Since a KG can contain hundreds of paths, it is not feasible to ask humans to evaluate the entire KG's usefulness. Thus, as a very rough representation of the KG, we uniformly sampled three paths from the KG. Then, for each KG, we asked ten human annotators to score each of the three paths' usefulness for predicting the same answer choice predicted by the RoBERTa+PathGen model. To score the paths, all annotators were also given the question, correct answer, and model's predicted answer. The paths were scored on the following 0-2 scale:

- **0** = definitely not useful (*i.e.*, this path is either irrelevant or would cause someone to NOT select the model's predicted answer)

- **1** = possibly useful (*i.e.*, this path provides some support for selecting the model's predicted answer)

- **2** = definitely useful (*i.e.*, this path provides strong support for selecting the model's predicted answer)

| Path Type | Usefulness Score (All Preds) | Usefulness Score (Correct Preds) | Usefulness Score (Incorrect Preds) |
|---|---|---|---|
| High-Saliency Path | 1.091 ± 0.805 | 1.298 ± 0.782 | 0.884 ± 0.776 |
| Med-Saliency Path | 1.222 ± 0.769 | 1.320 ± 0.729 | 1.124 ± 0.798 |
| Low-Saliency Path | 1.060 ± 0.733 | 1.182 ± 0.730 | 0.938 ± 0.717 |

Table 20: **Human Evaluation of Fine Saliency Explanations.** Human-annotated usefulness scores for high-, median-, and low-saliency paths. We display the usefulness scores for paths from all predictions, correct predictions, and incorrect predictions.

Finally, each KG's score is computed as the mean of its three constituent path scores. Below, we show the mean and standard deviation scores for high-saliency and low-saliency graphs. We find that the two graph types have similar mean usefulness scores, while also having relatively large standard deviations. This suggests that coarse saliency explanations do not align strongly with human judgment. One key limitation of this study is that the three sampled paths may not be representative of the entire KG. In the future, we plan to redesign the user study to provide annotators a more comprehensive representation of the KG to evaluate.

### A.17.2 User Study 2: Fine Saliency Explanations

The second user study measures how well the fine (path-level) explanations align with human judgment of usefulness. Given a RoBERTa+PathGen model trained on CSQA, we begin by uniformly sampling 25 correctly answered questions and 25 incorrectly answered questions from the CSQA training set. For each question, we take the model's predicted answer choice and the KG corresponding to the predicted answer choice, then select: **(1)** the path with the highest fine saliency score, **(2)** the path with median fine saliency score, and **(3)** the path with the lowest saliency score. To get finer-grained saliency signal in this study, we consider the raw fine saliency scores, instead of the binarized fine explanations actually used to regularize the model. Recall that a path's fine saliency score (Sec. 3.2) is calculated with respect to the given model.

Next, we asked ten human annotators to score each path's usefulness for predicting the same answer choice predicted by the RoBERTa+PathGen model. Like before, to score the paths, all annotators were also given the question, correct answer, and model's predicted answer. Again, the paths were scored on the following 0-2 scale:

- **0** = definitely not useful (*i.e.*, this path is either irrelevant or would cause someone to NOT select the model's predicted answer)

- **1** = possibly useful (*i.e.*, this path provides some support for selecting the model's predicted answer)

- **2** = definitely useful (*i.e.*, this path provides strong support for selecting the model's predicted answer)

Below, we show the mean scores for high-saliency, median-saliency, and low-saliency paths. We display these scores for paths from all predictions, correct predictions, and incorrect predictions. Overall, we find that the three path types have similar mean usefulness scores, although the mean score for median-saliency paths is somewhat higher than the other two path types'. Still, the standard deviations for all scores are relatively large, so this trend may not be meaningful. These results suggest that fine saliency explanations do not strongly align with human judgment. Additionally, we find that the path usefulness scores for correct predictions tend to be higher than those from incorrect predictions. This makes sense, since, intuitively, a model is more likely to predict the correct answer if it is using more useful knowledge as context.

### A.17.3 Inter-Annotator Agreement

Here, we measure inter-annotator agreement for both user studies, using Fleiss' kappa. For the user study of coarse explanations, the kappa score is 0.2089, which is on the borderline of slight agreement and fair agreement. For the user study of fine explanations, the kappa score is 0.1296, which indicates slight agreement.

These low kappa scores show that even humans can hardly agree on whether the coarse/fine explanations are useful. Therefore, it may not always be beneficial to measure explanation quality in terms of

alignment with human judgment. Moreover, this shows that weak alignment with human judgment does not necessarily imply poor explanation quality.

### A.17.4 Analysis

In our user studies, we did not find strong evidence that coarse/fine saliency explanations align well with human judgment. However, we also found that human annotators had very low agreement about the usefulness of the explanations, which suggests that alignment with human judgment may not be the best measure of explanation quality.

| User Study | Fleiss' Kappa |
|---|---|
| Coarse Explanations | 0.2089 |
| Fine Explanations | 0.1296 |

Table 21: **Inter-Annotator Agreement for Explanation User Studies.** Using Fleiss' kappa, we measure the inter-annotator agreement for the human evaluation of coarse and fine saliency explanations. In both settings, the inter-annotator agreement is relatively low.

In light of this, we emphasize that the user study results do not contradict our paper's conclusions, as our work does not claim that the generated saliency explanations are plausible. Rather, we merely claim that using KG-based saliency explanations as additional supervision to regularize KG-augmented models can yield higher performance.

Our work appeals to the view that an explanation's quality should be measured by how well it distills knowledge for improving performance on some task [43]. Furthermore, the results of our user studies are actually in line with the conclusions from [45], which found that KG-augmented models can effectively leverage KG information to improve performance, but in a manner that may not make sense to humans.

### A.18 Training Hyperparameters

Since we consider a very large number of models and settings in our experiments, we only describe the core hyperparameters here. Let bsz denote batch size, let $lr_{text}$ denote text encoder learning rate, let $lr_{graph}$ denote graph encoder learning rate, and let $lr_{task}$ denote task predictor learning rate. Across all models (both baselines and SALKG), we generally used the following hyperparameter sweeps: $bsz = [8, 16, 32, 64]$, $lr_{text} = [1e-5, 2e-5, 3e-5, 5e-5]$, $lr_{graph} = [1e-4, 2e-4, 3e-4, 5e-4]$, and $lr_{task} = [1e-4, 2e-4, 3e-4, 5e-4]$. For CSQA and OBQA, we set the maximum number of epochs to 100. For CODAH, we set the maximum number of epochs to 30. For all three datasets, we used early stopping with a patience of 5 epochs. For more details about hyperparameters, please refer to our code repository.

### A.19 Computational Costs and Resources

Since the SALKG pipeline (as well as ORACLE, RANDOM, and HEURISTIC) involves training models across multiple stages, its computational costs are considerably greater than those from just training a No-KG or KG model individually. Specifically, the pipeline involves: **(1)** training the No-KG and KG models; **(2)** creating coarse/fine explanations from the No-KG and KG models; **(3)** training the SALKG-Coarse model; **(4)** training the SALKG-Fine model; and **(5)** training the SALKG-Hybrid model. In particular, using the Occl method to create fine explanations can be especially costly since it requires $n + 1$ KG model forward passes per KG, where $n$ is the number of units in the given KG. Also, if we tune the $T$ or $k$ thresholds comprehensively, then the total training time further increases. For reference, each of our experiments was run on one NVIDIA Quadro RTX 8000 GPU.

Nonetheless, since we are the first to propose regularizing KG-augmented models with saliency explanations, it is expected that not all components of our method will already be fully optimized. That is, the goal of our work is simply to introduce a new paradigm for training KG-augmented models and demonstrate its potential by showing that it can yield improved performance. Certainly, there are various parts of the SalKG pipeline whose efficiency can be improved. For example, we could explore faster explanation generation via some KG-specific heuristic/approximation, training SalKG-Hybrid with coarse/fine explanations in a single step (instead of Steps 3-5 above), or generating explanations that can cover multiple instances at a time. Such potential improvements could be interesting directions for future work.

### A.20 Related Work (Extended)

**Text-Based Explanations**   Many works have been proposed for explaining the predictions of language models, especially PLMs. Although some of these works focus on abstractive (free-text) explanations [44, 50, 64], most aim to provide extractive explanations which highlight salient tokens in the model's text input. Such extractive explanations typically use either gradient-based [51, 29, 10], attention-based [40, 53, 14, 25], and occlusion-based [12, 42, 22, 30] feature attribution methods. How feature attribution methods should be chosen remains an open question and the subject of much recent debate [2, 59, 48, 20]. While SALKG also uses feature attribution methods (e.g., G×I) to create extractive explanations, our study is limited to explanations regarding KG-augmented models' graph inputs.

**Graph-Based Explanations**   There are also methods proposing extractive explanations for graph encoders, especially GNNs. Such explanations are designed to point out components in the graph input that contribute most to the model's prediction. Some GNNs use attention for pooling, which naturally highlights nodes with higher attention weights [27, 26]. More sophisticated approaches use post-hoc optimization to identify salient nodes [19, 62] or subgraphs [62].

Unlike individual PLMs and graph encoders, KG-augmented models take both text and graph inputs. The KG-augmented model's graph encoder usually computes graph embeddings via attention pooling of nodes/paths, and the attention weights can be used to explain which nodes/paths in the input KG are salient [31, 13, 34, 56, 60]. These KG explanations can be interpreted as identifying knowledge in the KG that is complementary to the knowledge encoded in the PLM. However, there is little work on how such KG explanations should be used. SALKG considers graph-based extractive explanations of KG-augmented models, but focuses more on how explanations are used rather than created.

**Learning From Model Explanations**   To improve the model's learning, explanations can be used in a diverse range of ways, including as extra supervision or regularization [43, 17, 41, 1], pruned inputs [21, 3, 28], additional inputs [16, 8], and intermediate variables [58, 66, 44]. The most similar work to ours is [43], which proposed training a student model to mimic a teacher model's predictions by regularizing the student model's attention via text explanations created from the teacher model. However, [43] aims to evaluate explanations, while our goal is to improve performance via explanations. Still, methods for learning from explanations have largely focused on domains like text and images, as opposed to graphs. To the best of our knowledge, SALKG is the first work to train KG-augmented models using KG explanations as supervision.

### A.21 Societal Impact

Our proposed SALKG approach for learning from KG explanations can be applied to any KG-augmented model and can be adapted from any off-the-shelf saliency method. This enables KG-augmented models to improve generalization ability and learn more efficiently from data, thus yielding better performance while requiring less labeled data. However, in the present version of SALKG, this generalization ability and data efficiency comes with increased computational costs, as described in Sec. A.19. In the future, we plan to explore methods for improving generalization and data efficiency while minimizing computational costs.