# OpenReview forum: "SalKG: Learning From Knowledge Graph Explanations for Commonsense Reasoning"
_NeurIPS.cc/2021/Conference — NeurIPS 2021 Poster_

### Official Review · Reviewer_qWGA · 2021-07-14

**Rating:** 6
**Confidence:** 4

**Summary:**

The paper explores how saliency explanations can be used to improve the performance of KG-augmented models. The authors compared two types of saliency explanations: coarse (whether the KG is useful) and fine (which nodes or paths in the KG are useful) They first showed that using oracle KG saliency explanations as inputs improves model performance (section 4).  They then introduced a model that jointly learns to predict the saliency explanations, and solves the task by attending to corresponding KG features highlighted by the predicted explanations (SALKG). The results showed that SALKG models yield large performance gains in three popular commonsense QA benchmarks (CSQA, OBQA, CODAH).

**Limitations And Societal Impact:**

There's no foreseeable negative societal impact of this work.

**Main Review:**

In terms of originality, I'm not sure if the work is innovative. Several papers have used attentional mechanisms in KG-augmented NLP models (Hao et al. 2018 CCM, Wang et al. 2019 KGAT, Song et al. 2020 KGAnet). And the attention in those models is similar to the saliency explanations in the current work. For instance, Hao et al. 2018 proposed a commonsense knowledge-aware conversational model (CCM). It retrieves the relevant knowledge graph from ConceptNet given a post and then uses the static graph attention mechanism to extract the semantic information of the post. It seems that the authors are aware of those studies since they cited some of them in the Related Work section. I was curious to know the difference between the current work and the previous ones. It seems to me that both of them train the model to predict which part of the knowledge graph is relevant and use such predictions to enhance model performance.

The submission is relatively sound. The authors first demonstrate that KG-augmented models have large performance gains when provided explanations in the Oracal setting. Then they showed that the explanation does not need to be given since the model can learn to predict the explanation. I do have some concerns about the robustness of the performance. According to the statistical test results reported in Appendix A.10, some improvements are statistically insignificant (line 759) and others are significant under the criteria p<.1, which is not the conventional cutoff value (0.05) for rejecting a null hypothesis. If the authors could report the exact p-values obtained and confirm that the improvements are still significant with a more conventional cutoff value for significance, that would be great.

Another question I have is about Line 316 in the main paper. The author mentioned that "for RoBERTa+PathGen on CODAH, RANDOM-Coarse beats all other models". This is a counter-intuitive result. The authors argued that "SALKG’s limited performance may stem from the coarse explanations being unable to consistently discriminate between No-KG and KG, which can happen if No-KG and KG have very similar predictions". It would be great if the author can quantitatively verify that No-KG and KG indeed have very similar predictions in this case so that their explanations are valid. I doubt that is the case since the performance of No-KG and KG are different (see Table 6, PathGen column, row No-KG and row KG).

Finally, in main text 6.4 and Appendix A.5, the authors presented some case studies showing "the positive KGs/paths are useful for predicting the correct answer, and vice versa. " (main text Line 371). I'm worried about the risk of cherry-picking associated with these examples. To make a more convincing argument, the authors may consider presenting random samples to human raters that are naïve to this study and let them rate the usefulness of the KG/path. We can then compare whether the model's judgment aligns with human judgment.

 The submission is clearly written and well organized. If the authors can address the issues I mentioned above and make the arguments and conclusions in the paper more convincing, the results are important since it demonstrates that models can learn to predict the usefulness of knowledge graph (saliency explanation) and leverage such predictions to further improve model performance, although I don't think this observation is new (see my first paragraph above). This approach can be used in common-sense reasoning, question answering, etc.

----------------

Updated: I've read the authors' responses to my review as well as the ones to the other reviewers'. I think their responses address some of my concerns, especially the statistical significance of the main results and the strong performance of random explanation baselines. I think the updated results support the authors' arguments much better. I appreciate the newly added user studies. Although the model's predictions about the saliency explanation did not align with the human annotators, I agree with the authors that this does not contradict the paper's main claim. I appreciate the authors' clarification on the novelty of this paper. However, I still think this work presents a combination of some existing ideas and thus the novelty is limited. Given the improvements and the limitations, I decided to raise my score to 6 but no further.

**Time Spent Reviewing:**

10 hours

---

> ### Author Response · Authors · 2021-08-10
> **Response to Reviewer qWGA (Part 1)**
>
> Thank you for your thoughtful review! We are glad to hear that you liked our paper’s experiments, writing, and applicability to commonsense reasoning and question answering. We also appreciate your concerns and questions, each of which we address in detail below.
>
> ---
>
> ## Novelty of SalKG and our Contributions
> Regarding novelty, we believe there is some misunderstanding about our proposed SalKG method and what our contributions are.
>
> First, SalKG is an **explanation-based training procedure** which consists of the following steps:
>
> 1. Train a **"teacher" KG-augmented model, supervised only by the task labels**. The teacher model corresponds to the attention-based models introduced in prior works, but the teacher model itself is not our final model.
>
> 2. **Use some saliency method to generate explanations for the teacher model, with respect to the ground truth labels**. As described in Sec. 3 of our paper, we repurpose (via the sign-flipping in Equations 1-2) saliency explanations [1] to indicate which KG inputs most strongly influence the teacher model to predict the *correct* label. Whereas attention scores tell us which inputs the model *already* focused on, saliency explanations tell us which inputs the model *should* focus on, making saliency explanations a good choice for regularizing the model (Sec. 1).
>
> 3. **Train a "student" KG-augmented model**, which is **supervised by *both* the task labels and the saliency explanations**. In addition to the high-level supervision from the task labels, the saliency explanations provide low-level supervision by explicitly teaching the model which inputs to focus on. This student model is what we call our SalKG model. SalKG-Coarse is indeed an adaptive ensemble of the No-KG and KG models, weighted by the predicted coarse saliency explanations. However, SalKG-Fine is not an ensemble. SalKG-Fine has the same architecture as the KG model, but is trained by regularizing its attention scores to approximate the fine saliency explanations (i.e., using the attention scores as the predicted fine saliency explanations). Meanwhile, SalKG-Hybrid is an adaptive ensemble of the No-KG and SalKG-Fine models, hence using both coarse and fine saliency explanations.
>
> Second, we discuss several related lines of work and how SalKG differs from prior works:
> - **KG-Augmented Models**: Various KG-augmented model architectures have been proposed in the commonsense reasoning literature [2, 3, 4]. However, SalKG is orthogonal to these works, since SalKG is not an architecture. Rather, SalKG is a general explanation-based training algorithm that can be applied to different kinds of KG-augmented model architectures. Whereas existing KG-augmented models are supervised only by the task labels, SalKG-trained KG-augmented models are supervised by both the task labels and saliency explanations.
>
> - **Generating Explanations**: There are many existing methods for generating explanations of model behavior. These approaches include saliency methods [1] (e.g., gradient-based [5], occlusion-based [6]) and attention [7, 8]. Our paper does not propose a new method for generating explanations. Rather, we take existing saliency methods and repurpose them (via the sign-flipping in Equations 1-2) to generate KG-based explanation supervision, which we then propose to use for regularizing KG-augmented models via the SalKG approach.
>
> - **Learning from Explanations**: In the explanation-based learning literature, there have been various methods for learning from explanations (usually text-based or vision-based) [9], including using explanations as supervision for regularizing the model [10]. However, to the best of our knowledge, there has not been any work on explanation-based learning in the KG domain. While SalKG is indeed an explanation-based learning method, SalKG’s novelty comes from being the first method for learning from KG-based explanations. In particular, SalKG leverages the coarse (graph) to fine (node, path) hierarchy naturally found in KGs, in order to provide rich explanation-based learning signal to the model.
>
> In other words, the key contribution of our work is proposing to regularize the KG-augmented model with supervision from KG-based saliency explanations, which has not been done in any prior works. SalKG models are trained not only to predict the right answer, but rather to predict the right answer using the "right" inputs. We hypothesize that this additional explanation-based supervision provides a strong inductive bias which can help the model generalize better to unseen data. We will make this message more clear in the final version of the paper.
>
> ---
>
> [1] Bastings & Filippova. *The elephant in the interpretability room: Why use attention as explanation when we have saliency methods?* BlackboxNLP 2020.
>
> [2] Lin et al. *KagNet: Knowledge-Aware Graph Networks for Commonsense Reasoning*. EMNLP 2019.
>
> [3] Bosselut et al. *Dynamic Neuro-Symbolic Knowledge Graph Construction for Zero-shot Commonsense Question Answering*. AAAI 2021.
>
> [4] Yasunaga et al. *QA-GNN: Reasoning with Language Models and Knowledge Graphs for Question Answering*. NAACL 2021.
>
> [5] Denil et al. *Extraction of Salient Sentences from Labelled Documents*. arXiv 2014.
>
> [6] Li et al. *Understanding Neural Networks through Representation Erasure*. arXiv 2017.
>
> [7] Wiegreffe & Pinter. *Attention is not not Explanation*. EMNLP 2019.
>
> [8] Mohankumar et al. *Towards Transparent and Explainable Attention Models*. ACL 2020.
>
> [9] Hase & Bansal. *When Can Models Learn From Explanations? A Formal Framework for Understanding the Roles of Explanation Data*. arXiv 2021.
>
> [10] Pruthi et al. *Evaluating Explanations: How much do explanations from the teacher aid students?* arXiv 2020.

---

> > ### Author Response · Authors · 2021-08-10
> > **Response to Reviewer qWGA (Part 2)**
> >
> > ## Statistical Significance of Main Results
> >
> > In Sec. A.10 and Tables 15-16 of the appendix, we discussed the statistical significance of the main results, as given by the two-sided unpaired T-test with unequal variance. However, we found an unfair comparison in our evaluation protocol, which we have now fixed. For more details about the updated evaluation protocol, please refer to the "Strong Performance of Random Explanation Baselines" section of our response to Reviewer TCsM. Thus, given our new evaluation protocol, we display the new T-test results below:
> >
> > ### CSQA p-values
> >
> > | Model Pair | BERT+MHGRN | RoBERTa+MHGRN | BERT+PathGen | RoBERTa+PathGen | BERT+RN | RoBERTa+RN |
> > | ----------- | ----------- | ----------- | ----------- | ----------- | ----------- | ----------- |
> > | Best SalKG Model *vs.* Best Non-SalKG Model | 0.1235 | 0.4238 | 0.0701 | 0.2690 | 0.1336 | 0.0441 |
> >
> >
> > ### OBQA p-values
> >
> > | Model Pair (RoBERTa) | MHGRN | PathGen | RN |
> > | ----------- | ----------- | ----------- | ----------- |
> > | Best SalKG Model *vs.* Best Non-SalKG Model | 0.2909 | 0.8890 | 0.0005 |
> >
> >
> > ### CODAH p-values
> >
> > | Model Pair (RoBERTa) | MHGRN | PathGen |
> > | ----------- | ----------- | ----------- |
> > | Best SalKG Model *vs.* Best Non-SalKG Model | 0.1223 | 0.2823 |
> >
> > If we use threshold \alpha=0.1 (i.e., p < 0.1), then we find that SalKG yields statistically significant improvements on CSQA BERT+PathGen, CSQA RN+RoBERTa, and OBQA RN+RoBERTa. If we use threshold \alpha=0.05 (i.e., p < 0.05), then we find that SalKG yields statistically significant improvements on CSQA RN+RoBERTa and OBQA RN+RoBERTa. In particular, the improvement on OBQA RN+RoBERTa is very statistically significant, with p=0.0005.
> >
> > Our T-test results show that SalKG can produce significant performance gains on a number of model-dataset settings, although SalKG currently does not achieve large gains in all settings. In the final version of the paper, we will update our T-test results and qualify our claims to reflect these T-test results. For example, we will use “competitive performance” to describe SalKG’s effect on settings for which the improvement is not statistically significant.

---

> > ### Author Response · Authors · 2021-08-10
> > **Response to Reviewer qWGA (Part 3)**
> >
> > ## Strong Performance of Random Explanation Baselines
> >
> > This is a good observation, and investigating it led us to discover an unfair comparison within our evaluation protocol. This unfair comparison resulted in the random baseline performance being artificially inflated. After fixing our evaluation protocol, we found that the random baseline was much weaker than before and had lower performance than SalKG overall.
> >
> > First, we reiterate that the goal of comparing Random and Heuristic to SalKG is to show that saliency explanations provide better learning signal than random or heuristic explanations. In addition, we note that Random-Hybrid and Heuristic-Hybrid are not really expected to outperform their coarse and fine counterparts, if random and heuristic explanations do not capture useful learning signal. Second, let us define: (1) non-explanation models (No-KG, KG, No-KG + KG) as models that are not regularized with any kind of explanation and (2) explanation models (Random, Heuristic, SalKG) as models that are regularized with some kind of explanation.
> >
> > In our submission, each non-explanation model’s performance was reported as the average over three non-explanation model seeds, which we denote as the non-explanation seeds. Also, recall that each explanation model is built from No-KG and/or KG models. In our submission, we used the No-KG and/or KG models from only the best non-explanation seed to build the explanation models. Then, we reported the explanation model’s performance as the average over three explanation model seeds, which we call the explanation seeds, with respect to the single best non-explanation seed. That is, we computed the non-explanation model performance by averaging over [three non-explanation seeds] = [three total seeds], while computing the explanation model performance by averaging over [one (best) non-explanation seed] * [three explanation seeds] = [three total seeds].
> >
> > Our submission makes an unfair comparison between non-explanation and explanation models, since the explanation models are based only on the best non-explanation seed but are compared to the average of all three non-explanation seeds. Instead, for each of the three non-explanation seeds, we should train the explanation model on three explanation seeds, then compute the explanation model performance by averaging over [three non-explanation seeds] * [three explanation seeds] = [nine total seeds].
> >
> > We summarize the evaluation protocol below:
> > - **Non-explanation seeds**: 1, 2, 3 (assume 1 yields best performance)
> > - **Explanation seeds**: A, B, C
> > - **Non-explanation performance**: *avg*(1, 2, 3)
> > - **Explanation performance (unfair; submission version)**: *avg*(1A, 1B, 1C)
> > - **Explanation performance (fair; updated version)**: *avg*(1A, 1B, 1C, 2A, 2B, 2C, 3A, 3B, 3C)
> >
> > Below are the updated results for the explanation models using the fair evaluation protocol:
> >
> > ### CSQA Test Acc (%)
> >
> > | Model | BERT+MHGRN | RoBERTa+MHGRN | BERT+PathGen | RoBERTa+PathGen | BERT+RN | RoBERTa+PathGen |
> > | ----------- | ----------- | ----------- | ----------- | ----------- | ----------- | ----------- |
> > | Random-Coarse | 55.04 ± 1.44 | 71.06 ± 1.09 | 55.09 ± 1.08 | 71.15 ± 1.06 | 55.15 ± 1.23 | 69.06 ± 2.96 |
> > | Random-Fine | 54.69 ± 2.54 | 73.09 ± 1.06 | 54.66 ± 0.97 | 71.26 ± 3.19 | 49.88 ± 1.75 | 69.08 ± 1.95 |
> > | Random-Hybrid | 52.43 ± 2.60 | 71.93 ± 0.77 | 55.24 ± 0.58 | 71.35 ± 0.34 | 54.36 ± 0.35 | 70.12 ± 0.35 |
> > | Heuristic-Coarse | 55.55 ± 2.29 | 72.15 ± 0.84 | 56.92 ± 0.18 | 72.57 ± 0.49 | 56.42 ± 1.11 | 71.18 ± 0.77 |
> > | Heuristic-Fine | 52.54 ± 1.67 | 71.50 ± 1.01 | 54.00 ± 1.89 | 71.11 ± 0.93 | 52.04 ± 2.13 | 65.08 ± 3.67 |
> > | Heuristic-Hybrid | 56.35 ± 0.81 | 72.58 ± 0.32 | 56.83 ± 0.48 | 71.33 ± 0.87 | 54.38 ± 3.30 | 65.07 ± 2.02 |
> > | SalKG-Coarse | 57.98 ± 0.90 | 73.64 ± 1.05 | 57.75 ± 0.77 | 73.07 ± 0.25 | 57.50 ± 1.25 | 73.11 ± 1.13 |
> > | SalKG-Fine | 54.36 ± 2.34 | 70.00 ± 0.81 | 54.39 ± 2.03 | 72.12 ± 0.91 | 54.30 ± 1.41 | 71.64 ± 1.51 |
> > | SalKG-Hybrid | 58.70 ± 0.65 | 73.37 ± 0.12 | 59.87 ± 0.42 | 72.67 ± 0.65 | 58.78 ± 0.14 | 74.13 ± 0.71 |
> >
> >
> > ### OBQA Test Acc (%)
> >
> > | Model (RoBERTa) | MHGRN | PathGen | RN |
> > | ----------- | ----------- | ----------- | ----------- |
> > | Random-Coarse | 68.11 ± 1.12 | 67.18 ± 4.13 | 65.02 ± 2.57 |
> > | Random-Fine | 57.60 ± 5.33 | 55.13 ± 7.00 | 48.53 ± 4.82 |
> > | Random-Hybrid | 68.33 ± 0.40 | 69.53 ± 0.31 | 69.27 ± 0.12 |
> > | Heuristic-Coarse | 69.24 ± 2.47 | 65.58 ± 6.08 | 64.29 ± 3.06 |
> > | Heuristic-Fine | 57.27 ± 3.76 | 51.80 ± 2.95 | 50.53 ± 3.51 |
> > | Heuristic-Hybrid | 68.47 ± 0.23 | 68.40 ± 0.00 | 68.60 ± 0.20 |
> > | SalKG-Coarse | 69.93 ± 0.56 | 70.02 ± 0.55 | 71.29 ± 0.57 |
> > | SalKG-Fine | 64.82 ± 0.97 | 51.51 ± 0.87 | 62.29 ± 0.85 |
> > | SalKG-Hybrid | 70.20 ± 0.69 | 69.80 ± 0.49 | 70.47 ± 0.91 |
> >
> >
> > ### CODAH Test Acc (%)
> >
> > | Model (RoBERTa) | MHGRN | PathGen |
> > | ----------- | ----------- | ----------- |
> > | Random-Coarse | 83.48 ± 0.91 | 84.68 ± 1.65 |
> > | Random-Fine | 74.77 ± 6.90 | 80.48 ± 1.23 |
> > | Random-Hybrid | 83.86 ± 0.69 | 83.75 ± 0.60 |
> > | Heuristic-Coarse | 82.64 ± 0.10 | 82.52 ± 0.18 |
> > | Heuristic-Fine | 82.25 ± 1.43 | 82.55 ± 2.03 |
> > | Heuristic-Hybrid | 82.16 ± 2.11 | 82.73 ± 1.51 |
> > | SalKG-Coarse | 85.79 ± 1.83 | 85.43 ± 1.88 |
> > | SalKG-Fine | 84.08 ± 1.14 | 83.36 ± 0.81 |
> > | SalKG-Hybrid | 85.17 ± 0.54 | 84.42 ± 0.64 |
> >
> > Now, with the fair evaluation protocol, we see that the Random baseline is not as strong as before. Previously, Random’s performance was artificially inflated because its average did not take into account the two weaker non-explanation seeds. Meanwhile, SalKG still outperforms all other methods overall, which shows the effectiveness of regularizing the model with saliency explanations. In the final version of the paper, we will update the tables to reflect this new evaluation protocol.

---

> > ### Author Response · Authors · 2021-08-10
> > **Response to Reviewer qWGA (Part 4)**
> >
> > ## Quantitative Analysis of KG Saliency Explanations
> >
> > We agree that the case studies would be more informative with some quantitative analysis, so we have provided two new user studies here. Overall, we did not find strong evidence that the KG saliency explanations align well with human judgment. However, we also show that alignment with human judgment may not be a good metric for explanation quality, as our human annotators have low agreement about which explanations are useful. Ultimately, the user studies do not contradict our paper’s main claims, since we argue that KG saliency explanations can be used as additional supervision for improving performance, not that the explanations are plausible (i.e., convincing) to humans.
> >
> > ### User Study 1: Coarse Saliency Explanations
> > The first user study measures how well the coarse (graph-level) explanations align with human judgment of usefulness. Given a RoBERTa+PathGen model, we begin by uniformly sampling 25 high-saliency (positive) KGs and 25 low-saliency (negative) KGs from the CSQA training set. Recall that whether a KG is high-saliency or low-saliency was determined by coarse explanations (Sec. 3) generated with respect to the given model.
> >
> > Note that each KG corresponds to one answer choice of a question, so each question in CSQA has up to five corresponding KGs. To ensure that none of the KGs in our sample come from the same question, we ended up pruning two high-saliency and two low-saliency KGs, yielding a final sample of 23 high-saliency and 23 low-saliency KGs.
> >
> > Since a KG can contain hundreds of paths, it is not feasible to ask humans to evaluate the entire KG’s usefulness. Thus, as a very rough representation of the KG, we uniformly sampled three paths from the KG. Then, for each KG, we asked ten human annotators to score each of the three paths’ usefulness for predicting the same answer choice predicted by the RoBERTa+PathGen model. To score the paths, all annotators were also given the question, correct answer, and model’s predicted answer. The paths were scored on the following 0-2 scale:
> > - 0 = definitely not useful (i.e., *this path is either irrelevant or would cause someone to NOT select the model’s predicted answer*)
> > - 1 = possibly useful (i.e., *this path provides some support for selecting the model’s predicted answer*)
> > - 2 = definitely useful (i.e., *this path provides strong support for selecting the model’s predicted answer*)
> >
> > Finally, each KG’s score is computed as the mean of its three constituent path scores. Below, we show the mean and standard deviation scores for high-saliency and low-saliency graphs. We find that the two graph types have similar mean usefulness scores, while also having relatively large standard deviations. This suggests that coarse saliency explanations do not align strongly with human judgment. One key limitation of this study is that the three sampled paths may not be representative of the entire KG. In the future, we plan to redesign the user study to provide annotators a more comprehensive representation of the KG to evaluate.
> >
> > | Graph Type | Usefulness Score |
> > | ----------- | ----------- |
> > | High-Saliency Graph | 0.929 ± 0.734 |
> > | Low-Saliency Graph | 0.935 ± 0.764 |
> >
> >
> > ### User Study 2: Fine Saliency Explanations
> > The second user study measures how well the fine (path-level) explanations align with human judgment of usefulness. Given a RoBERTa+PathGen model trained on CSQA, we begin by uniformly sampling 25 correctly answered questions and 25 incorrectly answered questions from the CSQA training set. For each question, we take the model’s predicted answer choice and the KG corresponding to the predicted answer choice, then select: (1) the path with the highest fine saliency score, (2) the path with median fine saliency score, and (3) the path with the lowest saliency score. To get finer-grained saliency signal in this study, we consider the raw fine saliency scores, instead of the binarized fine explanations actually used to regularize the model. Recall that a path’s fine saliency score (Sec. 3) is calculated with respect to the given model.
> >
> > Next, we asked ten human annotators to score each path’s usefulness for predicting the same answer choice predicted by the RoBERTa+PathGen model. Like before, to score the paths, all annotators were also given the question, correct answer, and model’s predicted answer. Again, the paths were scored on the following 0-2 scale:
> > - 0 = definitely not useful (i.e., *this path is either irrelevant or would cause someone to NOT select the model’s predicted answer*)
> > - 1 = possibly useful (i.e., *this path provides some support for selecting the model’s predicted answer*)
> > - 2 = definitely useful (i.e., *this path provides strong support for selecting the model’s predicted answer*)
> >
> > Below, we show the mean scores for high-saliency, median-saliency, and low-saliency paths. We display these scores for paths from all predictions, correct predictions, and incorrect predictions. Overall, we find that the three path types have similar mean usefulness scores, although the mean score for median-saliency paths is somewhat higher than the other two path types’. Still, the standard deviations for all scores are relatively large, so this trend may not be meaningful. These results suggest that fine saliency explanations do not align strongly with human judgment. Additionally, we find that the path usefulness scores for correct predictions tend to be higher than those from incorrect predictions. This makes sense, since, intuitively, a model is more likely to predict the correct answer if it is using more useful knowledge as context.
> >
> > | Path Type | Usefulness Score (All Preds) | Usefulness Score (Correct Preds) | Usefulness Score (Incorrect Preds) |
> > | ----------- | ----------- | ----------- | ----------- |
> > | High-Saliency Graph | 1.091 ± 0.805 | 1.298 ± 0.782 | 0.884 ± 0.776 |
> > | Med-Saliency Graph | 1.222 ± 0.769 | 1.320 ± 0.729 | 1.124 ± 0.798 |
> > | Low-Saliency Graph | 1.060 ± 0.733 | 1.182 ± 0.730 | 0.938 ± 0.717 |
> >
> >
> > ### Inter-Annotator Agreement
> >
> > Here, we measure inter-annotator agreement for both user studies, using Fleiss’ kappa. For the user study of coarse explanations, the kappa score is 0.2089, which is on the borderline of slight agreement and fair agreement. For the user study of fine explanations, the kappa score is 0.1296, which indicates slight agreement.
> >
> > These low kappa scores show that even humans can hardly agree on whether the coarse/fine explanations are useful. Therefore, it may not always be beneficial to measure explanation quality in terms of alignment with human judgment. Moreover, this shows that weak alignment with human judgment does not necessarily imply poor explanation quality.
> >
> > | User Study | Fleiss' Kappa |
> > | ----------- | ----------- |
> > | Coarse Explanations | 0.2089 |
> > | Fine Explanations | 0.1296 |
> >
> >
> > ### Analysis
> > In our user studies, we did not find that coarse/fine saliency explanations align well with human judgment. However, we also found that human annotators had very low agreement about the usefulness of the explanations, which suggests that alignment with human judgment may not be the best measure of explanation quality.
> >
> > In light of this, we emphasize that the user study results do not contradict our paper’s conclusions, as our work does not claim that the generated saliency explanations are plausible. Rather, we merely claim that using KG-based saliency explanations as additional supervision to regularize KG-augmented models can yield higher performance. Our work appeals to the view that an explanation’s quality should be measured by how well it distills knowledge for improving performance on some task [1]. Furthermore, the results of our user studies are actually in line with the conclusions from [2], which found that KG-augmented models can effectively leverage KG information to improve performance, but in a manner that does not make sense to humans. In the final version of the paper, we will include the new user studies and analysis, so that readers can better understand the role and limitations of the KG saliency explanations.
> >
> > ---
> >
> > [1] Pruthi et al. *Evaluating Explanations: How much do explanations from the teacher aid students?* arXiv 2020.
> >
> > [2] Raman et al., *Learning to Deceive Knowledge Graph Augmented Models via Targeted Perturbation*. ICLR 2021.

---

> ### Author Response · Authors · 2021-08-15
> **Overview of Response to Reviewer qWGA**
>
> Hi Reviewer qWGA,
>
> Thank you again for your review! For your convenience, below is an overview of the points covered in our rebuttal:
> - Novelty of SalKG and our Contributions
> - Statistical Significance of Main Results
> - Strong Performance of Random Explanation Baselines
> - Quantitative Analysis of KG Saliency Explanations
>
> We hope that our rebuttal sufficiently addresses your concerns and questions. If there is still any uncertainty, please feel free to leave additional comments, so that we can help resolve any unclear aspects of the paper. Looking forward to discussing with you soon!
>
> Sincerely,
>
> Paper2834 Authors

---

### Official Review · Reviewer_jw6C · 2021-07-15

**Rating:** 6
**Confidence:** 3

**Summary:**

This paper proposes a framework, SALKG, for commonsense question answering (i.e., Multiple choice). It employs saliency explanations (i.e., coarse and fine) to regularize the attention mechanism and provide direct supervision for the task. The proposed framework is shown to achieve state-of-the-art or competitive results on three QA benchmarks, namely CSQA, OBQA, and CODAH. The proposed work has highlighted the right research questions and answered quantitatively and to some extent qualitatively. Overall, the paper is well motivated and proposes a technically solid framework for multiple-choice commonsense question answering tasks. I have some concerns regarding the details on using the knowledge graph that I have highlighted in the weaknesses section.

**Ethics Review Area:**

["I don’t know"]

**Main Review:**

Strengths:
1. The proposed framework, SALKG, is a technically solid framework for multiple-choice commonsense question answering tasks.

2. The idea of coarse/fine saliency explanations has the potential to provide direct supervision to the model focus on useful KG features.

3. The comprehensive quantitative analyses are conducted on three standard benchmarks.


==============================================================

Weaknesses:
1. ConceptNet has approximately 34 relation types. But, the most common types of positive relations are: RelatedTo, IsA, PartOf, and AtLocation. how many relation types were considered in this work? and Why? Moreover, ConceptNet has negative relation types (e.g., Antonym, DistinctFrom), were negative relation types also considered? Would these not influence the model in a bad way, if not excluded. In any case, the paper fails to explain any details on this part.

2. How the words/phrases (for saliency explanations) were selected from the text of the question and/or choices. For example, for the question, "What would you put in a teakettle?" how the n-grams for the paths are picked, as provided in the appendix, "(1) teakettle –[is a kind of]–> x <–[is at location]– water , (2) put –[is related to]–> x –[is used for]–> water , and (3) teakettle –[is a kind of]–> x –[is used for]–> water". What words/phrases from the question and the choice are picked and how? the paper fails to provide any details on this part as well. Moreover, was there any kind of text pre-processing (e.g., stemming,  lemmatization) performed? If yes, what are the trade-offs?

3. Concatenating the question with the answer choice(s) would make the text inarticulate. Does it have any effect on the embeddings?

4. The paper claims that the proposed framework achieves significant performance gains. But, are the results statistically significant? Was any statistical significance test performed?



**Time Spent Reviewing:**

15-20 hours

---

> ### Author Response · Authors · 2021-08-10
> **Response to Reviewer jw6C (Part 1)**
>
> Thank you for your thorough review! We are pleased to hear that you feel that SalKG is technically solid, that the idea of supervision via coarse/fine saliency explanations has potential, and that the quantitative analyses are comprehensive. We also appreciate your concerns and questions, each of which we address in detail below.
>
> ---
>
> ## Details about ConceptNet Relation Types Used in Our Work
>
> Following prior works on KG-augmented models [1, 2, 3], we considered all 34 ConceptNet relation types, including both positive and negative relation types. It is not clear why manually excluding all of the negative relation types would improve the model performance, since the negative relations can still provide potentially useful information for solving the task.  For example, suppose we are given: (1) the question "What is *concept1* capable of?", (2) answer choices *concept2* and *concept3*, and (3) the facts (*concept1*, *is capable of*, *concept2*) and (*concept1*, *is not capable of*, *concept3*). Here, the negative fact (*concept1*, *is not capable of*, *concept3*) can help the model rule out *concept3* as a potential answer choice. KG-augmented models (including the ones we consider in this work) are typically designed to reason over directed edges/paths in the KG, which may provide greater modeling flexibility for using negative relations appropriately.
>
> Often, the KG may contain irrelevant facts (can be positive or negative), so the model would ideally be trained to focus on relevant facts while ignoring irrelevant facts. If many of the negative facts are irrelevant, then we let the model learn from the training data how to ignore these facts. Therefore, since we do not have any preexisting knowledge of negative facts being useless overall, then we do not perform a priori removal of negative facts.
>
> In the table below, for each data split of CSQA, we provide the frequency count and percentage for each of the 34 ConceptNet relation types, across all contextualized KGs in the split. Here, we follow the *node-based* approach (see "Clarification about Construction of Contextualized KG" for more details) for building the contextualized KGs. In particular, we see that negative relation types (i.e., *is the antonym of*, *is not capable of*, *does not desire*) are relatively uncommon, constituting only 5.34%, 5.24%, and 5.34% of all relation types in the train, validation, and test splits, respectively.
>
> | Relation ID | Relation Type | Train Freq | Train Pct | Valid Freq | Valid Pct | Test Freq | Test Pct |
> | ----------- | ----------- | ----------- | ----------- | ----------- | ----------- | ----------- | ----------- |
> | 0 | is the antonym of (forward) | 2113903 | 2.65% | 255835 | 2.60% | 2113903 | 2.65% |
> | 1 | is at location of (forward) | 2306961 | 2.89% | 293787 | 2.99% | 2306961 | 2.89% |
> | 2 | is capable of (forward) | 198891 | 0.25% | 24892 | 0.25% | 198891 | 0.25% |
> | 3 | causes (forward) | 709966 | 0.89% | 87204 | 0.89% | 709966 | 0.89% |
> | 4 | is created by (forward) | 22479 | 0.03% | 2902 | 0.03% | 22479 | 0.03% |
> | 5 | is a kind of (forward) | 2148527 | 2.69% | 271080 | 2.76% | 2148527 | 2.69% |
> | 6 | desires (forward) | 55105 | 0.07% | 6893 | 0.07% | 55105 | 0.07% |
> | 7 | has subevent (forward) | 2177654 | 2.73% | 266785 | 2.72% | 2177654 | 2.73% |
> | 8 | is part of (forward) | 446759 | 0.56% | 55857 | 0.57% | 446759 | 0.56% |
> | 9 | has context (forward) | 0 | 0.00% | 0 | 0.00% | 0 | 0.00% |
> | 10 | has property (forward) | 187105 | 0.23% | 23144 | 0.24% | 187105 | 0.23% |
> | 11 | is made of (forward) | 64031 | 0.08% | 7861 | 0.08% | 64031 | 0.08% |
> | 12 | is not capable of (forward) | 2346 | 0.00% | 323 | 0.00% | 2346 | 0.00% |
> | 13 | does not desire (forward) | 14842 | 0.02% | 1745 | 0.02% | 14842 | 0.02% |
> | 14 | is (forward) | 39094 | 0.05% | 4380 | 0.04% | 39094 | 0.05% |
> | 15 | is related to (forward) | 28341680 | 35.53% | 3477057 | 35.39% | 28341680 | 35.53% |
> | 16 | is used for (forward) | 1052158 | 1.32% | 133350 | 1.36% | 1052158 | 1.32% |
> | 17 | is the antonym of (backward) | 2113903 | 2.65% | 255835 | 2.60% | 2113903 | 2.65% |
> | 18 | is at location of (backward) | 2306961 | 2.89% | 293787 | 2.99% | 2306961 | 2.89% |
> | 19 | is capable of (backward) | 198891 | 0.25% | 24892 | 0.25% | 198891 | 0.25% |
> | 20 | causes (backward) | 709966 | 0.89% | 87204 | 0.89% | 709966 | 0.89% |
> | 21 | is created by (backward) | 22479 | 0.03% | 2902 | 0.03% | 22479 | 0.03% |
> | 22 | is a kind of (backward) | 2148527 | 2.69% | 271080 | 2.76% | 2148527 | 2.69% |
> | 23 | desires (backward) | 55105 | 0.07% | 6893 | 0.07% | 55105 | 0.07% |
> | 24 | has subevent (backward) | 2177654 | 2.73% | 266785 | 2.72% | 2177654 | 2.73% |
> | 25 | is part of (backward) | 446759 | 0.56% | 55857 | 0.57% | 446759 | 0.56% |
> | 26 | has context (backward) | 0 | 0.00% | 0 | 0.00% | 0 | 0.00% |
> | 27 | has property (backward) | 187105 | 0.23% | 23144 | 0.24% | 187105 | 0.23% |
> | 28 | is made of (backward) | 64031 | 0.08% | 7861 | 0.08% | 64031 | 0.08% |
> | 29 | is not capable of (backward) | 2346 | 0.00% | 323 | 0.00% | 2346 | 0.00% |
> | 30 | does not desire (backward) | 14842 | 0.02% | 1745 | 0.02% | 14842 | 0.02% |
> | 31 | is (backward) | 39094 | 0.05% | 4380 | 0.04% | 39094 | 0.05% |
> | 32 | is related to (backward) | 28341680 | 35.53% | 3477057 | 35.39% | 28341680 | 35.53% |
> | 33 | is used for (backward) | 1052158 | 1.32% | 133350 | 1.36% | 1052158 | 1.32% |
> | Total | | 79763002 | 100.00% | 9826190 | 100.00% | 79763002 | 100.00% |
>
>
>
> ---
>
> [1] Feng et al. *Scalable Multi-Hop Relational Reasoning for Knowledge-Aware Question Answering*. EMNLP 2020.
>
> [2] Wang et al. *Connecting the Dots: A Knowledgeable Path Generator for Commonsense Question Answering*. Findings of EMNLP 2020.
>
> [3] Lin et al. *KagNet: Knowledge-Aware Graph Networks for Commonsense Reasoning*. EMNLP 2019.

---

> > ### Author Response · Authors · 2021-08-10
> > **Response to Reviewer jw6C (Part 2)**
> >
> > ## Clarification about Construction of Contextualized KG
> >
> > This is a great question, which we originally answered in Sec. 2 but unfortunately had to cut out to make the page limit.
> >
> > In Sec. 2, we defined the full KG as \tilde{G} = (\tilde{V}, \tilde{R}, \tilde{E}), where \tilde{V}, \tilde{R}, and \tilde{E} are all of the KG’s nodes (concepts), relations, and edges (facts), respectively. For each instance, we assume access to \tilde{G} but do not use the entire KG in practice. Given a question q and an answer choice a_i for some instance, we construct the contextualized KG, G_i = (V_i, R_i, E_i) by heuristically extracting edges from \tilde{G}, following the approach taken by most prior KG-augmented model works.
> >
> > G_i = (V_i, R_i, E_i) is built differently for node-based models and path-based models, and we describe both types of contextualized KG construction procedures below. Note that these procedures are not designed by us, but simply follow what was proposed and shown to work well in the KG-augmented models’ original papers [1, 2]. Thus, we do not experiment with different contextualized KG construction procedures, since it is out of the scope of our work.
> >
> > - Let us define the KG nodes mentioned in q and a_i as **QA nodes**. For example, for the question "What would you put in a teakettle?" and answer choice "water", the QA nodes would be *put*, *teakettle*, and *water*. We ground raw mentions of QA nodes to the KG via spaCy-based lemmatization and stop-word filtering.
> >
> > - For **node-based models** (MHGRN [1]), we select V_i \subseteq \tilde{V} as the QA nodes and all nodes in the QA nodes’ 1-hop KG neighborhood. Next, we choose R_i \subseteq \tilde{R} as all of the relations between concepts in V_i. Finally, we take E_i \subseteq \tilde{E} as all of the edges involving V_i and R_i.
> >
> > - For **path-based models** (PathGen [2], RN [1, 2]), we select G_i as all 2-hop paths between all question-answer node pairs. Thus, V_i \subseteq \tilde{V} consists of the QA nodes as well as all intermediate nodes in the 2-hop paths. Meanwhile, R_i \subseteq \tilde{R} and E_i \subseteq \tilde{E} consist of all relations and edges within the 2-hop paths. When reasoning over the 2-hop paths, the model does not actually use the intermediate nodes, perhaps in order to keep the path more general.
> >
> > In the final version of the paper, we will include these details in the appendix, since we agree it is important for understanding how the KG-augmented models work.
> >
> > ---
> >
> > [1] Feng et al. *Scalable Multi-Hop Relational Reasoning for Knowledge-Aware Question Answering*. EMNLP 2020.
> >
> > [2] Wang et al. *Connecting the Dots: A Knowledgeable Path Generator for Commonsense Question Answering*. Findings of EMNLP 2020.

---

> > ### Author Response · Authors · 2021-08-10
> > **Response to Reviewer jw6C (Part 3)**
> >
> > ## Effects of Concatenating Question with Answer Choice
> >
> > Thanks for pointing this out. To clarify, when we concatenate the question with the answer choice, the question and answer choice are separated by a special separator token ('[SEP]' for BERT, and '</s>' for RoBERTa). The special separator token lets the model know which part of the text input is the question and which part is the answer choice.
> >
> > For example, given question "What would you put in a teakettle?" and answer choice "water", the BERT tokenizer would output: "[CLS] what would you put in a teakettle ? [SEP] water [SEP]", where '[CLS]' is BERT's beginning-of-sentence token which is used to represent the entire sentence. Note that a second '[SEP]' token is inserted after the answer choice to indicate the end of the text input.
> >
> > This practice of concatenating the question and answer choice (delimited by the aforementioned special tokens) follows many prior works about KG-augmented models (with BERT-style text encoders) trained on multi-choice QA [1, 2].
> >
> > ---
> >
> > [1] Feng et al. *Scalable Multi-Hop Relational Reasoning for Knowledge-Aware Question Answering*. EMNLP 2020.
> >
> > [2] Wang et al. *Connecting the Dots: A Knowledgeable Path Generator for Commonsense Question Answering*. Findings of EMNLP 2020.

---

> > ### Author Response · Authors · 2021-08-10
> > **Response to Reviewer jw6C (Part 4)**
> >
> > ## Statistical Significance of Main Results
> >
> > In Sec. A.10 and Tables 15-16 of the appendix, we discussed the statistical significance of the main results, as given by the two-sided unpaired T-test with unequal variance. However, we found an unfair comparison in our evaluation protocol, which we have now fixed. For more details about the updated evaluation protocol, please refer to the "Strong Performance of Random Explanation Baselines" section of our response to Reviewer TCsM. Thus, given our new evaluation protocol, we display the new T-test results below:
> >
> > ### CSQA p-values
> >
> > | Model Pair | BERT+MHGRN | RoBERTa+MHGRN | BERT+PathGen | RoBERTa+PathGen | BERT+RN | RoBERTa+RN |
> > | ----------- | ----------- | ----------- | ----------- | ----------- | ----------- | ----------- |
> > | Best SalKG Model *vs.* Best Non-SalKG Model | 0.1235 | 0.4238 | 0.0701 | 0.2690 | 0.1336 | 0.0441 |
> >
> >
> > ### OBQA p-values
> >
> > | Model Pair (RoBERTa) | MHGRN | PathGen | RN |
> > | ----------- | ----------- | ----------- | ----------- |
> > | Best SalKG Model *vs.* Best Non-SalKG Model | 0.2909 | 0.8890 | 0.0005 |
> >
> >
> > ### CODAH p-values
> >
> > | Model Pair (RoBERTa) | MHGRN | PathGen |
> > | ----------- | ----------- | ----------- |
> > | Best SalKG Model *vs.* Best Non-SalKG Model | 0.1223 | 0.2823 |
> >
> > If we use threshold \alpha=0.1 (i.e., p < 0.1), then we find that SalKG yields statistically significant improvements on CSQA BERT+PathGen, CSQA RN+RoBERTa, and OBQA RN+RoBERTa. If we use threshold \alpha=0.05 (i.e., p < 0.05), then we find that SalKG yields statistically significant improvements on CSQA RN+RoBERTa and OBQA RN+RoBERTa. In particular, the improvement on OBQA RN+RoBERTa is very statistically significant, with p=0.0005.
> >
> > Our T-test results show that SalKG can produce significant performance gains on a number of model-dataset settings, although SalKG currently does not achieve large gains in all settings. In the final version of the paper, we will update our T-test results and qualify our claims to reflect these T-test results. For example, we will use “competitive performance” to describe SalKG’s effect on settings for which the improvement is not statistically significant.

---

> ### Author Response · Authors · 2021-08-15
> **Overview of Response to Reviewer jw6C**
>
> Hi Reviewer jw6C,
>
> Thank you again for your review! For your convenience, below is an overview of the points covered in our rebuttal:
> - Details about ConceptNet Relation Types Used in Our Work
> - Clarification about Construction of Contextualized KG
> - Effects of Concatenating Question with Answer Choice
> - Statistical Significance of Main Results
>
> We hope that our rebuttal sufficiently addresses your concerns and questions. If there is still any uncertainty, please feel free to leave additional comments, so that we can help resolve any unclear aspects of the paper. Looking forward to discussing with you soon!
>
> Sincerely,
>
> Paper2834 Authors

---

> ### Author Response · Authors · 2021-08-25
> **Friendly reminder to respond to author rebuttal**
>
> Hi Reviewer jw6C,
>
> Thank you again for your review! Based on your thoughtful feedback, we wrote a detailed rebuttal covering the following points:
> - Details about ConceptNet Relation Types Used in Our Work
> - Clarification about Construction of Contextualized KG
> - Effects of Concatenating Question with Answer Choice
> - Statistical Significance of Main Results
>
> We would love to hear your thoughts about our rebuttal, including whether it sufficiently addresses your concerns and questions. Any feedback is welcome and greatly appreciated!
>
> Sincerely,
>
> Paper2834 Authors

---

> > ### Author Response · Authors · 2021-08-31
> > **Follow-up reminder to respond to author rebuttal**
> >
> > Hi Reviewer jw6C,
> >
> > Just wanted to follow up on our previous reminder to respond to the rebuttal. In light of the imminent discussion deadline on **September 1**, it would be awesome to know if our rebuttal sufficiently addressed your concerns and questions. We understand that you are busy, so we would appreciate it a lot!
> >
> > Thanks!
> >
> > Paper2834 Authors

---

### Official Review · Reviewer_e1cd · 2021-07-16

**Rating:** 6
**Confidence:** 3

**Summary:**

Motivated by that the knowledge graph used in KG-augmented models is useless sometimes, the authors propose two ensemble models by using coarse/fine saliency explanations. Specifically, the authors first get saliency explanations through F_KG, F_NO-KG and saliency methods. Then, the authors propose oracle model using coarse/fine/hybrid explanations as extra inputs, and achieves superb performance than its baselines. Finally, based on oracle models, the authors develop the SALKG framework that uses explanations as supervision for distillation learning. SALKG achieves SOTA performance on CommonsenseQA and competitive performance on OBQA. It is the first to supervise KG-augmented models with KG explanations.

**Limitations And Societal Impact:**

Please refer to the main reviews.

**Main Review:**

Pros:
- Well-written introduction of the basic concepts and structure of KG-augmented models.
- The idea of learning from KG explanations can be applied to other KG-augmented methods and probably generalizes to other tasks such as reading comprehension, and thus would be useful to a wider community.
- The coarse saliency explanations indicate when KG helps the model and are gained from the differences between the confidence probabilities of F_KG and F_NO-KG. The fine saliency explanations show whether a unit in KG is useful and are derived from existing effective saliency method. The design of two kinds of saliency explanations is reasonable and effective.
- Thorough ablation studies to explain the behavior of the model, which contain the discussion of two kind of explanations, the saliency methods and the loss function.

Cons and Questions:
- Intuitively, the fine-based model should be better than coarse-based model. However, SALKG-Coarse outperforms SALKG-Fine in experiments. And the SALKG-Fine performs worse than its original KG-augmented model as shown in Table 6. It seems that there is no analysis of these results.
- The case studies seem insufficient. SARKG learns the distilled knowledge from saliency explanations. In the words of authors, SARKG is explicitly taught which KG components should be used. However, the case study just gives examples that the knowledge graph is useless or introduces noise. There is no comparison of attention weights learned by SARKG and KG-augmented models.
- The different KG unit types of fine explanation in Table 1, i.e., node and path, which one is more efficient when they server as fine saliency explanations?
- The training costs of the proposed model is much greater than baselines.
- Presentation problems:
    (1) Many references have been published, but the authors still used their reference formats in arxiv.
    (2) In Figure 3, it seems like the arrows indicating the SALKG pipeline is purple rather than blue described in the caption.

**Time Spent Reviewing:**

4

---

> ### Author Response · Authors · 2021-08-10
> **Response to Reviewer e1cd (Part 1)**
>
> Thank you for your comprehensive review! We are happy to hear that you liked our introduction of KG-augmented models, direction of learning from KG explanations, KG saliency explanation design, and ablation studies. We also appreciate your concerns and questions, each of which we address in detail below.
>
> ---
>
> ## Analysis of SalKG-Fine Performance
> Thank you for pointing this out! The KG model and SalKG-Fine model both assume that the KG should always be used to solve the given instance. Meanwhile, the success of SalKG-Coarse shows that the KG sometimes may not be useful. But why does SalKG-Fine generally perform worse than the KG model?
>
> The reason is that SalKG-Fine is more strongly committed to the flawed assumption of universal KG usefulness. Whereas the KG model is trained to solve the task always using the KG as context, the SalKG-Fine model is trained to both solve the task always using the KG as context (global KG supervision) and attend to specific parts of the KG (local KG supervision). Since SalKG-Fine is trained with both global and local KG supervision, it is much more likely to overfit, as the KG is not actually useful for all instances. That is, for training instances where the KG should not be used, SalKG-Fine is pushed to not only use the KG, but also to attend to specific parts of the KG. This leads to a SalKG-Fine model that does not generalize well to test instances where the KG is not useful.
>
> To address this issue, we proposed the SalKG-Hybrid model, which takes the best of both SalKG-Coarse and SalKG-Fine. For a given instance, SalKG-Hybrid does the following:
> - Use its SalKG-Coarse component to predict whether the KG is useful.
> - If the KG is predicted to be useful, use its SalKG-Fine component to attend to the useful parts of the KG.
>
> Indeed, we find that SalKG-Hybrid is the best-performing model overall, among all models we experimented with. These results support our hypothesis about why SalKG-Fine performs relatively poorly. In the final version of our paper, we will include a more comprehensive discussion of SalKG-Fine’s limited performance.

---

> > ### Author Response · Authors · 2021-08-10
> > **Response to Reviewer e1cd (Part 2)**
> >
> > ## Quantitative Analysis of KG Saliency Explanations
> >
> > We agree that the case studies would be more informative with some quantitative analysis, so we have provided two new user studies here. Overall, we did not find strong evidence that the KG saliency explanations align well with human judgment. However, we also show that alignment with human judgment may not be a good metric for explanation quality, as our human annotators have low agreement about which explanations are useful. Ultimately, the user studies do not contradict our paper’s main claims, since we argue that KG saliency explanations can be used as additional supervision for improving performance, not that the explanations are plausible (i.e., convincing) to humans.
> >
> > ### User Study 1: Coarse Saliency Explanations
> > The first user study measures how well the coarse (graph-level) explanations align with human judgment of usefulness. Given a RoBERTa+PathGen model, we begin by uniformly sampling 25 high-saliency (positive) KGs and 25 low-saliency (negative) KGs from the CSQA training set. Recall that whether a KG is high-saliency or low-saliency was determined by coarse explanations (Sec. 3) generated with respect to the given model.
> >
> > Note that each KG corresponds to one answer choice of a question, so each question in CSQA has up to five corresponding KGs. To ensure that none of the KGs in our sample come from the same question, we ended up pruning two high-saliency and two low-saliency KGs, yielding a final sample of 23 high-saliency and 23 low-saliency KGs.
> >
> > Since a KG can contain hundreds of paths, it is not feasible to ask humans to evaluate the entire KG’s usefulness. Thus, as a very rough representation of the KG, we uniformly sampled three paths from the KG. Then, for each KG, we asked ten human annotators to score each of the three paths’ usefulness for predicting the same answer choice predicted by the RoBERTa+PathGen model. To score the paths, all annotators were also given the question, correct answer, and model’s predicted answer. The paths were scored on the following 0-2 scale:
> > - 0 = definitely not useful (i.e., *this path is either irrelevant or would cause someone to NOT select the model’s predicted answer*)
> > - 1 = possibly useful (i.e., *this path provides some support for selecting the model’s predicted answer*)
> > - 2 = definitely useful (i.e., *this path provides strong support for selecting the model’s predicted answer*)
> >
> > Finally, each KG’s score is computed as the mean of its three constituent path scores. Below, we show the mean and standard deviation scores for high-saliency and low-saliency graphs. We find that the two graph types have similar mean usefulness scores, while also having relatively large standard deviations. This suggests that coarse saliency explanations do not align strongly with human judgment. One key limitation of this study is that the three sampled paths may not be representative of the entire KG. In the future, we plan to redesign the user study to provide annotators a more comprehensive representation of the KG to evaluate.
> >
> > | Graph Type | Usefulness Score |
> > | ----------- | ----------- |
> > | High-Saliency Graph | 0.929 ± 0.734 |
> > | Low-Saliency Graph | 0.935 ± 0.764 |
> >
> >
> > ### User Study 2: Fine Saliency Explanations
> > The second user study measures how well the fine (path-level) explanations align with human judgment of usefulness. Given a RoBERTa+PathGen model trained on CSQA, we begin by uniformly sampling 25 correctly answered questions and 25 incorrectly answered questions from the CSQA training set. For each question, we take the model’s predicted answer choice and the KG corresponding to the predicted answer choice, then select: (1) the path with the highest fine saliency score, (2) the path with median fine saliency score, and (3) the path with the lowest saliency score. To get finer-grained saliency signal in this study, we consider the raw fine saliency scores, instead of the binarized fine explanations actually used to regularize the model. Recall that a path’s fine saliency score (Sec. 3) is calculated with respect to the given model.
> >
> > Next, we asked ten human annotators to score each path’s usefulness for predicting the same answer choice predicted by the RoBERTa+PathGen model. Like before, to score the paths, all annotators were also given the question, correct answer, and model’s predicted answer. Again, the paths were scored on the following 0-2 scale:
> > - 0 = definitely not useful (i.e., *this path is either irrelevant or would cause someone to NOT select the model’s predicted answer*)
> > - 1 = possibly useful (i.e., *this path provides some support for selecting the model’s predicted answer*)
> > - 2 = definitely useful (i.e., *this path provides strong support for selecting the model’s predicted answer*)
> >
> > Below, we show the mean scores for high-saliency, median-saliency, and low-saliency paths. We display these scores for paths from all predictions, correct predictions, and incorrect predictions. Overall, we find that the three path types have similar mean usefulness scores, although the mean score for median-saliency paths is somewhat higher than the other two path types’. Still, the standard deviations for all scores are relatively large, so this trend may not be meaningful. These results suggest that fine saliency explanations do not align strongly with human judgment. Additionally, we find that the path usefulness scores for correct predictions tend to be higher than those from incorrect predictions. This makes sense, since, intuitively, a model is more likely to predict the correct answer if it is using more useful knowledge as context.
> >
> > | Path Type | Usefulness Score (All Preds) | Usefulness Score (Correct Preds) | Usefulness Score (Incorrect Preds) |
> > | ----------- | ----------- | ----------- | ----------- |
> > | High-Saliency Graph | 1.091 ± 0.805 | 1.298 ± 0.782 | 0.884 ± 0.776 |
> > | Med-Saliency Graph | 1.222 ± 0.769 | 1.320 ± 0.729 | 1.124 ± 0.798 |
> > | Low-Saliency Graph | 1.060 ± 0.733 | 1.182 ± 0.730 | 0.938 ± 0.717 |
> >
> >
> > ### Inter-Annotator Agreement
> >
> > Here, we measure inter-annotator agreement for both user studies, using Fleiss’ kappa. For the user study of coarse explanations, the kappa score is 0.2089, which is on the borderline of slight agreement and fair agreement. For the user study of fine explanations, the kappa score is 0.1296, which indicates slight agreement.
> >
> > These low kappa scores show that even humans can hardly agree on whether the coarse/fine explanations are useful. Therefore, it may not always be beneficial to measure explanation quality in terms of alignment with human judgment. Moreover, this shows that weak alignment with human judgment does not necessarily imply poor explanation quality.
> >
> > | User Study | Fleiss' Kappa |
> > | ----------- | ----------- |
> > | Coarse Explanations | 0.2089 |
> > | Fine Explanations | 0.1296 |
> >
> >
> > ### Analysis
> > In our user studies, we did not find that coarse/fine saliency explanations align well with human judgment. However, we also found that human annotators had very low agreement about the usefulness of the explanations, which suggests that alignment with human judgment may not be the best measure of explanation quality.
> >
> > In light of this, we emphasize that the user study results do not contradict our paper’s conclusions, as our work does not claim that the generated saliency explanations are plausible. Rather, we merely claim that using KG-based saliency explanations as additional supervision to regularize KG-augmented models can yield higher performance. Our work appeals to the view that an explanation’s quality should be measured by how well it distills knowledge for improving performance on some task [1]. Furthermore, the results of our user studies are actually in line with the conclusions from [2], which found that KG-augmented models can effectively leverage KG information to improve performance, but in a manner that does not make sense to humans. In the final version of the paper, we will include the new user studies and analysis, so that readers can better understand the role and limitations of the KG saliency explanations.
> >
> > ---
> >
> > [1] Pruthi et al. *Evaluating Explanations: How much do explanations from the teacher aid students?* arXiv 2020.
> >
> > [2] Raman et al., *Learning to Deceive Knowledge Graph Augmented Models via Targeted Perturbation*. ICLR 2021.

---

> > ### Author Response · Authors · 2021-08-10
> > **Response to Reviewer e1cd (Part 3)**
> >
> > ## Comparison of Fine Unit Types
> >
> > In our work, we designed SalKG to be a general framework for explanation-based regularization of KG-augmented models. Thus, to describe SalKG-Fine in a more unified manner, we abstracted nodes and paths into the higher-level concept of fine units. This may have created the misleading impression that, for a given KG-augmented model, we can generate fine explanations w.r.t. arbitrary fine unit types. However, in practice, KG-augmented models can be quite diverse yet restrictive in how they process KG information --- e.g., some KG-augmented models explicitly reason over the KG’s nodes, while others reason specifically over the KG’s paths. Hence, it is not straightforward to create path explanations for a node-based model, and vice versa.
> >
> > In light of this, for a given KG-augmented model, it would not make sense to compare both node and path explanations, since the model is only capable of reasoning over one of the fine unit types. Therefore, for each KG-augmented model, we only consider its natural fine unit type and use the model’s performance as a proxy for how fine explanations of its unit type perform in general (i.e., MHGRN is representative of node explanations, while PathGen and RN are representative of path explanations).
> >
> > ---
> >
> > ## Acknowledgment of Training Costs
> >
> > We acknowledge that SalKG currently involves higher training costs than just training the No-KG or KG model. To recap, the full SalKG pipeline involves:
> > 1. Training the No-KG and KG models.
> > 2. Creating coarse/fine explanations from the No-KG and KG models.
> > 3. Training the SalKG-Coarse model.
> > 4. Training the SalKG-Fine model.
> > 5. Training the SalKG-Hybrid model.
> >
> > In particular, using the Occl method to create fine explanations can be especially costly since it requires n+1 KG model forward passes per KG, where n is the number of units in the given KG.
> >
> > Nonetheless, since we are the first to propose regularizing KG-augmented models with saliency explanations, it is expected that not all components of our method will already be fully optimized. That is, the goal of our work is simply to introduce a new paradigm for training KG-augmented models and demonstrate its potential by showing that it can yield improved performance. Certainly, there are various parts of the SalKG pipeline whose efficiency can be improved. For example, we could explore faster explanation generation via some KG-specific heuristic/approximation, training SalKG-Hybrid with coarse/fine explanations in a single step (instead of Steps 3-5 above), or generating explanations that can cover multiple instances at a time. However, we feel that these potential improvements are out of the scope of our current submission and should be addressed in future work.
> >
> > ---
> >
> > ## Presentation Issues
> > Thank you for pointing out these issues. We will make sure the references and color coding are fixed in the final version of the paper.

---

> > > ### Comment · Reviewer_e1cd · 2021-09-03
> > > **Thanks for your responses**
> > >
> > > Thanks for your detailed response and new experimental results. These responses resolve some of my concerns and make this paper more convincing. But I'd like to keep my score due to the novelty limitation.

---

> > > > ### Author Response · Authors · 2021-09-03
> > > > **Clarification about SalKG's novelty**
> > > >
> > > > Hi Reviewer e1cd,
> > > >
> > > > Thanks for your comment! We are glad to hear that our rebuttal helped resolve some of your concerns. Since concerns about novelty were not mentioned in your initial review, it would be great if you elaborate on the reasons for your updated assessment of SalKG's novelty.
> > > >
> > > > Also, we would like to clarify why we believe SalKG is novel. In the summary section of your review, you stated that "the authors propose two ensemble models by using coarse/fine saliency explanations", so we feel there may be some confusion about SalKG and our contributions.
> > > >
> > > > First, SalKG is an **explanation-based training procedure** which consists of the following steps:
> > > >
> > > > 1. Train a **"teacher" KG-augmented model, supervised only by the task labels**. The teacher model corresponds to the KG-augmented models introduced in prior works, but the teacher model itself is not our final model.
> > > >
> > > > 2. **Use some saliency method to generate explanations for the teacher model, with respect to the ground truth labels**. As described in Sec. 3 of our paper, we repurpose (via the sign-flipping in Equations 1-2) saliency explanations [1] to indicate which KG inputs most strongly influence the teacher model to predict the *correct* label. Whereas attention scores tell us which inputs the model *already* focused on, saliency explanations tell us which inputs the model *should* focus on, making saliency explanations a good choice for regularizing the model (Sec. 1).
> > > >
> > > > 3. **Train a "student" KG-augmented model**, which is **supervised by *both* the task labels and the saliency explanations**. In addition to the high-level supervision from the task labels, the saliency explanations provide low-level supervision by explicitly teaching the model which inputs to focus on. This student model is what we call our SalKG model. SalKG-Coarse is indeed an adaptive ensemble of the No-KG and KG models, weighted by the predicted coarse saliency explanations. However, SalKG-Fine is not an ensemble. SalKG-Fine has the same architecture as the KG model, but is trained by regularizing its attention scores to approximate the fine saliency explanations (i.e., using the attention scores as the predicted fine saliency explanations). Meanwhile, SalKG-Hybrid is an adaptive ensemble of the No-KG and SalKG-Fine models, hence using both coarse and fine saliency explanations.
> > > >
> > > > Second, we discuss several related lines of work and how SalKG differs from prior works:
> > > > - **KG-Augmented Models**: Various KG-augmented model architectures have been proposed in the commonsense reasoning literature [2, 3, 4]. However, SalKG is orthogonal to these works, since SalKG is not an architecture. Rather, SalKG is a general explanation-based training algorithm that can be applied to different kinds of KG-augmented model architectures. Whereas existing KG-augmented models are supervised only by the task labels, SalKG-trained KG-augmented models are supervised by both the task labels and saliency explanations.
> > > >
> > > > - **Generating Explanations**: There are many existing methods for generating explanations of model behavior. These approaches include saliency methods [1] (e.g., gradient-based [5], occlusion-based [6]) and attention [7, 8]. Our paper does not propose a new method for generating explanations. Rather, we take existing saliency methods and repurpose them (via the sign-flipping in Equations 1-2) to generate KG-based explanation supervision, which we then propose to use for regularizing KG-augmented models via the SalKG approach.
> > > >
> > > > - **Learning from Explanations**: In the explanation-based learning literature, there have been various methods for learning from explanations (usually text-based or vision-based) [9], including using explanations as supervision for regularizing the model [10]. However, to the best of our knowledge, there has not been any work on explanation-based learning in the KG domain. While SalKG is indeed an explanation-based learning method, SalKG’s novelty comes from being the first method for learning from KG-based explanations. In particular, SalKG leverages the coarse (graph) to fine (node, path) hierarchy naturally found in KGs, in order to provide rich explanation-based learning signal to the model.
> > > >
> > > > In other words, the key contribution of our work is proposing to regularize the KG-augmented model with supervision from KG-based saliency explanations, which has not been done in any prior works. SalKG models are trained not only to predict the right answer, but rather to predict the right answer using the "right" inputs. We hypothesize that this additional explanation-based supervision provides a strong inductive bias which can help the model generalize better to unseen data. We will make this message more clear in the final version of the paper.
> > > >
> > > > ---
> > > >
> > > > [1] Bastings & Filippova. *The elephant in the interpretability room: Why use attention as explanation when we have saliency methods?* BlackboxNLP 2020.
> > > >
> > > > [2] Lin et al. *KagNet: Knowledge-Aware Graph Networks for Commonsense Reasoning*. EMNLP 2019.
> > > >
> > > > [3] Bosselut et al. *Dynamic Neuro-Symbolic Knowledge Graph Construction for Zero-shot Commonsense Question Answering*. AAAI 2021.
> > > >
> > > > [4] Yasunaga et al. *QA-GNN: Reasoning with Language Models and Knowledge Graphs for Question Answering*. NAACL 2021.
> > > >
> > > > [5] Denil et al. *Extraction of Salient Sentences from Labelled Documents*. arXiv 2014.
> > > >
> > > > [6] Li et al. *Understanding Neural Networks through Representation Erasure*. arXiv 2017.
> > > >
> > > > [7] Wiegreffe & Pinter. *Attention is not not Explanation*. EMNLP 2019.
> > > >
> > > > [8] Mohankumar et al. *Towards Transparent and Explainable Attention Models*. ACL 2020.
> > > >
> > > > [9] Hase & Bansal. *When Can Models Learn From Explanations? A Formal Framework for Understanding the Roles of Explanation Data*. arXiv 2021.
> > > >
> > > > [10] Pruthi et al. *Evaluating Explanations: How much do explanations from the teacher aid students?* arXiv 2020.

---

> ### Author Response · Authors · 2021-08-15
> **Response to Reviewer e1cd**
>
> Hi Reviewer e1cd,
>
> Thank you again for your review! For your convenience, below is an overview of the points covered in our rebuttal:
> - Analysis of SalKG-Fine Performance
> - Quantitative Analysis of KG Saliency Explanations
> - Comparison of Fine Unit Types
> - Acknowledgment of Training Costs
> - Presentation Issues
>
> We hope that our rebuttal sufficiently addresses your concerns and questions. If there is still any uncertainty, please feel free to leave additional comments, so that we can help resolve any unclear aspects of the paper. Looking forward to discussing with you soon!
>
> Sincerely,
>
> Paper2834 Authors

---

> ### Author Response · Authors · 2021-08-25
> **Friendly reminder to respond to author rebuttal**
>
> Hi Reviewer e1cd,
>
> Thank you again for your review! Based on your thoughtful feedback, we wrote a detailed rebuttal covering the following points:
> - Analysis of SalKG-Fine Performance
> - Quantitative Analysis of KG Saliency Explanations
> - Comparison of Fine Unit Types
> - Acknowledgment of Training Costs
> - Presentation Issues
>
> We would love to hear your thoughts about our rebuttal, including whether it sufficiently addresses your concerns and questions. Any feedback is welcome and greatly appreciated!
>
> Sincerely,
>
> Paper2834 Authors

---

> > ### Author Response · Authors · 2021-08-31
> > **Follow-up reminder to respond to author rebuttal**
> >
> > Hi Reviewer e1cd,
> >
> > Just wanted to follow up on our previous reminder to respond to the rebuttal. In light of the imminent discussion deadline on **September 1**, it would be awesome to know if our rebuttal sufficiently addressed your concerns and questions. We understand that you are busy, so we would appreciate it a lot!
> >
> > Thanks!
> >
> > Paper2834 Authors

---

### Official Review · Reviewer_TCsM · 2021-07-20

**Rating:** 5
**Confidence:** 4

**Summary:**

This paper explores how saliency explanations can be used to improve the performance of knowledge-graph-augmented commonsense reasoning (CSR) models, by investigating coarse to fine KB components (an entire graph to its nodes and paths). The paper first explores how KG influences the models in making correct predictions using oracle models, which have access to true saliency explanations. Base on that, the paper proposes SALKG, a framework for KG-augmented CSR that learns from coarse and/or fine saliency explanations that are automatically created. The proposed models achieve better performance compared to the baselines used in the paper.

**Ethical Concerns:**

No concerns

**Limitations And Societal Impact:**

No concerns

**Main Review:**

The paper carefully constructed and performed detailed experiments and analyses on KG-augmented models. The detailed answers to three questions for the oracle KG-augmented models are sound and contribute to understanding the role of KG for commonsense reasoning. The proposed SALKG model is sound too, and improves the performance over the baseline models compared in the paper.  The paper is well structured and easy to follow. The methods and ideas are clearly presented.

My main concern is that the SALKG method proposed in this work is not novel enough for the conference. In addition, the entire study is rather incremental. Using attention scores to obtain explanation over the input components is not new.

The paper should provide more comparison between the proposed models (SALKG) to other strong baselines listed in the leaderboards (e.g., on CSQA and OBQA) to render a more comprehensive understanding, in addition to using in-house splits of CSQA. This limits the significance of the paper and raises some concerns; e.g., if the observations made on the proposed model can transfer to the state-of-the-art models, although it may not be straightforward to construct saliency explanation for all types of existing SOTA models.

According to Table 5 and Table 6, random explanations can achieve pretty good results compared to the baselines and can sometimes be better than heuristic explanations, which somehow shows that the saliency explanation inputs are less influential. Can the authors explain more on this?

The authors may consider providing some quantitative analysis on how good the generated explanations are in addition to the case study.

The paper is in general well written, but still contains typos or grammatical errors, e.g., on page 4, “There are many existing saliency computation method which calculate the importance”. (“method” should be “methods”).


**Time Spent Reviewing:**

7

---

> ### Author Response · Authors · 2021-08-10
> **Response to Reviewer TCsM (Part 1)**
>
> Thank you for your detailed review! We are delighted to hear that you liked our experiments, analyses, and presentation. We also appreciate your concerns and questions, each of which we address in detail below.
>
> ---
>
> ## Novelty of SalKG and our Contributions
> Regarding novelty, we believe there is some misunderstanding about our proposed SalKG method and what our contributions are.
>
> First, SalKG is an **explanation-based training procedure** which consists of the following steps:
>
> 1. Train a **"teacher" KG-augmented model, supervised only by the task labels**. The teacher model corresponds to the attention-based models introduced in prior works, but the teacher model itself is not our final model.
>
> 2. **Use some saliency method to generate explanations for the teacher model, with respect to the ground truth labels**. As described in Sec. 3 of our paper, we repurpose (via the sign-flipping in Equations 1-2) saliency explanations [1] to indicate which KG inputs most strongly influence the teacher model to predict the *correct* label. Whereas attention scores tell us which inputs the model *already* focused on, saliency explanations tell us which inputs the model *should* focus on, making saliency explanations a good choice for regularizing the model (Sec. 1).
>
> 3. **Train a "student" KG-augmented model**, which is **supervised by *both* the task labels and the saliency explanations**. In addition to the high-level supervision from the task labels, the saliency explanations provide low-level supervision by explicitly teaching the model which inputs to focus on. This student model is what we call our SalKG model. SalKG-Coarse is indeed an adaptive ensemble of the No-KG and KG models, weighted by the predicted coarse saliency explanations. However, SalKG-Fine is not an ensemble. SalKG-Fine has the same architecture as the KG model, but is trained by regularizing its attention scores to approximate the fine saliency explanations (i.e., using the attention scores as the predicted fine saliency explanations). Meanwhile, SalKG-Hybrid is an adaptive ensemble of the No-KG and SalKG-Fine models, hence using both coarse and fine saliency explanations.
>
> Second, we discuss several related lines of work and how SalKG differs from prior works:
> - **KG-Augmented Models**: Various KG-augmented model architectures have been proposed in the commonsense reasoning literature [2, 3, 4]. However, SalKG is orthogonal to these works, since SalKG is not an architecture. Rather, SalKG is a general explanation-based training algorithm that can be applied to different kinds of KG-augmented model architectures. Whereas existing KG-augmented models are supervised only by the task labels, SalKG-trained KG-augmented models are supervised by both the task labels and saliency explanations.
>
> - **Generating Explanations**: There are many existing methods for generating explanations of model behavior. These approaches include saliency methods [1] (e.g., gradient-based [5], occlusion-based [6]) and attention [7, 8]. Our paper does not propose a new method for generating explanations. Rather, we take existing saliency methods and repurpose them (via the sign-flipping in Equations 1-2) to generate KG-based explanation supervision, which we then propose to use for regularizing KG-augmented models via the SalKG approach.
>
> - **Learning from Explanations**: In the explanation-based learning literature, there have been various methods for learning from explanations (usually text-based or vision-based) [9], including using explanations as supervision for regularizing the model [10]. However, to the best of our knowledge, there has not been any work on explanation-based learning in the KG domain. While SalKG is indeed an explanation-based learning method, SalKG’s novelty comes from being the first method for learning from KG-based explanations. In particular, SalKG leverages the coarse (graph) to fine (node, path) hierarchy naturally found in KGs, in order to provide rich explanation-based learning signal to the model.
>
> In other words, the key contribution of our work is proposing to regularize the KG-augmented model with supervision from KG-based saliency explanations, which has not been done in any prior works. SalKG models are trained not only to predict the right answer, but rather to predict the right answer using the "right" inputs. We hypothesize that this additional explanation-based supervision provides a strong inductive bias which can help the model generalize better to unseen data. We will make this message more clear in the final version of the paper.
>
> ---
>
> [1] Bastings & Filippova. *The elephant in the interpretability room: Why use attention as explanation when we have saliency methods?* BlackboxNLP 2020.
>
> [2] Lin et al. *KagNet: Knowledge-Aware Graph Networks for Commonsense Reasoning*. EMNLP 2019.
>
> [3] Bosselut et al. *Dynamic Neuro-Symbolic Knowledge Graph Construction for Zero-shot Commonsense Question Answering*. AAAI 2021.
>
> [4] Yasunaga et al. *QA-GNN: Reasoning with Language Models and Knowledge Graphs for Question Answering*. NAACL 2021.
>
> [5] Denil et al. *Extraction of Salient Sentences from Labelled Documents*. arXiv 2014.
>
> [6] Li et al. *Understanding Neural Networks through Representation Erasure*. arXiv 2017.
>
> [7] Wiegreffe & Pinter. *Attention is not not Explanation*. EMNLP 2019.
>
> [8] Mohankumar et al. *Towards Transparent and Explainable Attention Models*. ACL 2020.
>
> [9] Hase & Bansal. *When Can Models Learn From Explanations? A Formal Framework for Understanding the Roles of Explanation Data*. arXiv 2021.
>
> [10] Pruthi et al. *Evaluating Explanations: How much do explanations from the teacher aid students?* arXiv 2020.

---

> > ### Author Response · Authors · 2021-08-10
> > **Response to Reviewer TCsM (Part 2)**
> >
> > ## Comparison to Other Strong Baselines on the Leaderboard
> >
> > In our main results (Tables 5-6), we show that near-SOTA KG-augmented models like MHGRN, PathGen, and RN can achieve improved performance when additionally regularized with saliency explanations (i.e., SalKG training). Plus, in Table 12 of our appendix, we compare SalKG models to a much larger number of published baselines on CSQA (in-house split) and OBQA, and we find that SalKG models maintain competitive performance. For CSQA, note that many works in the KG-augmented model literature follow the in-house split, and it has become a standard evaluation protocol for comparing KG-augmented model performance.
> >
> > Furthermore, we have trained our SalKG-Hybrid (RoBERTa-Large + MHGRN) model on the CSQA official split and submitted it to the CSQA leaderboard. We find that our SalKG models are competitive with other RoBERTa-Large based models on the leaderboard. We display the results for RoBERTa-Large based models in the table below:
> >
> > | Model | CSQA Test Acc (%) |
> > | ----------- | ----------- |
> > | RoBERTa + QA-GNN | 76.1 |
> > | RoBERTa + PEAR | 76.1 |
> > | SalKG-Hybrid (RoBERTa + MHGRN) | 74.9 |
> > | RoBERTa + KE | 73.3 |
> > | RoBERTA + HyKAS 2.0 | 73.2 |
> > | RoBERTa + KEDGN | 72.5 |
> > | RoBERTa + FreeLB | 72.2 |
> > | RoBERTa + IR | 72.1 |
> > | RoBERTa + CSPT | 69.6 |
> >
> > On the CSQA leaderboard, the overall top-performing models use either T5-3B or ALBERT-XXLarge text encoders, so we cannot directly compare our RoBERTa-Large based SalKG models to them. In our experiments, we did not use T5-3B or ALBERT-XXLarge, since they were too computationally expensive for our relatively modest GPU resources. T5-3B is extremely large and thus requires much more GPU memory than what we have. ALBERT-XXLarge uses repeating layers, so its parameter count and GPU memory footprint are similar to RoBERTa-Large’s. However, since ALBERT-XXLarge needs to iterate through each of its layers multiple times, we find that ALBERT-XXLarge’s training time is 5-8x higher than RoBERTa-Large’s in practice.
> >
> > We emphasize that SalKG is designed to be applicable to any KG-augmented model and is agnostic to the choice of text encoder or graph encoder. The saliency methods we used are general-purpose, off-the-shelf algorithms that can create explanations for arbitrary neural network models. In our experiments, we demonstrate this general-purpose usage by applying SalKG to various combinations of two Transformer text encoders (RoBERTa-Large, BERT-Base) and three very different graph encoders (MHGRN, PathGen, RN). Since T5-3B and ALBERT-XXLarge are also Transformer text encoders, there is nothing (besides compute resources) fundamentally preventing SalKG from working on T5-3B and ALBERT-XXLarge as well. In the final version of our paper, we plan to also include some ALBERT-XXLarge experiments, since ALBERT-XXLarge is still relatively feasible compared to T5-3B.

---

> > ### Author Response · Authors · 2021-08-10
> > **Response to Reviewer TCsM (Part 3)**
> >
> > ## Strong Performance of Random Explanation Baselines
> >
> > This is a good observation, and investigating it led us to discover an unfair comparison within our evaluation protocol. This unfair comparison resulted in the random baseline performance being artificially inflated. After fixing our evaluation protocol, we found that the random baseline was much weaker than before and had lower performance than SalKG overall.
> >
> > First, we reiterate that the goal of comparing Random and Heuristic to SalKG is to show that saliency explanations provide better learning signal than random or heuristic explanations. In addition, we note that Random-Hybrid and Heuristic-Hybrid are not really expected to outperform their coarse and fine counterparts, if random and heuristic explanations do not capture useful learning signal. Second, let us define: (1) non-explanation models (No-KG, KG, No-KG + KG) as models that are not regularized with any kind of explanation and (2) explanation models (Random, Heuristic, SalKG) as models that are regularized with some kind of explanation.
> >
> > In our submission, each non-explanation model’s performance was reported as the average over three non-explanation model seeds, which we denote as the non-explanation seeds. Also, recall that each explanation model is built from No-KG and/or KG models. In our submission, we used the No-KG and/or KG models from only the best non-explanation seed to build the explanation models. Then, we reported the explanation model’s performance as the average over three explanation model seeds, which we call the explanation seeds, with respect to the single best non-explanation seed. That is, we computed the non-explanation model performance by averaging over [three non-explanation seeds] = [three total seeds], while computing the explanation model performance by averaging over [one (best) non-explanation seed] * [three explanation seeds] = [three total seeds].
> >
> > Our submission makes an unfair comparison between non-explanation and explanation models, since the explanation models are based only on the best non-explanation seed but are compared to the average of all three non-explanation seeds. Instead, for each of the three non-explanation seeds, we should train the explanation model on three explanation seeds, then compute the explanation model performance by averaging over [three non-explanation seeds] * [three explanation seeds] = [nine total seeds].
> >
> > We summarize the evaluation protocol below:
> > - **Non-explanation seeds**: 1, 2, 3 (assume 1 yields best performance)
> > - **Explanation seeds**: A, B, C
> > - **Non-explanation performance**: *avg*(1, 2, 3)
> > - **Explanation performance (unfair; submission version)**: *avg*(1A, 1B, 1C)
> > - **Explanation performance (fair; updated version)**: *avg*(1A, 1B, 1C, 2A, 2B, 2C, 3A, 3B, 3C)
> >
> > Below are the updated results for the explanation models using the fair evaluation protocol:
> >
> > ### CSQA Test Acc (%)
> >
> > | Model | BERT+MHGRN | RoBERTa+MHGRN | BERT+PathGen | RoBERTa+PathGen | BERT+RN | RoBERTa+PathGen |
> > | ----------- | ----------- | ----------- | ----------- | ----------- | ----------- | ----------- |
> > | Random-Coarse | 55.04 ± 1.44 | 71.06 ± 1.09 | 55.09 ± 1.08 | 71.15 ± 1.06 | 55.15 ± 1.23 | 69.06 ± 2.96 |
> > | Random-Fine | 54.69 ± 2.54 | 73.09 ± 1.06 | 54.66 ± 0.97 | 71.26 ± 3.19 | 49.88 ± 1.75 | 69.08 ± 1.95 |
> > | Random-Hybrid | 52.43 ± 2.60 | 71.93 ± 0.77 | 55.24 ± 0.58 | 71.35 ± 0.34 | 54.36 ± 0.35 | 70.12 ± 0.35 |
> > | Heuristic-Coarse | 55.55 ± 2.29 | 72.15 ± 0.84 | 56.92 ± 0.18 | 72.57 ± 0.49 | 56.42 ± 1.11 | 71.18 ± 0.77 |
> > | Heuristic-Fine | 52.54 ± 1.67 | 71.50 ± 1.01 | 54.00 ± 1.89 | 71.11 ± 0.93 | 52.04 ± 2.13 | 65.08 ± 3.67 |
> > | Heuristic-Hybrid | 56.35 ± 0.81 | 72.58 ± 0.32 | 56.83 ± 0.48 | 71.33 ± 0.87 | 54.38 ± 3.30 | 65.07 ± 2.02 |
> > | SalKG-Coarse | 57.98 ± 0.90 | 73.64 ± 1.05 | 57.75 ± 0.77 | 73.07 ± 0.25 | 57.50 ± 1.25 | 73.11 ± 1.13 |
> > | SalKG-Fine | 54.36 ± 2.34 | 70.00 ± 0.81 | 54.39 ± 2.03 | 72.12 ± 0.91 | 54.30 ± 1.41 | 71.64 ± 1.51 |
> > | SalKG-Hybrid | 58.70 ± 0.65 | 73.37 ± 0.12 | 59.87 ± 0.42 | 72.67 ± 0.65 | 58.78 ± 0.14 | 74.13 ± 0.71 |
> >
> >
> > ### OBQA Test Acc (%)
> >
> > | Model (RoBERTa) | MHGRN | PathGen | RN |
> > | ----------- | ----------- | ----------- | ----------- |
> > | Random-Coarse | 68.11 ± 1.12 | 67.18 ± 4.13 | 65.02 ± 2.57 |
> > | Random-Fine | 57.60 ± 5.33 | 55.13 ± 7.00 | 48.53 ± 4.82 |
> > | Random-Hybrid | 68.33 ± 0.40 | 69.53 ± 0.31 | 69.27 ± 0.12 |
> > | Heuristic-Coarse | 69.24 ± 2.47 | 65.58 ± 6.08 | 64.29 ± 3.06 |
> > | Heuristic-Fine | 57.27 ± 3.76 | 51.80 ± 2.95 | 50.53 ± 3.51 |
> > | Heuristic-Hybrid | 68.47 ± 0.23 | 68.40 ± 0.00 | 68.60 ± 0.20 |
> > | SalKG-Coarse | 69.93 ± 0.56 | 70.02 ± 0.55 | 71.29 ± 0.57 |
> > | SalKG-Fine | 64.82 ± 0.97 | 51.51 ± 0.87 | 62.29 ± 0.85 |
> > | SalKG-Hybrid | 70.20 ± 0.69 | 69.80 ± 0.49 | 70.47 ± 0.91 |
> >
> >
> > ### CODAH Test Acc (%)
> >
> > | Model (RoBERTa) | MHGRN | PathGen |
> > | ----------- | ----------- | ----------- |
> > | Random-Coarse | 83.48 ± 0.91 | 84.68 ± 1.65 |
> > | Random-Fine | 74.77 ± 6.90 | 80.48 ± 1.23 |
> > | Random-Hybrid | 83.86 ± 0.69 | 83.75 ± 0.60 |
> > | Heuristic-Coarse | 82.64 ± 0.10 | 82.52 ± 0.18 |
> > | Heuristic-Fine | 82.25 ± 1.43 | 82.55 ± 2.03 |
> > | Heuristic-Hybrid | 82.16 ± 2.11 | 82.73 ± 1.51 |
> > | SalKG-Coarse | 85.79 ± 1.83 | 85.43 ± 1.88 |
> > | SalKG-Fine | 84.08 ± 1.14 | 83.36 ± 0.81 |
> > | SalKG-Hybrid | 85.17 ± 0.54 | 84.42 ± 0.64 |
> >
> > Now, with the fair evaluation protocol, we see that the Random baseline is not as strong as before. Previously, Random’s performance was artificially inflated because its average did not take into account the two weaker non-explanation seeds. Meanwhile, SalKG still outperforms all other methods overall, which shows the effectiveness of regularizing the model with saliency explanations. In the final version of the paper, we will update the tables to reflect this new evaluation protocol.

---

> > ### Author Response · Authors · 2021-08-10
> > **Response to Reviewer TCsM (Part 4)**
> >
> > ## Quantitative Analysis of KG Saliency Explanations
> >
> > We agree that the case studies would be more informative with some quantitative analysis, so we have provided two new user studies here. Overall, we did not find strong evidence that the KG saliency explanations align well with human judgment. However, we also show that alignment with human judgment may not be a good metric for explanation quality, as our human annotators have low agreement about which explanations are useful. Ultimately, the user studies do not contradict our paper’s main claims, since we argue that KG saliency explanations can be used as additional supervision for improving performance, not that the explanations are plausible (i.e., convincing) to humans.
> >
> > ### User Study 1: Coarse Saliency Explanations
> > The first user study measures how well the coarse (graph-level) explanations align with human judgment of usefulness. Given a RoBERTa+PathGen model, we begin by uniformly sampling 25 high-saliency (positive) KGs and 25 low-saliency (negative) KGs from the CSQA training set. Recall that whether a KG is high-saliency or low-saliency was determined by coarse explanations (Sec. 3) generated with respect to the given model.
> >
> > Note that each KG corresponds to one answer choice of a question, so each question in CSQA has up to five corresponding KGs. To ensure that none of the KGs in our sample come from the same question, we ended up pruning two high-saliency and two low-saliency KGs, yielding a final sample of 23 high-saliency and 23 low-saliency KGs.
> >
> > Since a KG can contain hundreds of paths, it is not feasible to ask humans to evaluate the entire KG’s usefulness. Thus, as a very rough representation of the KG, we uniformly sampled three paths from the KG. Then, for each KG, we asked ten human annotators to score each of the three paths’ usefulness for predicting the same answer choice predicted by the RoBERTa+PathGen model. To score the paths, all annotators were also given the question, correct answer, and model’s predicted answer. The paths were scored on the following 0-2 scale:
> > - 0 = definitely not useful (i.e., *this path is either irrelevant or would cause someone to NOT select the model’s predicted answer*)
> > - 1 = possibly useful (i.e., *this path provides some support for selecting the model’s predicted answer*)
> > - 2 = definitely useful (i.e., *this path provides strong support for selecting the model’s predicted answer*)
> >
> > Finally, each KG’s score is computed as the mean of its three constituent path scores. Below, we show the mean and standard deviation scores for high-saliency and low-saliency graphs. We find that the two graph types have similar mean usefulness scores, while also having relatively large standard deviations. This suggests that coarse saliency explanations do not align strongly with human judgment. One key limitation of this study is that the three sampled paths may not be representative of the entire KG. In the future, we plan to redesign the user study to provide annotators a more comprehensive representation of the KG to evaluate.
> >
> > | Graph Type | Usefulness Score |
> > | ----------- | ----------- |
> > | High-Saliency Graph | 0.929 ± 0.734 |
> > | Low-Saliency Graph | 0.935 ± 0.764 |
> >
> >
> > ### User Study 2: Fine Saliency Explanations
> > The second user study measures how well the fine (path-level) explanations align with human judgment of usefulness. Given a RoBERTa+PathGen model trained on CSQA, we begin by uniformly sampling 25 correctly answered questions and 25 incorrectly answered questions from the CSQA training set. For each question, we take the model’s predicted answer choice and the KG corresponding to the predicted answer choice, then select: (1) the path with the highest fine saliency score, (2) the path with median fine saliency score, and (3) the path with the lowest saliency score. To get finer-grained saliency signal in this study, we consider the raw fine saliency scores, instead of the binarized fine explanations actually used to regularize the model. Recall that a path’s fine saliency score (Sec. 3) is calculated with respect to the given model.
> >
> > Next, we asked ten human annotators to score each path’s usefulness for predicting the same answer choice predicted by the RoBERTa+PathGen model. Like before, to score the paths, all annotators were also given the question, correct answer, and model’s predicted answer. Again, the paths were scored on the following 0-2 scale:
> > - 0 = definitely not useful (i.e., *this path is either irrelevant or would cause someone to NOT select the model’s predicted answer*)
> > - 1 = possibly useful (i.e., *this path provides some support for selecting the model’s predicted answer*)
> > - 2 = definitely useful (i.e., *this path provides strong support for selecting the model’s predicted answer*)
> >
> > Below, we show the mean scores for high-saliency, median-saliency, and low-saliency paths. We display these scores for paths from all predictions, correct predictions, and incorrect predictions. Overall, we find that the three path types have similar mean usefulness scores, although the mean score for median-saliency paths is somewhat higher than the other two path types’. Still, the standard deviations for all scores are relatively large, so this trend may not be meaningful. These results suggest that fine saliency explanations do not align strongly with human judgment. Additionally, we find that the path usefulness scores for correct predictions tend to be higher than those from incorrect predictions. This makes sense, since, intuitively, a model is more likely to predict the correct answer if it is using more useful knowledge as context.
> >
> > | Path Type | Usefulness Score (All Preds) | Usefulness Score (Correct Preds) | Usefulness Score (Incorrect Preds) |
> > | ----------- | ----------- | ----------- | ----------- |
> > | High-Saliency Graph | 1.091 ± 0.805 | 1.298 ± 0.782 | 0.884 ± 0.776 |
> > | Med-Saliency Graph | 1.222 ± 0.769 | 1.320 ± 0.729 | 1.124 ± 0.798 |
> > | Low-Saliency Graph | 1.060 ± 0.733 | 1.182 ± 0.730 | 0.938 ± 0.717 |
> >
> >
> > ### Inter-Annotator Agreement
> >
> > Here, we measure inter-annotator agreement for both user studies, using Fleiss’ kappa. For the user study of coarse explanations, the kappa score is 0.2089, which is on the borderline of slight agreement and fair agreement. For the user study of fine explanations, the kappa score is 0.1296, which indicates slight agreement.
> >
> > These low kappa scores show that even humans can hardly agree on whether the coarse/fine explanations are useful. Therefore, it may not always be beneficial to measure explanation quality in terms of alignment with human judgment. Moreover, this shows that weak alignment with human judgment does not necessarily imply poor explanation quality.
> >
> > | User Study | Fleiss' Kappa |
> > | ----------- | ----------- |
> > | Coarse Explanations | 0.2089 |
> > | Fine Explanations | 0.1296 |
> >
> >
> > ### Analysis
> > In our user studies, we did not find that coarse/fine saliency explanations align well with human judgment. However, we also found that human annotators had very low agreement about the usefulness of the explanations, which suggests that alignment with human judgment may not be the best measure of explanation quality.
> >
> > In light of this, we emphasize that the user study results do not contradict our paper’s conclusions, as our work does not claim that the generated saliency explanations are plausible. Rather, we merely claim that using KG-based saliency explanations as additional supervision to regularize KG-augmented models can yield higher performance. Our work appeals to the view that an explanation’s quality should be measured by how well it distills knowledge for improving performance on some task [1]. Furthermore, the results of our user studies are actually in line with the conclusions from [2], which found that KG-augmented models can effectively leverage KG information to improve performance, but in a manner that does not make sense to humans. In the final version of the paper, we will include the new user studies and analysis, so that readers can better understand the role and limitations of the KG saliency explanations.
> >
> > ---
> >
> > [1] Pruthi et al. *Evaluating Explanations: How much do explanations from the teacher aid students?* arXiv 2020.
> >
> > [2] Raman et al., *Learning to Deceive Knowledge Graph Augmented Models via Targeted Perturbation*. ICLR 2021.

---

> > ### Author Response · Authors · 2021-08-10
> > **Response to Reviewer TCsM (Part 5)**
> >
> > ## Spelling/Grammar Errors
> > Thank you for pointing out these mistakes. We will make sure to fix all spelling/grammar errors in the final version of the paper.

---

> > ### Author Response · Authors · 2021-08-15
> > **Overview of Response to Reviewer TCsM**
> >
> > Hi Reviewer TCsM,
> >
> > Thank you again for your review! For your convenience, below is an overview of the points covered in our rebuttal:
> > - Novelty of SalKG and our Contributions
> > - Comparison to Other Strong Baselines on the Leaderboard
> > - Strong Performance of Random Explanation Baselines
> > - Quantitative Analysis of KG Saliency Explanations
> > - Spelling/Grammar Errors
> >
> > We hope that our rebuttal sufficiently addresses your concerns and questions. If there is still any uncertainty, please feel free to leave additional comments, so that we can help resolve any unclear aspects of the paper. Looking forward to discussing with you soon!
> >
> > Sincerely,
> >
> > Paper2834 Authors

---

> ### Author Response · Authors · 2021-08-25
> **Friendly reminder to respond to author rebuttal**
>
> Hi Reviewer TCsM,
>
> Thank you again for your review! Based on your thoughtful feedback, we wrote a detailed rebuttal covering the following points:
> - Novelty of SalKG and our Contributions
> - Comparison to Other Strong Baselines on the Leaderboard
> - Strong Performance of Random Explanation Baselines
> - Quantitative Analysis of KG Saliency Explanations
> - Spelling/Grammar Errors
>
> We would love to hear your thoughts about our rebuttal, including whether it sufficiently addresses your concerns and questions. Any feedback is welcome and greatly appreciated!
>
> Sincerely,
>
> Paper2834 Authors

---

> > ### Author Response · Authors · 2021-08-31
> > **Follow-up reminder to respond to author rebuttal**
> >
> > Hi Reviewer TCsM,
> >
> > Just wanted to follow up on our previous reminder to respond to the rebuttal. In light of the imminent discussion deadline on **September 1**, it would be awesome to know if our rebuttal sufficiently addressed your concerns and questions. We understand that you are busy, so we would appreciate it a lot!
> >
> > Thanks!
> >
> > Paper2834 Authors

---

### Author Response · Authors · 2021-09-05
**CSQA leaderboard results using ALBERT**

In a previous comment (https://openreview.net/forum?id=FUxXaBop-J_&noteId=ZDvPGJ5myEf), we compared SalKG-Hybrid (RoBERTa + MHGRN)	to other RoBERTa-based models on the CSQA leaderboard. We also explained that computational limitations prevented us from presenting results on ALBERT at that time.

Since then, we have trained SalKG-Hybrid (ALBERT + PathGen) on the CSQA official split and evaluated it on the CSQA leaderboard, achieving a test accuracy of 75.9%. Meanwhile, ALBERT + PathGen achieved a test accuracy of 75.6% on the CSQA leaderboard. This result suggests that the proposed SalKG training procedure can yield some improvements over baselines without explanation-based regularization.

---

### Decision · Program_Chairs · 2021-09-27

**Decision:**

Accept (Poster)

**Comment:**

This paper proposes SalKG, a framework for using saliency explanations to learn to better use knowledge graphs (KG) for commonsense reasoning (CSR) tasks. Saliency is considered at two levels: (1) the usefulness of the KG overall called "coarse" and (2) the usefulness of specific nodes and paths in the KG called "fine". To create coarse explanations the authors introduce an ensemble based saliency method for comparing models with and without access to a KG. Fine explanations use off the shelf saliency methods. This leads to SalKG, where explanations for the training data of a CSR task are used to regularize the learning of the attention mechanisms in the CSR architecture. Experiments on the CSQA, OBQA, and CODAH experiments show that the hybrid model significantly outperforms prior work baselines using both BERT and RoBERTa base models.

Many of the reviewers agreed on the strengths of the paper. They found that method is sound and leads to significant results. They generally all agreed that the paper was well written and clear in its presentation. They also remarked on the quality of the analysis, such as the oracle-based approach for analyzing how learning from saliency can lead to improved accuracy.

The reviewers also raised several main concerns. First, some expressed concern that the method was not sufficiently "novel." As the authors clarified in their response, while attention and saliency are both well studied, the significance of their work is using saliency as a training signal for incorporating KGs into CSR.

The other main concern regarded comparison with a wider range of baselines, as well as using the official train/test split for CSQA. In the response, the authors addressed these concerns. First, additional comparisons on the same datasets are included in the appendix. Second, they also submitted a model to the official CSQA test server. Their RoBERTa based model places third among other RoBERTa based models. The authors are encouraged to include these additional results in the final version of the paper.

Ultimately, the strengths outweigh the potential weaknesses. As one reviewer noted, this work has potential application beyond CSR to other tasks like reading comprehension, and it is likely to be of interest to the wider community.